# Learning Optimal Flows for Non-Equilibrium Importance Sampling

**Yu Cao**

Courant Institute of Mathematical Sciences, New York University

`yucaoyc@outlook.com`

**Eric Vanden-Eijnden**

Courant Institute of Mathematical Sciences, New York University

`eve2@cims.nyu.edu`

## Abstract

Many applications in computational sciences and statistical inference require the computation of expectations with respect to complex high-dimensional distributions with unknown normalization constants, as well as the estimation of these constants. Here we develop a method to perform these calculations based on generating samples from a simple base distribution, transporting them by the flow generated by a velocity field, and performing averages along these flowlines. This non-equilibrium importance sampling (NEIS) strategy is straightforward to implement and can be used for calculations with arbitrary target distributions. On the theory side, we discuss how to tailor the velocity field to the target and establish general conditions under which the proposed estimator is a perfect estimator with zero-variance. We also draw connections between NEIS and approaches based on mapping a base distribution onto a target via a transport map. On the computational side, we show how to use deep learning to represent the velocity field by a neural network and train it towards the zero variance optimum. These results are illustrated numerically on benchmark examples (with dimension up to 10), where after training the velocity field, the variance of the NEIS estimator is reduced by up to 6 orders of magnitude than that of a vanilla estimator. We also compare the performances of NEIS with those of Neal's annealed importance sampling (AIS).

## 1 Introduction

Given a potential function $U_1 : \Omega \to \mathbb{R}$ on the domain $\Omega \subseteq \mathbb{R}^d$, the main goal of this paper is to evaluate

$$\mathcal{Z}_1 := \int_\Omega e^{-U_1(x)} \, \mathrm{d}x. \tag{1}$$

The calculations of such integrals arise in many applications from several scientific fields. For instance, $\mathcal{Z}_1$ is known as the partition function in statistical physics [21], where it is used to characterize the thermodynamic properties of a system with energy $U_1$, and as the evidence in Bayesian statistics, where it is used for model selection [12].

When the dimension $d$ of the domain $\Omega$ is large, standard numerical quadrature methods are inapplicable to (1) and the method of choice to estimate $\mathcal{Z}_1$ is Monte-Carlo sampling [22, 8]. This requires expressing $\mathcal{Z}_1$ as an expectation, which can be done, e.g., by realizing that

$$\mathcal{Z}_1 = \mathbb{E}_0\big[e^{-U_1}/\rho_0\big], \tag{2}$$

36th Conference on Neural Information Processing Systems (NeurIPS 2022).

where $\mathbb{E}_0$ denotes the expectation with respect to the probability density function $\rho_0 > 0$. If $\rho_0$ is both known (i.e., we can evaluate it pointwise in $\Omega$, normalization factor included) and simple to sample from, we can build an estimator for $\mathcal{Z}_1$ by replacing the expectation on the right hand side of (2) by the empirical average of $e^{-U_1}/\rho_0$ over samples drawn from $\rho_0$. Unfortunately, finding a density $\rho_0$ that has the two properties above is hard: unless $\rho_0$ is well-adapted to $e^{-U_1}$, the estimator based on (2) is terrible in general, with a standard deviation that is typically much larger than its mean or even infinite. A similar issue arises if we want to estimate the expectation $\mathbb{E}_0 f$ of some function $f : \Omega \to \mathbb{R}$, and the two problems are in fact connected when $f > 0$, since the second reduces to (2) for $U_1 = -\log(f\rho_0)$.

These difficulties have prompted the development of importance sampling strategies [14] whose aim is to produce estimators with a reasonably low variance for $\mathcal{Z}_1$ or $\mathbb{E}_0 f$. These include for example umbrella sampling [43, 41], replica exchange (aka parallel tempering) [14, 23], nested sampling [37, 38], in which the estimation of $\mathcal{Z}_1$ is factorized into the calculation of several expectations of the type (2), but with better properties, that can then be recombined using thermodynamic integration [17] or Bennett acceptance ratio method [6].

Complementary to these equilibrium techniques, non-equilibrium sampling strategies have also been introduced for the calculation of (1). For example, Neal's annealed importance sampling (AIS) [26] based on the Jarzynski equality [15, 16, 1] calculates $\mathcal{Z}_1$ using properly weighted averages over sequences of states evolving from samples from $\rho_0$, without requiring that the kernel used to generate these states be in detailed-balance with respect to either $\rho_0$, or $\rho_1 := e^{-U_1}/\mathcal{Z}_1$, or any density interpolating between these two. Instead, the weight factors are based on the probability distribution of the sequence of states in the path space. Other non-equilibrium sampling strategies in this vein include bridge and path sampling [13], and sequential Monte Carlo (SMC) sampling [24, 2].

In this paper, we analyze another non-equilibrium importance sampling (NEIS) method, originally introduced in [33]. NEIS is based on generating samples from a simple base density $\rho_0$, then propagating them forward and backward in time along the flowlines of a velocity field, and computing averages along these trajectories—the basic idea of the method is to use the flow induced by this velocity field to sweep samples from $\rho_0$ through regions in $\Omega$ that contribute most to the expectation. As shown in [33] and recalled below, this procedure leads to consistent estimators for the calculation of $\mathcal{Z}_1$ or $\mathbb{E}_0 f$ via a generalization of (2). One advantage of the method, which is a rare feature among importance sampling strategies, is that it leads to estimators that always have lower variance than the vanilla estimator based on (2) [33]. The question we investigate in this paper is how low their variance can be made, both in theory and in practice. Our **main contributions** are:

- Under mild assumptions on $U_1$ and $\rho_0$, we show that if the NEIS velocity field is the gradient of a potential that satisfies a Poisson equation, the NEIS estimator for $\mathcal{Z}_1$ has zero variance.

- Under the same assumptions, we show that this optimal flow can be used to construct a perfect transport map from $\rho_0$ to $\rho_1$. This allows us to compare NEIS with importance sampling strategies involving transport maps like normalizing flows (NF) that have recently gained popularity [32, 19, 30], and highlight some potential advantages of the former over the latter.

- On the practical side, we derive variational problems for the optimal velocity field in NEIS, and show how to solve these problems by approximating the velocity by a neural network and optimizing its parameters using deep learning training strategies, similar to what is done with neural ODE [9].

- We illustrate the feasibility and usefulness of this approach by testing it on numerical examples. First we consider Gaussian mixtures in up to 10 dimensions. In this context, we show that training the velocity used in NEIS allows to reduce the variance of a vanilla estimator using a standard Gaussian distribution as $\rho_0$ by up to 6 orders of magnitude. Second we study Neal's 10-dimensional funnel distribution [27, 2], for which the variance of the vanilla importance sampling method is infinity; training a linear dynamics with 2 parameters in NEIS can lead to an estimator with more accurate estimate of $\mathcal{Z}_1$. In these examples we also show that after training, NEIS leads to estimators with lower variance than AIS [26].

**Related works.** The idea of transporting samples from $\rho_0$ to lower the variance of the vanilla estimator based on (2) is also at the core of importance sampling strategies using normalizing flows (NF) [40, 39, 32, 19, 30, 46, 45, 29, 25]. The type of transport used in NF-based method is however different in nature from the one used in NEIS. With NF, one tries to construct a map that transforms each sample from $\rho_0$ into a sample from the target $\rho_1 = e^{-U_1}/\mathcal{Z}_1$. In contrast, NEIS uses samples

from $\rho_0$ as initial conditions to generate trajectories, and uses the data along these entire trajectories to build an estimator. Intuitively, this means that samples likely on $\rho_0$ must become likely on $\rho_1$ sometime along these trajectories rather than at a given time specified beforehand, which is easier to enforce.

NEIS bears similarities with Neal's AIS [26], except that in NEIS the sampling is done once from $\rho_0$ to generate deterministic trajectories to gather data for the estimator, whereas AIS uses random trajectories. There are some methods based on AIS that optimize the transition kernel: for instance, stochastic normalizing flows (SNF) proposed in [46] incorporate NF between annealing steps; and annealed flow transport (AFT) in [2] combines NF with the sequential Monte Carlo method to provide optimized flow transport. These approaches require learning several maps along the annealed transition, whereas the NEIS herein only needs to learn a single flow dynamics.

A time-discrete version of NEIS, termed NEO, was proposed in [42]. The current implementation of NEO iterates on a map that needs to be prescribed beforehand, but this map could perhaps be optimized using a strategy similar to the one proposed here.

From a practical standpoint, the idea of optimizing the velocity field in NEIS using a neural network approximation for this field can be viewed as an application of neural ODEs [9] that uses the variance of the NEIS estimator as the objective function to minimize. The nature of this objective poses specific challenges in the training procedure, which we investigate here.

**Notations.** For symmetry, we denote $\rho_0(x) = e^{-U_0(x)}$ with $U_0 = -\log \rho_0 : \Omega \to \mathbb{R}$ and $\mathcal{Z}_0 = \int_\Omega e^{-U_0(x)} \, \mathrm{d}x = 1$. We denote a $d$-dimensional vector filled with zeros as $\mathbf{0}_d$ and the $d \times d$ identity matrix as $\mathbb{I}_d$. $\langle \cdot, \cdot \rangle$ is the Euclidean inner product in $\mathbb{R}^d$. We assume that the domain $\Omega$ is either an open and connected subset of $\mathbb{R}^d$ with smooth boundary or a $d$-dimensional torus (without boundary). We denote by $\mathcal{N}(\mu, \Sigma)$ the multivariate Gaussian density with mean $\mu$ and covariance matrix $\Sigma$. For two functions $f, g : D \to \mathbb{R}$ where $D$ is a domain of interest, the notation $f \lesssim g$ means that there exists a constant $C > 0$ such that $f(x) \leq Cg(x)$ for any $x \in D$. Suppose $\boldsymbol{T} : \Omega \to \mathbb{R}^d$ is a map and $\rho$ is a distribution, then the pushforward distribution of $\rho$ by the map $\boldsymbol{T}$ is denoted as $\boldsymbol{T} \# \rho$. The notation $|\cdot|$ is the usual $\ell_2$ norm for vectors and $\|\cdot\|$ is the matrix norm or functional norm.

## 2 Flow-based NEIS method

Here we recall the main ingredients of the non-equilibrium importance sampling (NEIS) method proposed in [33]. Let $\boldsymbol{b} : \Omega \to \mathbb{R}^d$ be a velocity field which we assume belongs to the vector space

$$\mathfrak{B} := \left\{ \boldsymbol{b} \in C^\infty\left(\overline{\Omega}, \mathbb{R}^d\right) \,\Big|\, \boldsymbol{b} \cdot \boldsymbol{n}|_{\partial\Omega} = 0, \, \sup_{x \in \Omega} |\nabla \boldsymbol{b}(x)| < \infty \right\}, \tag{3}$$

where $\boldsymbol{n}$ is the normal vector at the boundary $\partial\Omega$. Define the associated flow map $\boldsymbol{X}_t : \Omega \to \Omega$ via

$$\frac{\mathrm{d}}{\mathrm{d}t} \boldsymbol{X}_t(x) = \boldsymbol{b}\left(\boldsymbol{X}_t(x)\right), \qquad \boldsymbol{X}_0(x) = x, \tag{4}$$

and let $\mathcal{J}_t(x)$ be the Jacobian of this map:

$$\mathcal{J}_t(x) := |\det\left(\nabla_x \boldsymbol{X}_t(x)\right)| \equiv \exp\left(\int_0^t \nabla \cdot \boldsymbol{b}\left(\boldsymbol{X}_s(x)\right) \, \mathrm{d}s\right). \tag{5}$$

Finally, let us denote

$$\mathcal{F}_t^{(k)}(x) := e^{-U_k(\boldsymbol{X}_t(x))} \mathcal{J}_t(x) \tag{6}$$

for $k \in \{0, 1\}$, $x \in \Omega$ and $t \in \mathbb{R}$. NEIS is based on the following result, proven in Appendix B.1:

**Proposition 2.1.** *If $\boldsymbol{b} \in \mathfrak{B}$, then for any $-\infty < t_- < t_+ < \infty$, we have*

$$\mathcal{Z}_1 = \mathbb{E}_0 \mathcal{A}_{t_-, t_+}, \tag{7}$$

*where*

$$\mathcal{A}_{t_-, t_+}(x) := \int_{t_-}^{t_+} \frac{\mathcal{F}_t^{(1)}(x)}{\int_{t-t_+}^{t-t_-} \mathcal{F}_s^{(0)}(x) \, \mathrm{d}s} \, \mathrm{d}t. \tag{8}$$

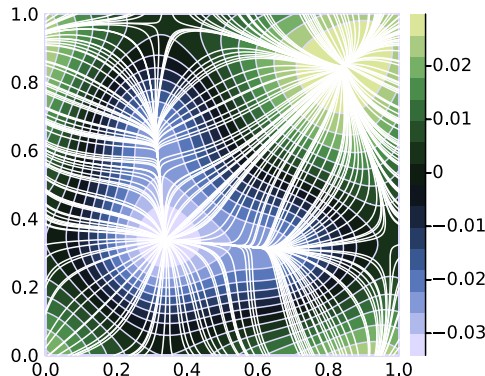

Figure 1: Contour plot of $V$ and flowlines of $\boldsymbol{b} = \nabla V$, where $V$ solves the Poisson's equation (11) with $\mathcal{D} = 1$, assuming that $\rho_0 = 1$ and $\rho_1$ is a mixture density with 3 modes; see (54). With this $\boldsymbol{b} = \nabla V$, we have $\mathcal{A}(x) = \mathcal{Z}_1$ for almost all $x \in [0,1]^2$.

*In addition, if*

$$\lim_{\substack{t_- \to -\infty \\ t_+ \to \infty}} \mathcal{A}_{t_-, t_+}(x) = \mathcal{A}(x) := \frac{\int_{\mathbb{R}} \mathcal{F}_t^{(1)}(x)\,\mathrm{d}t}{\int_{\mathbb{R}} \mathcal{F}_t^{(0)}(x)\,\mathrm{d}t} \tag{9}$$

*exists for almost all $x \sim \rho_0$, then*

$$\mathcal{Z}_1 = \mathbb{E}_0 \mathcal{A}. \tag{10}$$

When $\boldsymbol{b} = \boldsymbol{0}_d$, or $t_- \uparrow 0$ and $t_+ \downarrow 0$, (7) reduces to (2). The aim, however, is to choose $\boldsymbol{b}$ so that the estimator based on (7) has a lower variance than the one based (2): we will show below that this can indeed be done. For now, note that Jensen's inequality implies that an estimator based on (10) for *any $\boldsymbol{b}$* has lower variance than the one based (2); see [33] or Proposition H.3 below for details.

Note also that, if one allows the magnitude of the flow $\boldsymbol{b}$ to be arbitrarily large, the finite-time NEIS (8) will behave like the infinite-time NEIS (9); such a relation will be discussed and elaborated in Appendix B.4.

Finally, note that the estimator (10) based on (9) is invariant with respect to the parameterization of the flowlines generated by the dynamics $\boldsymbol{b}$, as shown by the following result proved in Appendix B.2:

**Proposition 2.2** (An invariance property)**.** *Suppose $\boldsymbol{b}, \alpha\boldsymbol{b} \in \mathfrak{B}$, where $\alpha \in C^\infty(\Omega, \mathbb{R})$ satisfies $\inf_{x \in \Omega} \alpha(x) > 0$. Then the fields $\boldsymbol{b}$ and $\alpha\boldsymbol{b}$ generate the same flowlines, and $\mathcal{A}^{\boldsymbol{b}} = \mathcal{A}^{\alpha\boldsymbol{b}}$ where $\mathcal{A}^{\boldsymbol{b}}$ and $\mathcal{A}^{\alpha\boldsymbol{b}}$ are the function defined in (9) using $\boldsymbol{b}$ and $\alpha\boldsymbol{b}$, respectively.*

## 3  Optimal NEIS

The NEIS estimator for (10) is unbiased no matter what $\boldsymbol{b}$ is. However, its performance relies on the choice of $\boldsymbol{b}$. Therefore, a natural question is to find the field $\boldsymbol{b}$ that achieves the largest variance reduction. The next result shows that an optimal $\boldsymbol{b}$ exists that leads to a zero-variance estimator:

**Proposition 3.1** (Existence of zero-variance dynamics)**.** *Assume that $\Omega = [0,1]^d$ is a torus and $U_0, U_1 \in C^\infty(\Omega, \mathbb{R})$. Let $\mathcal{D} : \Omega \to (0, \infty)$ be some smooth positive function with $\inf_{x \in \Omega} \mathcal{D}(x) > 0$, and suppose that $V \in C^\infty(\Omega)$ solves the following Poisson's equation on $\Omega$*

$$\nabla \cdot (\mathcal{D}\nabla V) = \rho_1 - \rho_0, \qquad \text{with} \quad \int_\Omega V(x)\,\mathrm{d}x = 0. \tag{11}$$

*If the solution $V$ is a Morse function, then $\boldsymbol{b} = \nabla V$ is a zero-variance velocity field: that is, if we use it to define (9), we have*

$$\int_{\mathbb{R}} \rho_0(\boldsymbol{X}_s(x))\mathcal{J}_s(x)\,\mathrm{d}s = \int_{\mathbb{R}} \rho_1(\boldsymbol{X}_s(x))\mathcal{J}_s(x)\,\mathrm{d}s, \qquad \text{for almost all } x \sim \rho_0, \tag{12}$$

*and as a result*

$$\mathcal{A}(x) = \mathcal{Z}_1 \quad \textit{for almost all } x \sim \rho_0. \tag{13}$$

This proposition is proven in Appendix E, where we also make a connection between (11) and Beckmann's transportation problem. We stress that the optimal $\boldsymbol{b}$ specified in Proposition 3.1 is not unique (see Proposition E.9): however, we show below in Proposition 4.1 that, under certain conditions, all local minima of the variance (viewed as a functional of $\boldsymbol{b}$) are global minima. We also note that the assumption that the solution is a Morse function is mostly a technicality, as discussed in Appendix E.3. Similarly, we consider the torus in Proposition 3.1 for simplicity mainly; we expect that the proposition will hold in general when $\Omega$ has compact closure or even when $\Omega = \mathbb{R}^d$, see Appendix E for examples including that of Gaussian mixture distributions.

For illustration, the contour plot of $V$ and the flowlines of $\boldsymbol{b} = \nabla V$ are shown in Figure 1 in a simple example in a two-dimensional torus where $\rho_0 = 1$ and $\rho_1$ is a mixture density with 3 modes; their explicit expressions are given in (54); in this example, we solved (11) numerically with $\mathcal{D} = 1$, see Appendix E.5 for more details. Some other examples where the zero-variance dynamics is explicit are discussed in Appendix F.

**Connection to transport maps and normalizing flows.** The zero-variance dynamics provides a transport map $\boldsymbol{T}$ from $\rho_0$ to $\rho_1$, as shown in:

**Proposition 3.2** (Existence of a perfect generator)**.** *Suppose* $\mathcal{D} = 1$ *for simplicity. Under the same assumption as in Proposition 3.1, let $V$ be the Morse function solving* (11) *and $\boldsymbol{b} = \nabla V$ the associated zero-variance dynamics. Then there exists a continuously differentiable function $\varkappa$ (defined almost everywhere on $\Omega$) such that*

$$\int_{-\infty}^0 \rho_0(\boldsymbol{X}_s(x)) \mathcal{J}_s(x) \, \mathrm{d}s = \int_{-\infty}^{\varkappa(x)} \rho_1(\boldsymbol{X}_s(x)) \mathcal{J}_s(x) \, \mathrm{d}s. \tag{14}$$

*Furthermore, the map $\boldsymbol{T}(x) := \boldsymbol{X}_{\varkappa(x)}(x)$ is a transport map from $\rho_0$ to $\rho_1$, i.e., $\boldsymbol{T} \# \rho_0 = \rho_1$.*

The proof is given in Appendix G.1. Note that we consider again $\boldsymbol{b} = \nabla V$ on the torus for technical simplicity: the statement of the proposition should hold in general for a zero-variance dynamics $\boldsymbol{b}$. The solution of (14) is particularly simple in one-dimension, where we can take $\boldsymbol{b}(x) = 1$, and straightforwardly verify that $\varkappa(x) = \boldsymbol{T}(x) - x$ with

$$\boldsymbol{T}(x) = F_1^{-1}(F_0(x)) \qquad \text{where} \quad F_i(x) = \int_{-\infty}^x \rho_i(y) \, \mathrm{d}y, \ \ i = 0, 1. \tag{15}$$

We also illustrate the statement of Proposition 3.2 via numerical examples in Appendix G.2.

To avoid confusion, we stress that we will *not* use the transport map $\boldsymbol{T}$ of Proposition 3.2 in the examples below. Indeed, using this map would require identifying $\varkappa$, which introduces an unnecessary additional calculation which we can avoid using the NEIS estimator directly. In addition, the NEIS estimator will likely have better properties than those based on transport maps, as we can think of NEIS as using a time-parameterized family of transport maps rather than a single one. In particular, the variance of the NEIS estimator will be small if samples likely on $\rho_0$ become likely on $\rho_1$ *sometime* along the NEIS trajectories, rather than at the same fixed time for all samples. The former seems easier to fulfill than the latter. For example, in one-dimension, the NEIS estimator has zero variance for any $\boldsymbol{b}$ bounded away from zero, whereas building a transport map from $\rho_0$ to $\rho_1$ is already nontrivial in that simple case since it requires solving (15).

## 4 Variational formulations

The Poisson equation (11) admits a variational formulation:

$$\min_V \int_\Omega \tfrac{1}{2} |\nabla V|^2 \mathcal{D} + V(\rho_1 - \rho_0). \tag{16}$$

If $\mathcal{D}$ is chosen to be a probability density function (for example $\mathcal{D} = \rho_0$ or $\mathcal{D} = \frac{1}{2}(\rho_1 + \rho_0)$), the two terms in the objective in (16) are expectations which can be estimated via sampling (using, e.g., direct sampling for the expectation with respect to $\rho_0$ and NEIS for the one with respect to $\rho_1$). This

means that we can in principle use an MCMC estimator of (16) as empirical loss, and minimize it over all $V$ in some parametric class. Here however, we will follow a different strategy that allows us to directly parametrize $\boldsymbol{b}$ instead of $V$ (i.e., relax the requirement that $\boldsymbol{b} = \nabla V$) and simply use the variance of the estimator as objective function.

Specifically, since we quantify the performance of the estimators based on (7) and (10) by their variance, we can view these quantities as functionals of $\boldsymbol{b}$ that we wish to minimize. Since the estimators are unbiased, these objectives are

$$
\begin{aligned}
\mathrm{Var}_{t_-,t_+}(\boldsymbol{b}) &= \mathcal{M}_{t_-,t_+}(\boldsymbol{b}) - \mathcal{Z}_1^2, \qquad \text{(finite-time)}; \\
\mathrm{Var}(\boldsymbol{b}) &= \mathcal{M}(\boldsymbol{b}) - \mathcal{Z}_1^2, \qquad\quad \text{(infinite-time)},
\end{aligned}
\tag{17}
$$

where we defined the second moments $\mathcal{M}_{t_-,t_+}(\boldsymbol{b}) := \mathbb{E}_0\big[|\mathcal{A}_{t_-,t_+}|^2\big]$ and $\mathcal{M}(\boldsymbol{b}) := \mathbb{E}_0\big[|\mathcal{A}|^2\big]$. With the finite-time objective, we know that with $\boldsymbol{b} = \boldsymbol{0}_d$, (7) reduces to (2). Therefore, minimizing $\mathcal{M}_{t_-,t_+}(\boldsymbol{b})$ over $\boldsymbol{b}$ by gradient descent starting from $\boldsymbol{b}$ near $\boldsymbol{0}_d$ will necessarily produce a better estimator: while we cannot guarantee that the variance of this optimized estimator will be zero, the experiments conducted below indicate that it can be a several order of magnitude below that of the vanilla estimator.

For the infinite-time objective, we know that for any $\boldsymbol{b}$, (9) leads to an estimator with a lower variance than the one based on (2) [33]. Minimizing $\mathrm{Var}(\boldsymbol{b})$ over $\boldsymbol{b}$ using gradient descent leads to a local minimum; the next result shows that all such local minima are global:

**Proposition 4.1** (Global minimum). *Under some technical assumptions listed in Proposition H.1, if $\boldsymbol{b}_* \in \mathfrak{B}$ is a local minimum of $\mathrm{Var}(\cdot)$ where the functional derivative of $\mathrm{Var}(\boldsymbol{b})$ with respect to $\boldsymbol{b}$ vanishes, i.e., $\delta \mathrm{Var}(\boldsymbol{b}_*)/\delta \boldsymbol{b} = \boldsymbol{0}_d$ on $\Omega$, then $\boldsymbol{b}_*$ is a global minimum and $\mathrm{Var}(\boldsymbol{b}_*) = 0$.*

The expression of the functional derivative $\delta \mathrm{Var}(\boldsymbol{b}_*)/\delta \boldsymbol{b}$ is given in Proposition D.1. The technical assumptions under which Proposition 4.1 holds are explained in Appendix A and the proof is given in Appendix H.

## 5 Training towards the optimal $b$

Here we discuss how to use deep learning techniques to find the optimal $\boldsymbol{b}$; these techniques will be illustrated on numerical examples in Section 6. Some technical details are deferred to Appendix I.

**Objective.** We use the finite-time objective $\mathcal{M}_{t_-,t_+}(\boldsymbol{b})$ in (7) with $t_- \in [-1,0], t_+ = t_- + 1$. Two natural choices are $t_- = 0$ and $t_- = -1/2$, which will be used below in the numerical experiments. This leads to no loss of generality *a priori* since in the training scheme we put no restriction on the magnitude that $\boldsymbol{b}$ can reach, and with large $\boldsymbol{b}$ the flow line can travel a large distance even during $t \in [-1,1]$ (the range of integration in $s, t$ in (8)); see the discussion in Appendix B.4 for more details. In practice, we use a time-discretized version of (8) with $2N_t$ discretization points, and use the standard Runge-Kutta scheme of order 4 (RK4) to integrate the ODE (4) over $t \in [-1,1]$ using uniform time step ($\Delta t = 1/N_t$). We note that this numerical discretization introduces a bias. However, this bias can be systematically controlled by changing the time step or using higher order integrators. In our experiments, we observed that the RK4 integrator led to negligible errors, see Table 5.

**Neural architecture.** In our experiments, we either parameterize $\boldsymbol{b}$ by a neural network directly, or we assume that $\boldsymbol{b}$ is a gradient field,

$$
\boldsymbol{b} = \nabla V \qquad \text{(gradient form)},
$$

and parameterize the potential $V$ by a neural network. We always use an $\ell$-layer neural network with width $m$ for all inner layers; therefore, from now on, we simply refer the neural network structure by a pair $(\ell, m)$; see Appendix I.2 for more details. When we parametrize the potential function $V$ instead of $\boldsymbol{b}$, the only difference is that the output dimension of the neural network becomes $1$ instead of $d$. The activation function is chosen as the *softplus* function (a smooth version of ReLU) that gave better empirical results compared to the sigmoid function. At initialization the neural parameters were randomly generated. Theoretical results about the gradient of $\mathcal{M}_{t_-,t_+}(\boldsymbol{b})$ with respect to parameters are given in Appendix C and corresponding numerical implementations are explained in Appendix I.

**Direct training method.** We minimize $\mathcal{M}_{t_-,t_+}(\boldsymbol{b})$ with respect to the parameters in the neural network using stochastic gradient descent (SGD) in which we evaluate the loss and its gradient empirically using mini-batches of data drawn from $\rho_0$ at every iteration step. For simplicity, we choose $\rho_0$ as the standard Gaussian in Section 6 below.

**Assisted training method.** When local minima of $U_1$ are far away from the local minimum of $U_0$, the direct training method by sampling data from $\rho_0$ and minimizing $\mathcal{M}_{t_-,t_+}(\boldsymbol{b})$ fails, because the flowlines may not reach the importance region of $\rho_1$ due to poor initializations of $\boldsymbol{b}$. More specifically, if along almost all trajectories, $e^{-U_1(\boldsymbol{X}_s(x))} \approx 0$ for $s \in [-1, 1]$, then with large probability $\mathcal{A}_{t_-,t_+}(x) \approx 0$ where $x \sim \rho_0$; as a result, the empirical variance of the estimator can be extremely small if the number of samples is small, while the true variance could be extremely large. Such a *mode collapse* phenomenon is common in rare event simulations.

To get around this difficulty, recall that ideally we would like to find a dynamics $\boldsymbol{b}$ such that $\mathcal{A}$ is approximately a constant function in the infinite-time case. That is, if $\boldsymbol{b}$ is a zero-variance dynamics, then $\mathcal{A}$ is a constant function and $\mathbb{E}_p\big[(\mathcal{A} - (\mathbb{E}_p\mathcal{A})^2)\big] = 0$ for *any* distribution $p$, and we are not constrained to use the base distribution $\rho_0$ and minimize the functional $\boldsymbol{b} \mapsto \mathrm{Var}(\boldsymbol{b})$ in (17). Motivated by this idea, we use an assisted training scheme in which, at iteration $i$ of SGD, the loss function is

$$\mathbb{E}_{p_i}\Big[\big(\mathcal{A}_{t_-,t_+} - \mathbb{E}_{p_i}\mathcal{A}_{t_-,t_+}\big)^2\Big]. \tag{18}$$

Here $\mathbb{E}_{p_i}$ denotes expectation with respect to the probability density $p_i$ defined as

$$p_i = (1 - c_i)\rho_0 + c_i \boldsymbol{Z}\#\rho_0, \qquad c_i = \max\Big\{c - i\frac{c}{\upsilon L}, 0\Big\}, \tag{19}$$

where $\upsilon \in (0, 1)$ controls the number of steps during which the training is assisted, $c \in (0, 1)$, $L$ is the total number of training steps, and $\boldsymbol{Z} := \boldsymbol{Z}_{t=1}$ is the time-1 map of the ODE $\dot{\boldsymbol{Z}}_t(x) = -\varsigma\nabla U_1\big(\boldsymbol{Z}_t(x)\big)$ with $\boldsymbol{Z}_{t=0}(x) = x$ and $\varsigma > 0$ is a parameter. In essence, using (18) means that, for the first $\upsilon L$ training steps, there is a probability $c_i$ that the data $x \sim \rho_0$ are replaced by $\boldsymbol{Z}(x)$, so that the training method can better explore important regions near local minima of $U_1$. Subsequently, the assistance is turned off so that some subtle adjustment can be made. If some samples from $\rho_1 = e^{-U_1}/\mathcal{Z}_1$ were available beforehand, we could equivalently replace $\boldsymbol{Z}\#\rho_0$ in (19) by the empirical distribution over these samples. We emphasize that the assisted training method is only used to guide the training initially and the NEIS estimator for $\mathcal{Z}_1$ is unbiased.

## 6 Numerical experiments

We consider three benchmark examples to illustrate the effectiveness of NEIS assisted with training. The first two examples involve Gaussian mixtures, for which we use NEIS with $t_- = 0$; the third example is Neal's funnel distribution, for which we use NEIS with $t_- = -1/2$. In all examples, we compare the performance of NEIS with those of annealed importance sampling (AIS) [26]; the number of transition steps in AIS is denoted as $K$ and we refer to this method as AIS-$K$ below; for more details see Appendix I.1. For the comparison, we choose to record the query costs to $U_1$ and $\nabla U_1$ as the measurement of computational complexity, which connects to the framework in the *theory of information complexity* (see e.g., [28]). The runtime could depend on coding, machine condition, etc., whereas query complexity more or less only depends on the computational problem $(U_0, U_1 \text{ and } \boldsymbol{b})$ itself; for most examples of interest, $U_0$ is simple whereas $U_1$ and its derivatives will be expensive to compute; as a remark, $\nabla U_1$ is almost always more expensive to compute than $U_1$.

For simplicity, we always use as base density $\rho_0(x) = (2\pi)^{-d/2}e^{-\frac{1}{2}|x|^2}$. We remark that a better choice of $\rho_0$ (i.e., more adapted to $\rho_1$) can significantly improve the sampling performance; our $\rho_0$ is precisely used to validate the performance of NEIS in situations where $\rho_0$ is *not* well chosen. It would be interesting to study how to adapt the choice of $\rho_0$ for easier training in NEIS, but this is left for future investigations.

When presenting results, we rescale the estimates so that the exact value is $\mathcal{Z}_1 = 1$ for all examples. More implementation details about training are deferred to Appendix I.2. All trainings and estimates of $\mathcal{Z}_1$ are conducted on a laptop with CPU i7-12700H; we use 15 threads at maximum. The runtime of training ranges approximately between $45 \sim 76$ seconds for Gaussian mixture (2D), $9.5\sim12$ minutes for Gaussian mixture (10D); for the Funnel distribution (10D), the runtime is around 25 minutes

for a generic linear ansatz and around 2 minutes for a two-parametric ansatz. Appendix J includes additional figures about training. When computing the gradient of the variance with respect to parameters, we use an integration-based method when $t_- = -1/2$ (for the convenience of numerical implementation) and use an ODE-based method when $t_- = 0$ (for higher accuracy); details about these two approximation methods are given in Appendix I.3 and Appendix I.4 respectively. The codes are accessible on `https://github.com/yucaoyc/NEIS`.

**An asymmetric $2$-mode Gaussian mixture in 2D.** As a first illustration, we consider an asymmetric 2-mode Gaussian mixture

$$e^{-U_1} \propto \frac{1}{5}\mathscr{N}(\lambda \boldsymbol{e}_1, \sigma_1^2 \mathbb{I}_d) + \frac{4}{5}\mathscr{N}(-\lambda \boldsymbol{e}_2, \sigma_1^2 \mathbb{I}_d), \tag{20}$$

where $\boldsymbol{e}_1 = [\begin{smallmatrix}1\\0\end{smallmatrix}]$, $\boldsymbol{e}_2 = [\begin{smallmatrix}0\\1\end{smallmatrix}]$, $\sigma_1 = \sqrt{0.1}$, $\lambda = 5$. With this choice of parameters, the variance of the vanilla estimator based on (2) is approximately $1.85 \times 10^6$. We use NEIS with $t_- = 0$ and set the time step to $\Delta t = 1/50$ for ODE discretization during both training and estimation of $\mathcal{Z}_1$. We train over $L = 50$ SGD steps using the loss (18) by imposing bias in the first $60\%$ of the training period only (i.e., with $\upsilon = 0.6$). The evolution of the variance during the training is shown in Figure 7 in Appendix; the best optimized flow has a variance of about 1, as opposed to $10^6$ for the vanilla estimator. Since $p_i$ in (19) is quite different from $\rho_0$ during the assisted learning period, it may happen that the empirical variance significantly exceeds the variance of the vanilla importance sampling; this does not contradict with Proposition H.3 below. As seen in Figure 7, at the end of the assisted period, the variance is already quite small and in most cases, the variance continues to reduce as $\boldsymbol{b}$ gets further optimized.

After training, we estimate $\mathcal{Z}_1$ using NEIS with the optimized flow and compare its performance with AIS-10 and AIS-100. We first record the query cost for training and then set a total number of queries to $U_1, \nabla U_1$ as budgets. Given the query budget, we estimate $\mathcal{Z}_1$ using each method 10 times, leading to the results given in Table 5 below. When we determine the estimation cost of NEIS, we deduct the query cost of training from the total query budget for fairer comparison. Note that NEIS uses less queries to produce more accurate estimate of $\mathcal{Z}_1$: in particular, the standard deviation of estimating $\mathcal{Z}_1$ by NEIS method is around 1 magnitude smaller than AIS-100. Moreover, the bias from ODE discretization appears to be negligible. Figure 2 shows an optimized flow and also provides a visual comparison of NEIS with AIS under fixed query budget; more comparisons using various ansatzes or architectures can be found in Figures 11 and 12.

**A symmetric $4$-mode Gaussian mixture in 10D.** Next we consider a symmetric 4-mode Gaussian mixture in $d = 10$ dimension with

$$e^{-U_1(x)} \propto \sum_{i=1}^{4} \mathscr{N}(\mu_i, \Sigma), \tag{21}$$

where the vector $\mu_i = \begin{bmatrix} \lambda\cos\left(\frac{i\pi}{2}\right) & \lambda\sin\left(\frac{i\pi}{2}\right) & 0 & 0 & \cdots & 0 \end{bmatrix}$ and $\Sigma = \mathrm{Diag}\begin{bmatrix} \sigma_1^2 & \sigma_1^2 & \sigma_2^2 & \sigma_2^2 & \cdots & \sigma_2^2 \end{bmatrix}$ is a diagonal matrix. The parameters are $d = 10$, $\sigma_1 = \sqrt{0.1}$, $\sigma_2 = \sqrt{0.5}$ and $\lambda = 5$. With this choice of parameters, the variance of the vanilla estimator based on (2) is approximately $2.15 \times 10^6$. We use NEIS with $t_- = 0$ and the time step $\Delta t = 1/60$ is used for ODE discretization during both training and estimation of $\mathcal{Z}_1$. We show the training result in Figure 8. The variance reduces to about 10 after 60 SGD steps for the gradient ansatz (here we only considered this ansatz as it produces more promising empirical results).

Similar to the last example, we compare NEIS using the optimized $\boldsymbol{b}$ with AIS, under fixed query budgets; see Table 5. The best result from NEIS has an estimator with the standard deviation less than $1/3$ of the one from AIS-100. This comparison suggests that AIS-100 needs more than 9 times more resources than NEIS with optimized flow in order to achieve similar precision and the cost spent on training indeed pays off if we require an accurate estimate of $\mathcal{Z}_1$ (meaning less fluctuation for Monte Carlo estimates). Moreover, this table also shows that the bias from ODE discretization is negligible.

Figure 2 shows a particular optimized flow: as can be seen, the mass near the origin flows towards four local minima of $U_1$, as we would intuitively expect. More optimized flows and comparisons can be found in Figure 13 in Appendix.

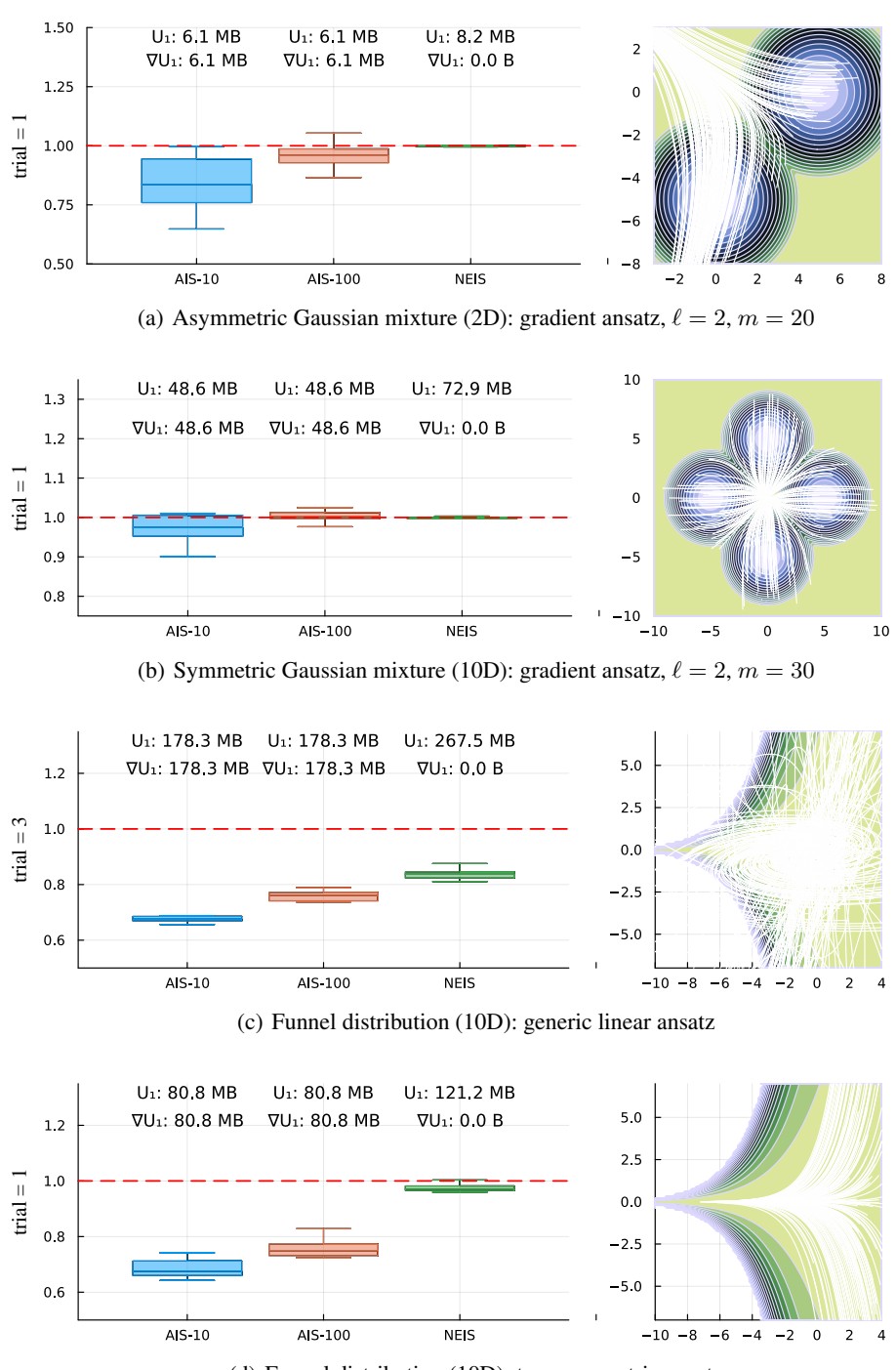

(a) Asymmetric Gaussian mixture (2D): gradient ansatz, $\ell = 2$, $m = 20$

(b) Symmetric Gaussian mixture (10D): gradient ansatz, $\ell = 2$, $m = 30$

(c) Funnel distribution (10D): generic linear ansatz

(d) Funnel distribution (10D): two-parametric ansatz

Figure 2: Selective comparison results for various models. Left panels: estimates of $\mathcal{Z}_1$ by AIS and NEIS with optimized flow under the fixed query budget shown above the panels; we repeat these calculations 10 times and show boxplots of these 10 estimates for each method. The trial number refers to the index for randomly chosen initialization. The query numbers refer to the queries used for each estimate of $\mathcal{Z}_1$ for each method; $1\text{MB} = 10^6$. The dashed red lines show the exact value $\mathcal{Z}_1 = 1$. Right panels: streamlines of optimized flows atop the contours of $U_1$, both projected into the $x_1$-$x_2$ plane for the two 10D examples. Full comparison and figures can be found in Figures 11,12,13, and 14 in Appendix.

**A funnel distribution in 10D.**  We consider the following 10D funnel distribution studied in [2, 27]: for the state $x = [x_1, x_2, \ldots, x_{10}] \in \mathbb{R}^{10}$,

$$x_1 \sim \mathcal{N}(0, 9), \qquad x_i \sim \mathcal{N}(0, e^{x_1}), \qquad 2 \leq i \leq d.$$

For numerical stability, we consider the above funnel distribution restricted to a unit ball centered at the origin with radius 25. Instead of heuristically parameterizing the dynamics via neural-networks, we consider a generic linear ansatz and a two-parametric linear ansatz:

$$\boldsymbol{b}(x) = W_1 x + b_1, \hspace{4cm} W_1 \in \mathbb{R}^{10 \times 10}, b_1 \in \mathbb{R}^{10}, \hspace{1cm} \text{(22a)}$$
$$\boldsymbol{b}(x) = -[\beta, \alpha x_2, \alpha x_3, \ldots, \alpha x_{10}], \hspace{2.5cm} \alpha, \beta \in \mathbb{R}. \hspace{1cm} \text{(22b)}$$

The generic linear ansatz can be regarded as a neural network without inner layers. With (22a), we drawn the entries in the matrix $W_1$ randomly and we set $b_1 = \boldsymbol{0}_{10}$ initially, and we use the assisted training method; with (22b), we set $\alpha = \beta = 2$ initially, and we use the direct training method. In both cases, we choose the finite-NEIS scheme with $t_- = -1/2$. We notice that the asymmetric choice $t_- = 0$ can also leads into more optimal dynamics, but its performance is not as competitive as the symmetric case $t_- = -1/2$. It is very likely that such a difference is due to the structure of funnel distribution: each coordinate $x_i$ ($1 \leq i \leq 10$) has mean 0 and therefore, a symmetric version can probably better weight the contribution from both forward and backward flowlines.

The training results are shown in Figures 9 and 10 in Appendix. In Figure 10, we can observe that both error and variance are overall decreasing during the training and the parameters $\alpha, \beta$ tend to increase with a similar speed. We use the same protocol as in the two previous examples to compare NEIS with AIS. As can be observed in Table 5, the two-parametric ansatz (22b) gives the best estimate; the generic linear ansatz (22a) is not as competitive as the two-parametric ansatz (probably due to over-parameterization), but it still outperforms the AIS-100, under fixed query budget. Figure 2 shows these optimized flows; more results can be found in Figure 14 in Appendix. The apparent gap between estimates and the ground truth in Figure 2 (or see Table 5) comes from insufficient sample size.

# 7    Conclusion and outlook

In this work, we revisited the NEIS strategy proposed in [33] and analyzed its capabilities, both from theoretical and computational standpoints. Regarding the former, we showed that NEIS leads to a zero-variance estimator for a velocity field $\boldsymbol{b} = \nabla V$ with a potential $V$ that satisfies a certain Poisson equation with the difference between the target and the base density as source. Moreover, a zero-variance dynamics can be used to construct a transport map from $\rho_0$ to $\rho_1$. In turn, we highlighted the connection and difference between NEIS and importance sampling strategies based on the normalizing flows (NF).

On the computational side, we showed that the variance of the NEIS estimator can be used as objective function to train the velocity field $\boldsymbol{b}$. This training procedure can be performed in practice by approximating the velocity field by a neural network, and optimizing the parameters in this network using SGD, similar to what is done in the context of neural ODE but with a different objective. Our numerical experiments showed that this strategy is effective and can lower the variance of a vanilla estimator for $\mathcal{Z}_1$ by several orders of magnitude.

While the numerical examples we used in the present paper are somewhat academic, the results suggest that NEIS has potential in more realistic settings. In order to explore other applications, it would be interesting to investigate how to best parametrize $\boldsymbol{b}$ (e.g., less parameters and non-stiff energy landscape with respect to these parameters) and how to best initiate the training procedure. It would also be interesting to ask whether we can improve the performance of NEIS by optimizing certain parameters in the base density $\rho_0$ in concert with $\boldsymbol{b}$. The answers to these questions are probably model specific and are left for future work.

## Acknowledgment

We would like to thank Jonathan Weare and Fang-Hua Lin for helpful discussions, and the anonymous referees for their useful comments and suggestions. The work of EVE is supported by the National Science Foundation under awards DMR-1420073, DMS-2012510, and DMS-2134216, by the Simons Collaboration on Wave Turbulence, Grant No. 617006, and by a Vannevar Bush Faculty Fellowship.

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
