# A The functional space for the infinite-time case

## A.1 Assumptions

For simplicity, we make:

**Assumption A.1.** (i) The domain $\Omega$ is either

- an open and connected subset of $\mathbb{R}^d$ with smooth boundary,
- or a $d$-dimensional torus (without boundary).

(ii) $U_0,\ U_1 \in C^\infty(\Omega, \mathbb{R})$.

(iii) $\mathcal{Z}_0 := \int_\Omega e^{-U_0} = 1$, $\mathcal{Z}_1 < \infty$ and $\mathrm{Var}^{(\mathrm{max})} \in (0, \infty)$, where

$$\mathrm{Var}^{(\mathrm{max})} := \mathbb{E}_{\rho_0}[e^{-2(U_1 - U_0)}] - (\mathcal{Z}_1)^2$$

is the variance for the vanilla importance sampling method.

### Notations

Denote the indicator function for a set $A$ as $\chi_A()$. The open ball around $x$ with radius $r$ is denoted as $B_r(x) := \{ y \in \Omega \mid |y - x| < r \}$. For any subset $D \subset \Omega$, let $\tau_D^+(x)$ and $\tau_D^-(x)$ be the first hitting times to the boundary $\partial D$ in the forward and backward directions, respectively. More specifically,

$$\begin{aligned}
\tau_D^+(x) &:= \inf\{ t \geq 0 : \boldsymbol{X}_t(x) \in \partial D \}, \\
\tau_D^-(x) &:= \sup\{ t \leq 0 : \boldsymbol{X}_t(x) \in \partial D \}.
\end{aligned} \tag{23}$$

For later convenience, let us denote

$$\mathcal{B}(x) := \int_{-\infty}^{\infty} \mathcal{F}_t^{(0)}(x)\, \mathrm{d}t, \tag{24}$$

and thus

$$\mathcal{A}(x) \overset{(9)}{=} \frac{\int_{-\infty}^{\infty} \mathcal{F}_t^{(1)}(x)\, \mathrm{d}t}{\mathcal{B}(x)}. $$

## A.2 Preliminaries

**Lemma A.2.** *If $\boldsymbol{b} \in \mathfrak{B}$ in (3), then for any $t, s \in \mathbb{R}$, $x \in \Omega$ and $k = 0, 1$,*

$$\mathcal{J}_t\big(\boldsymbol{X}_s(x)\big) = \mathcal{J}_{t+s}(x) / \mathcal{J}_s(x), \qquad \mathcal{F}_t^{(k)}\big(\boldsymbol{X}_s(x)\big) = \mathcal{F}_{t+s}^{(k)}(x) / \mathcal{J}_s(x). \tag{25}$$

This lemma can be verified easily by definition and thus the proof is omitted. A simple consequence of (25) is the following result that we also state without proof:

**Lemma A.3.** *For any $s \in \mathbb{R}$ and $x \in \Omega$,*

$$\mathcal{B}\big(\boldsymbol{X}_s(x)\big) = \frac{\mathcal{B}(x)}{\mathcal{J}_s(x)}, \qquad \mathcal{A}\big(\boldsymbol{X}_s(x)\big) = \mathcal{A}(x), \tag{26}$$

*provided that these terms are well-defined.*

## A.3 The functional space

For the infinite-time case, we consider the following family of vector fields denoted as $\mathfrak{B}_\infty$.

**Definition A.4.** $\mathfrak{B}_\infty$ is a set that contains all $\boldsymbol{b} \in \mathfrak{B}$ such that $\Omega \backslash \mho(\boldsymbol{b})$ has Lebesgue measure zero, where $\mho(\boldsymbol{b}) \subset \Omega$ is the collection of points $x$ at which the functions

$$z \mapsto \int_0^\infty \mathcal{F}_t^{(k)}(z)\, \mathrm{d}t, \qquad z \mapsto \int_{-\infty}^0 \mathcal{F}_t^{(k)}(z)\, \mathrm{d}t \tag{27}$$

are continuous on a local neighborhood of $x$ for $k = 0, 1$.

We use the notation $\mho(\boldsymbol{b})$ because in general, such a subset depends on the choice of $\boldsymbol{b}$. The main reason behind the above definition is that we need functions in (27) to behave nicely almost everywhere. The integrability of $t \mapsto \mathcal{F}_t^{(k)}(x)$ depends on the long-term behavior of the flow, which is not easy to characterize in general; thus we simply include the integrability into the assumption. However, we can indeed expect that the above conditions should hold for most interesting examples, for instance:

*Example* A.5. If $U_0(x) = \frac{|x|^2}{2} + \frac{d}{2}\ln(2\pi)$ and $U_1(x) = \frac{|x|^2}{2\sigma^2}$ (so that both $\rho_0$ and $\rho_1$ are Gaussian densities), one optimal flow is $\boldsymbol{b}(x) = x$ (cf. Appendix F below). For this choice, we can direct compute that when $z \neq \boldsymbol{0}_d$, $\boldsymbol{X}_t(z) = e^t z$ and hence

$$\int_0^\infty \mathcal{F}_t^{(1)}(z)\,\mathrm{d}t = \frac{1}{2}\Big(\frac{|z|^2}{2\sigma^2}\Big)^{-d/2}\int_{\frac{|z|^2}{2\sigma^2}}^\infty s^{\frac{d}{2}-1}e^{-s}\,\mathrm{d}s,$$

$$\int_{-\infty}^0 \mathcal{F}_t^{(1)}(z)\,\mathrm{d}t = \frac{1}{2}\Big(\frac{|z|^2}{2\sigma^2}\Big)^{-d/2}\int_0^{\frac{|z|^2}{2\sigma^2}} s^{\frac{d}{2}-1}e^{-s}\,\mathrm{d}s,$$

and similar expressions hold for $\mathcal{F}^{(0)}$ by letting $\sigma = 1$ in above equations. Then clearly, $\mho(\boldsymbol{b}) = \Omega\backslash\{\boldsymbol{0}_d\}$ and such a dynamics belongs to $\mathfrak{B}_\infty$. However, the constant function $\boldsymbol{b} = \boldsymbol{0}_d \notin \mathfrak{B}_\infty$: we can easily verify that e.g., for this dynamics, $\int_0^\infty \mathcal{F}_t^{(1)}(z)\,\mathrm{d}t = \infty$ for any $z$ and hence $\mho(\boldsymbol{0}_d) = \emptyset$; see also Lemma A.7 below.

Here are a few immediate properties of the set $\mho(\boldsymbol{b})$:

**Lemma A.6.** *We have:*

(i) $\mho(\boldsymbol{b})$ *is an open subset of $\Omega$.*

(ii) *The set $\mho(\boldsymbol{b})$ is closed under the evolution of the dynamics $\boldsymbol{b}$, i.e., if $x^\star \in \mho(\boldsymbol{b})$, then $\boldsymbol{X}_t(x^\star) \in \mho(\boldsymbol{b})$ for any $t \in \mathbb{R}$.*

(iii) *If $x^\star \in \overline{\mho(\boldsymbol{b})}$, then $\boldsymbol{X}_t(x^\star) \in \overline{\mho(\boldsymbol{b})}$ for any $t \in \mathbb{R}$, where $\overline{\mho(\boldsymbol{b})}$ is the closure of $\mho(\boldsymbol{b})$ in $\mathbb{R}^d$.*

(iv) $x \mapsto \int_{-\infty}^\infty \mathcal{F}_t^{(k)}(x)\,\mathrm{d}t$ *is continuous on $\mho(\boldsymbol{b})$.*

(v) $(x, t) \mapsto \int_t^\infty \mathcal{F}_s^{(k)}(x)\,\mathrm{d}s$ *is continuous on $\mho(\boldsymbol{b}) \times \mathbb{R}$.*

*Proof.* Part (i) follows easily from the definition of $\mho(\boldsymbol{b})$ and part (iv) also trivially holds. For part (ii), notice that if $y = \boldsymbol{X}_t(x)$, then

$$\int_0^\infty \mathcal{F}_s^{(k)}(y)\,\mathrm{d}s = \int_0^\infty \mathcal{F}_s^{(k)}\big(\boldsymbol{X}_t(x)\big)\,\mathrm{d}s \overset{(25)}{=} \Big(\int_0^\infty \mathcal{F}_{t+s}^{(k)}(x)\,\mathrm{d}s\Big)/\mathcal{J}_t(x)$$

$$= \Big(\int_0^\infty \mathcal{F}_s^{(k)}(x)\,\mathrm{d}s - \int_0^t \mathcal{F}_s^{(k)}(x)\,\mathrm{d}s\Big)/\mathcal{J}_t(x)$$

$$= \Big(\int_0^\infty \mathcal{F}_s^{(k)}\big(\boldsymbol{X}_{-t}(y)\big)\,\mathrm{d}s - \int_0^t \mathcal{F}_s^{(k)}\big(\boldsymbol{X}_{-t}(y)\big)\,\mathrm{d}s\Big)/\mathcal{J}_t\big(\boldsymbol{X}_{-t}(y)\big).$$

Since $(x, t) \mapsto \boldsymbol{X}_t(x)$, $(x, t) \mapsto \mathcal{J}_t(x)$ are continuous by the smoothness assumption of $\boldsymbol{b}$, it is clear that $\mathcal{J}_t\big(\boldsymbol{X}_{-t}(y)\big)$ is continuous on $\Omega$. The continuity of $y \mapsto \int_0^t \mathcal{F}_s^{(k)}\big(\boldsymbol{X}_{-t}(y)\big)\,\mathrm{d}s$ also trivially holds on $\Omega$. Next, the continuity of $y \mapsto \int_0^\infty \mathcal{F}_s^{(k)}\big(\boldsymbol{X}_{-t}(y)\big)\,\mathrm{d}s$ in a local neighborhood of $\boldsymbol{X}_t(x^\star)$ comes from the assumption that $x^\star \in \mho(\boldsymbol{b})$. Thus, $y \mapsto \int_0^\infty \mathcal{F}_s^{(k)}(y)\,\mathrm{d}s$ is continuous in a local neighborhood of $\boldsymbol{X}_t(x^\star)$, if $x^\star \in \mho(\boldsymbol{b})$. The other case for $y \mapsto \int_{-\infty}^0 \mathcal{F}_s^{(k)}(y)\,\mathrm{d}s$ can be similarly verified.

Part (iii) is an immediate consequence of part (ii). Let us prove it by contradiction. Assume that the conclusion in part (iii) does not hold, then there exists $x^\star \in \overline{\mho(\boldsymbol{b})}$ such that $y^\star := \boldsymbol{X}_t(x^\star) \notin \overline{\mho(\boldsymbol{b})}$ for some $t \in \mathbb{R}$. By part (ii), we know this $x^\star \notin \mho(\boldsymbol{b})$ and thus $x^\star \in \partial\mho(\boldsymbol{b})$ is in the boundary. As $\overline{\mho(\boldsymbol{b})}$ is closed and $y^\star \notin \overline{\mho(\boldsymbol{b})}$, there must exist an $\epsilon > 0$ such that for any $y \in \mathbb{R}^d$ with $|y - y^\star| < \epsilon$, we have $y \notin \overline{\mho(\boldsymbol{b})}$. By the smoothness assumption on $\boldsymbol{b}$, we know there exists a $\delta > 0$ such that for

any $x \in B_\delta(x^\star)$, we have $|\boldsymbol{X}_t(x) - y^\star| < \epsilon$ outside of $\overline{\mho(\boldsymbol{b})}$. Since $x^\star$ is in the boundary of $\mho(\boldsymbol{b})$, we must be able to find an $x \in \mho(\boldsymbol{b})$ such that $x \in B_\delta(x^\star)$ and by part (ii), $\boldsymbol{X}_t(x) \in \mho(\boldsymbol{b})$. Thus, we reach a contradiction.

For part (v),

$$\int_t^\infty \mathcal{F}_s^{(k)}(x)\, \mathrm{d}s = \int_0^\infty \mathcal{F}_{t+s}^{(k)}(x)\, \mathrm{d}s \overset{(25)}{=} \mathcal{J}_t(x) \int_0^\infty \mathcal{F}_s^{(k)}\big(\boldsymbol{X}_t(x)\big)\, \mathrm{d}s.$$

Because $(x,t) \mapsto \boldsymbol{X}_t(x)$ is continuous (by the smoothness of $\boldsymbol{b} \in \mathfrak{B}$) and $z \mapsto \int_0^\infty \mathcal{F}_s^{(k)}(z)\, \mathrm{d}s$ is continuous in a local neighborhood of $\boldsymbol{X}_t(x) \in \mho(\boldsymbol{b})$ by part (ii), it is then clear that $\int_0^\infty \mathcal{F}_s^{(k)}\big(\boldsymbol{X}_t(x)\big)\, \mathrm{d}s$ is continuous with respect to $(x,t)$, and hence the conclusion follows easily. $\qquad\square$

A notable family of points that are excluded from $\mho(\boldsymbol{b})$ are points at which $\int_{-\infty}^\infty \mathcal{F}_t^{(k)}(x)\, \mathrm{d}t = \infty$. For instance, these include stationary points of $\boldsymbol{b}$ and periodic orbits, which we state as:

**Lemma A.7.** *Given a vector field $\boldsymbol{b} \in \mathfrak{B}$, we have $x \notin \mho(\boldsymbol{b})$,*

*(i) if $x$ is a stationary point of $\boldsymbol{b}$ (i.e., $\boldsymbol{b}(x) = \boldsymbol{0}_d$), or*

*(ii) if $x$ is on a periodic orbit.*

*Proof.* If $x$ is a stationary point of $\boldsymbol{b}$, the trajectory is $\boldsymbol{X}_t(x) = x$ for all $t \in \mathbb{R}$. Then it is obvious that $\int_{-\infty}^\infty \mathcal{F}_t^{(k)}(x)\, \mathrm{d}t = e^{-U_k(x)} \int_{-\infty}^\infty e^{(\boldsymbol{\nabla}\cdot\boldsymbol{b})(x)t}\, \mathrm{d}t = \infty$, which establishes part (i). For part (ii), let us assume that the period is $\tau$. If $c := \int_0^\tau \boldsymbol{\nabla}\cdot\boldsymbol{b}\big(\boldsymbol{X}_s(x)\big)\, \mathrm{d}s \geq 0$, then

$$\int_0^\infty \mathcal{F}_t^{(k)}(x)\, \mathrm{d}t = \int_0^\infty e^{-U_k\big(\boldsymbol{X}_t(x)\big) + \int_0^t \boldsymbol{\nabla}\cdot\boldsymbol{b}\big(\boldsymbol{X}_s(x)\big)\, \mathrm{d}s}\, \mathrm{d}t$$

$$\geq \int_0^\infty e^{-\max_{0\leq s\leq \tau} U_k\big(\boldsymbol{X}_s(x)\big) + c\lfloor t/\tau\rfloor + \int_{\tau\lfloor t/\tau\rfloor}^{t-\tau\lfloor t/\tau\rfloor} \boldsymbol{\nabla}\cdot\boldsymbol{b}\big(\boldsymbol{X}_s(x)\big)\, \mathrm{d}s}\, \mathrm{d}t$$

$$\geq C \int_0^\infty e^{c(t/\tau - 1)}\, \mathrm{d}t = \infty,$$

where the constant $C = e^{-\max_{0\leq s\leq \tau} U_k\big(\boldsymbol{X}_s(x)\big)} e^{\min_{0\leq s\leq \tau} \int_0^s \boldsymbol{\nabla}\cdot\boldsymbol{b}\big(\boldsymbol{X}_r(x)\big)\, \mathrm{d}r}$. Therefore, $\int_{-\infty}^\infty \mathcal{F}_t^{(k)}(x) = \infty$. If $c := \int_0^\tau \boldsymbol{\nabla}\cdot\boldsymbol{b}\big(\boldsymbol{X}_s(x)\big)\, \mathrm{d}s < 0$, we can similarly show that $\int_{-\infty}^0 \mathcal{F}_t^{(0)}(x)\, \mathrm{d}t = \infty$, and the same conclusion holds. $\qquad\square$

### A.4 Perturbation of the dynamics

Next we investigate the following question: given $\boldsymbol{b} \in \mathfrak{B}_\infty$ and $\delta\boldsymbol{b} \in C_c^\infty(\Omega, \mathbb{R}^d)$, can we guarantee that $\boldsymbol{b} + \epsilon\delta\boldsymbol{b} \in \mathfrak{B}_\infty$ for small enough $\epsilon$? Such a conclusion trivially holds for the finite-time case; however, more underlying structures are needed for the infinite-time case, due to the fact that the long-term behavior of the flow $\boldsymbol{b}$ can sensitively depend on the local perturbation $\delta\boldsymbol{b}$. This question is probably unavoidable in order to understand the mathematical structure of $\mathfrak{B}_\infty$.

**Notation:** We shall use the notation $\boldsymbol{X}_t(\cdot)$ to represent the flow map under $\boldsymbol{b}$, and use $\boldsymbol{X}_t^\epsilon(\cdot)$ to represent the flow map under $\boldsymbol{b}^\epsilon := \boldsymbol{b} + \epsilon\delta\boldsymbol{b}$ in this section below. Moreover, we shall use $\mathcal{F}_\cdot^{(k,\epsilon)}(\cdot)$ to denote the function defined in (6) for the dynamics $\boldsymbol{b}^\epsilon$ and the Jacobian of the flow $\mathcal{J}_{(\cdot)}^\epsilon(\cdot)$ is similarly defined.

**Definition A.8** ($\boldsymbol{b}$-stability). *Given $\boldsymbol{b} \in \mathfrak{B}_\infty$:*

(a) (For an open bounded set). A nonempty open bounded set $D \subset \mho(\boldsymbol{b})$ is said to be *$\boldsymbol{b}$-stable* if:

    (i) There exists a point $x^\star \in D$ and $\zeta \in (0,1)$ such that

$$|\boldsymbol{b}(x^\star)| > 0, \qquad |\boldsymbol{b}(x) - \boldsymbol{b}(x^\star)| < \zeta|\boldsymbol{b}(x^\star)|, \qquad \forall x \in D; \qquad (28)$$

    (ii) For any $x \in D$, the trajectory $t \mapsto \boldsymbol{X}_t(x)$ intersects with the boundary $\partial D$ at exactly two points.

(b) (For a point). A point $x \in \mho(\boldsymbol{b})$ is said to be $\boldsymbol{b}$-stable if there exists a neighborhood $B_\epsilon(x)$ such that the region $B_\epsilon(x)$ is $\boldsymbol{b}$-stable. Otherwise, the point $x$ is said to be $\boldsymbol{b}$-unstable.

The assumption in part (i) ensures that the trajectory $t \mapsto \boldsymbol{X}_t(x)$ will leave this region $D$ within a finite amount of time; see Lemma A.9 below. Part (ii) is used to ensure that the trajectory is not (infinitely) recurrent to $D$; once the trajectory leaves $D$, it will not return to $D$ again.

**Lemma A.9.** *Suppose $\boldsymbol{b} \in \mathfrak{B}$ and $D \subset \Omega$ is open and bounded. If there exists a point $x^\star \in D$ and $\zeta \in (0,1)$ such that*

$$|\boldsymbol{b}(x^\star)| > 0, \qquad |\boldsymbol{b}(x) - \boldsymbol{b}(x^\star)| < \zeta |\boldsymbol{b}(x^\star)|, \qquad \forall x \in D;$$

*then for any $x \in D$, we have*

$$\tau_D^+(x) - \tau_D^-(x) \le \frac{Diameter(D)}{(1-\xi)|\boldsymbol{b}(x^\star)|} < \infty.$$

*Proof.* Consider the quantity $h_t(x) := \langle \boldsymbol{b}(x^\star), \boldsymbol{X}_t(x) - x^\star \rangle$ for $x \in D$. Then when $t \in (\tau_D^-(x), \tau_D^+(x))$,

$$\frac{\mathrm{d}}{\mathrm{d}t} h_t(x) = \langle \boldsymbol{b}(x^\star), \boldsymbol{b}(\boldsymbol{X}_t(x)) - \boldsymbol{b}(x^\star) + \boldsymbol{b}(x^\star) \rangle \ge (1-\zeta)|\boldsymbol{b}(x^\star)|^2 > 0.$$

Therefore,

$$
\begin{aligned}
(1-\xi)|\boldsymbol{b}(x^\star)|^2 \left( \tau_D^+(x) - \tau_D^-(x) \right) &\le \int_{\tau_D^-(x)}^{\tau_D^+(x)} \frac{\mathrm{d}}{\mathrm{d}t} h_t(x) \, \mathrm{d}t \\
&= h_{\tau_D^+(x)}(x) - h_{\tau_D^-(x)}(x) \le |\boldsymbol{b}(x^\star)| \, \mathrm{Diameter}(D).
\end{aligned}
$$

Then the conclusion can be immediately obtained. $\qquad \square$

We now state the main result of this section:

**Proposition A.10.** *Suppose $\boldsymbol{b} \in \mathfrak{B}_\infty$ and $x^\star \in \mho(\boldsymbol{b})$ is $\boldsymbol{b}$-stable. Then there exists an open bounded neighborhood of $x^\star$, denoted as $D \subset \mho(\boldsymbol{b})$, such that for an arbitrary $\delta\boldsymbol{b} \in C_c^\infty(D, \mathbb{R}^d)$, there exists an $\epsilon_0 > 0$ and $\boldsymbol{b} + \epsilon\delta\boldsymbol{b} \in \mathfrak{B}_\infty$ for any $\epsilon \in (0, \epsilon_0)$.*

*Proof.* The main idea is that if the path $t \mapsto \boldsymbol{X}_t(x)$ passes through $D$, then a small perturbation within a bounded time period does not affect the long-term behavior; if the path does not pass through $D$, then $\boldsymbol{b}^\epsilon = \boldsymbol{b}$ along this path and therefore, $\int_0^\infty \mathcal{F}_t^{(k,\epsilon)}(x) \, \mathrm{d}t = \int_0^\infty \mathcal{F}_t^{(k)}(x) \, \mathrm{d}t$; hence, the continuity is also preserved locally around $x$.

**Step (I):** Setup and the choice of $D$.

Since $x^\star \in \mho(\boldsymbol{b})$, we know $|\boldsymbol{b}(x^\star)| > 0$ by Lemma A.7. Moreover, we can find a small $\boldsymbol{b}$-stable ball $B_\theta(x^\star)$ by Definition A.8 with a parameter $\zeta \in (0,1)$ in (28). Then consider the following cross-section within $B_\theta(x^\star)$

$$S := \left\{ y : |y - x^\star| < \theta/2, \qquad \langle y - x^\star, \boldsymbol{b}(x^\star) \rangle = 0 \right\},$$

and define the streamtube $\mathfrak{T}$ passing through $S$ as

$$\mathfrak{T} := \left\{ \boldsymbol{X}_t(y) : y \in S, \ t \in \mathbb{R} \right\}.$$

It is not hard to see that $\mathfrak{T}$ is an open subset of $\mho(\boldsymbol{b})$ by Lemma A.6. Then let us choose $D$ as an arbitrary open ball around $x^\star$ such that

$$D \subset \mathfrak{T} \cap B_{\theta/2}(x^\star).$$

Next let us consider an arbitrary $\delta\boldsymbol{b} \in C_c^\infty(D, \mathbb{R}^d)$ and from now on, we shall fix $\delta\boldsymbol{b}$. It is easy to verify that $\boldsymbol{b} + \epsilon\delta\boldsymbol{b} \in \mathfrak{B}$ for any $\epsilon \in \mathbb{R}$. The non-trivial part is to check that $\Omega \backslash \mho(\boldsymbol{b}^\epsilon)$ has measure zero for sufficiently small $\epsilon$ and hence $\boldsymbol{b}^\epsilon \in \mathfrak{B}_\infty$.

**Step (II):** Choice of $\epsilon_0$.

Let us pick

$$\epsilon_0 = \min\left\{\frac{|\boldsymbol{b}(x^\star)|}{1+|\delta\boldsymbol{b}(x^\star)|},\ \frac{(\zeta^\star-\zeta)|\boldsymbol{b}(x^\star)|}{2\|\delta\boldsymbol{b}\|_{L^\infty(D)}+\zeta^\star|\delta\boldsymbol{b}(x^\star)|}\right\} > 0, \tag{29}$$

for any $\zeta^\star \in (\zeta, 1)$. The main motivation is that we need (28) to hold for $\boldsymbol{b}^\epsilon$ as well. Indeed, if $\epsilon \leq \epsilon_0$,

$$|\boldsymbol{b}^\epsilon(x^\star)| = |\boldsymbol{b}(x^\star) + \epsilon\delta\boldsymbol{b}(x^\star)| \geq |\boldsymbol{b}(x^\star)| - \epsilon|\delta\boldsymbol{b}(x^\star)| \overset{(29)}{\geq} \frac{|\boldsymbol{b}(x^\star)|}{1+|\delta\boldsymbol{b}(x^\star)|} > 0,$$

and for any $x \in B_\theta(x^\star)$,

$$\begin{aligned}
|\boldsymbol{b}^\epsilon(x) - \boldsymbol{b}^\epsilon(x^\star)| &\leq |\boldsymbol{b}(x)-\boldsymbol{b}(x^\star)| + \epsilon|\delta\boldsymbol{b}(x)-\delta\boldsymbol{b}(x^\star)|\\
&\overset{(28)}{\leq} \zeta|\boldsymbol{b}(x^\star)| + \epsilon(2\|\delta\boldsymbol{b}\|_{L^\infty(D)})\\
&\overset{(29)}{<} \zeta^\star|\boldsymbol{b}(x^\star)| - \zeta^\star\epsilon|\delta\boldsymbol{b}(x^\star)|\\
&\leq \zeta^\star|\boldsymbol{b}^\epsilon(x^\star)|.
\end{aligned}$$

As discussed above, this property ensures that any trajectory $t \mapsto \boldsymbol{X}_t^\epsilon(x)$ with $x \in B_\theta(x^\star)$ will pass through the boundary $\partial B_\theta(x^\star)$, and at the same time, since $B_\theta(x^\star)$ is $\boldsymbol{b}$-stable, the condition (ii) in Definition A.8 ensures that such a trajectory only intersects with the boundary $\partial B_\theta(x^\star)$ at exactly 2 points.

Let us denote

$$\tau := \sup_{\epsilon \in [0,\epsilon_0)} \frac{\text{Diameter}(D)}{(1-\xi^\star)|\boldsymbol{b}^\epsilon(x^\star)|} \leq \frac{\text{Diameter}(D)\big(1+|\delta\boldsymbol{b}(x^\star)|\big)}{(1-\xi^\star)|\boldsymbol{b}(x^\star)|} < \infty.$$

By Lemma A.9, we know

$$\boldsymbol{X}_t^\epsilon(x) \notin B_\theta(x^\star), \qquad \forall x \in B_\theta(x^\star),\ \forall\epsilon \in [0,\epsilon_0),\ \forall t \in (-\infty,-\tau]\cup[\tau,\infty). \tag{30}$$

**Step (III):** Prove that $\Omega\backslash\mho(\boldsymbol{b}^\epsilon)$ has measure zero for any $\epsilon \in (0,\epsilon_0)$.

We will prove that for any $\epsilon \in (0,\epsilon_0)$,

$$\mathfrak{T} \subset \mho(\boldsymbol{b}^\epsilon), \qquad \mho(\boldsymbol{b})\backslash\overline{\mathfrak{T}} \subset \mho(\boldsymbol{b}^\epsilon).$$

It could be observed that the boundary $\partial\mathfrak{T}$ has measure zero: the boundary contains flowlines from a hyper-surface with dimension $d-2$, that is, $\partial\mathfrak{T}$ is a set of flowlines passing through $\big\{y:\ |y-x^\star| = \theta/2,\ \langle y-x^\star, \boldsymbol{b}(x^\star)\rangle = 0\big\}$. Hence, provided that the above equation holds, we immediately know that

$$\Omega\backslash\mho(\boldsymbol{b}^\epsilon) \equiv \mho(\boldsymbol{b}^\epsilon)^c \subset \big(\mathfrak{T}\cup(\mho(\boldsymbol{b})\backslash\overline{\mathfrak{T}})\big)^c = \mathfrak{T}^c\cap(\mho(\boldsymbol{b})\cap\overline{\mathfrak{T}}^c)^c = \mathfrak{T}^c\cap(\mho(\boldsymbol{b})^c\cup\overline{\mathfrak{T}})$$

$$= (\mathfrak{T}^c\cap\mho(\boldsymbol{b})^c)\cup(\mathfrak{T}^c\cap\overline{\mathfrak{T}}) \subset \mho(\boldsymbol{b})^c\cup\partial\mathfrak{T} = \big(\Omega\backslash\mho(\boldsymbol{b})\big)\cup\partial\mathfrak{T}$$

has measure zero, where the superscript $c$ means set complement. As a remark, from now on, we shall fix $\epsilon \in (0,\epsilon_0)$.

*Proof of $\mathfrak{T} \subset \mho(\boldsymbol{b}^\epsilon)$.* Let us pick an arbitrary $x \in \mathfrak{T}$ and we shall prove that $z \mapsto \int_0^\infty \mathcal{F}_t^{(k,\epsilon)}(z)\,\mathrm{d}t$ is continuous locally near $x$. Similarly, we can verify that $z \mapsto \int_{-\infty}^0 \mathcal{F}_t^{(k,\epsilon)}(z)\,\mathrm{d}t$ is locally continuous near $x$. Therefore, $x \in \mho(\boldsymbol{b}^\epsilon)$ and thus $\mathfrak{T} \subset \mho(\boldsymbol{b}^\epsilon)$.

Next we return to verify that $z \mapsto \int_0^\infty \mathcal{F}_t^{(k,\epsilon)}(z)\,\mathrm{d}t$ is continuous locally near $x$. We claim that there exists a $y \in S \cup D \in B_{\theta/2}(x^\star)$ and $s \in \mathbb{R}$ such that $\boldsymbol{X}_s^\epsilon(y) = x$. To prove this, we need to discuss two cases:

- Suppose the path $t \mapsto \boldsymbol{X}_t^\epsilon(x)$ never enters $D$. Because $\boldsymbol{b}^\epsilon = \boldsymbol{b}$ on $\Omega\backslash D$, we know $\boldsymbol{X}_t^\epsilon(x) = \boldsymbol{X}_t(x)$ for any $t \in \mathbb{R}$. By the definition of the streamtube $\mathfrak{T}$, there exists a $y \in S$ and $s \in \mathbb{R}$ such that $y = \boldsymbol{X}_{-s}(x) = \boldsymbol{X}_{-s}^\epsilon(x)$, which immediately gives $\boldsymbol{X}_s^\epsilon(y) = x$.

- Suppose the path $t \mapsto \boldsymbol{X}_t^\epsilon(x)$ enters $D$ at some time. Then the above conclusion follows easily.

Because $\boldsymbol{b}^\epsilon$ is smooth, for small enough $\delta$, we can ensure that $B_\delta(x) \subset \mathfrak{T}$ and also $\boldsymbol{X}^\epsilon_{-s}(z) \in B_{\theta/2}(x^\star)$ for any $z \in B_\delta(x)$. By (30),

$$\boldsymbol{X}^\epsilon_{t-s}(z) = \boldsymbol{X}^\epsilon_t\big(\boldsymbol{X}^\epsilon_{-s}(z)\big) \notin B_\theta(x^\star), \qquad \forall z \in B_\delta(x),\ \forall t \geq \tau. \tag{31}$$

We divide the proof of continuity of $z \mapsto \int_0^\infty \mathcal{F}^{(k,\epsilon)}_t(z)\,\mathrm{d}t$ into two cases:

(a) If $\tau \leq s$, then we already know for any point $z \in B_\delta(x)$, we have $\boldsymbol{X}^\epsilon_t(z) \notin B_\theta(x^\star)$ for $t \geq 0$. Recall that $\boldsymbol{b} = \boldsymbol{b}^\epsilon$ outside of $B_\theta(x^\star)$. Hence, $\boldsymbol{X}^\epsilon_t(z) = \boldsymbol{X}_t(z)$ for any $t \geq 0$ and

$$\int_0^\infty \mathcal{F}^{(k,\epsilon)}_t(z)\,\mathrm{d}t = \int_0^\infty \mathcal{F}^{(k)}_t(z)\,\mathrm{d}t,$$

which is continuous on a neighbor of $x$ by $x \in \mho(\boldsymbol{b})$.

(b) If $\tau > s$, then by (31), we know $\boldsymbol{X}^\epsilon_t\big(\boldsymbol{X}^\epsilon_{\tau-s}(z)\big) \notin B_\theta(x^\star)$ for any $z \in B_\delta(x)$ and $t \geq 0$ and in particular, $\boldsymbol{X}^\epsilon_{\tau-s}(z) \notin B_\theta(x^\star)$ for any $z \in B_\delta(x)$. Let us rewrite

$$\int_0^\infty \mathcal{F}^{(k,\epsilon)}_t(z)\,\mathrm{d}t = \int_0^{\tau-s} \mathcal{F}^{(k,\epsilon)}_t(z)\,\mathrm{d}t + \int_{\tau-s}^\infty \mathcal{F}^{(k,\epsilon)}_t(z)\,\mathrm{d}t$$

$$= \int_0^{\tau-s} \mathcal{F}^{(k,\epsilon)}_t(z)\,\mathrm{d}t + \int_0^\infty \mathcal{F}^{(k,\epsilon)}_{t+(\tau-s)}(z)\,\mathrm{d}t$$

$$\overset{(25)}{=} \int_0^{\tau-s} \mathcal{F}^{(k,\epsilon)}_t(z)\,\mathrm{d}t + \mathcal{J}^\epsilon_{\tau-s}(z)\int_0^\infty \mathcal{F}^{(k,\epsilon)}_t\big(\boldsymbol{X}^\epsilon_{\tau-s}(z)\big)\,\mathrm{d}t.$$

Since $\boldsymbol{b}^\epsilon$ is smooth, apparently $z \mapsto \int_0^{\tau-s} \mathcal{F}^{(k,\epsilon)}_t(z)\,\mathrm{d}t$ and $z \mapsto \mathcal{J}^\epsilon_{\tau-s}(z)$ are continuous on $\Omega$. The continuity of $z \mapsto \int_0^\infty \mathcal{F}^{(k,\epsilon)}_t\big(\boldsymbol{X}^\epsilon_{\tau-s}(z)\big)\,\mathrm{d}t$ locally near $x$ can be exactly proved in the same way as Part (a) above for the new point $\boldsymbol{X}^\epsilon_{\tau-s}(x)$.

One small technical result to verify is that $\boldsymbol{X}^\epsilon_{\tau-s}(x) \in \mho(\boldsymbol{b})$ in order to apply the same argument from Part (a). Note that $\boldsymbol{X}^\epsilon_{\tau-s}(x) = \boldsymbol{X}^\epsilon_\tau(y) = \boldsymbol{X}^\epsilon_{\tau-\widetilde{\tau}}\big(\boldsymbol{X}^\epsilon_{\widetilde{\tau}}(y)\big)$ where

$$\widetilde{\tau} := \inf\big\{t \geq 0 \,\big|\, \boldsymbol{X}^\epsilon_t(y) \in \partial B_\theta(x^\star)\big\}.$$

Since $\boldsymbol{X}^\epsilon_{\widetilde{\tau}}(y) \in \partial B_\theta(x^\star)$, we know $\boldsymbol{X}^\epsilon_{\widetilde{\tau}}(y) \in \mho(\boldsymbol{b})$ (e.g., by choosing a small enough $\theta$). Outside of $B_\theta(x^\star)$, we know $\boldsymbol{b}^\epsilon = \boldsymbol{b}$ and thus $\boldsymbol{X}^\epsilon_{\tau-\widetilde{\tau}}\big(\boldsymbol{X}^\epsilon_{\widetilde{\tau}}(y)\big) = \boldsymbol{X}_{\tau-\widetilde{\tau}}\big(\boldsymbol{X}^\epsilon_{\widetilde{\tau}}(y)\big) \in \mho(\boldsymbol{b})$ by Lemma A.6.

*Proof of $\mho(\boldsymbol{b})\backslash\overline{\mathfrak{T}} \subset \mho(\boldsymbol{b}^\epsilon)$.* Let us consider an arbitrary point $x \in \mho(\boldsymbol{b})\backslash\overline{\mathfrak{T}}$. Note that $\boldsymbol{b}^\epsilon = \boldsymbol{b}$ outside of $D \subset \mathfrak{T}$. For a local neighborhood $B_\delta(x)$ outside of $\overline{\mathfrak{T}}$, we also know for any $y \in B_\delta(x)$, $\boldsymbol{X}^\epsilon_t(y) = \boldsymbol{X}_t(y) \notin \overline{\mathfrak{T}}$ for $t \in \mathbb{R}$, by both the definition of $\mathfrak{T}$ and the construction $\boldsymbol{b}^\epsilon = \boldsymbol{b}$ outside of $\mathfrak{T}$. By the same argument as in Part (a) above, it could be readily shown that $x \in \mho(\boldsymbol{b}^\epsilon)$ and thus $\mho(\boldsymbol{b})\backslash\overline{\mathfrak{T}} \subset \mho(\boldsymbol{b}^\epsilon)$. $\qquad\square$

## B  Supplementary material for Section 2

### B.1  Proof of Proposition 2.1

We first prove the finite-time case. As $\boldsymbol{b} \in \mathfrak{B}$, we know that $\int_{t_-}^{t_+} \mathcal{F}^{(0)}_{-s}(x)\,\mathrm{d}s \equiv \int_{t_-}^{t_+} e^{-U_0\big(\boldsymbol{X}_{-s}(x)\big)} \mathcal{J}_{-s}(x)\,\mathrm{d}s < \infty$ by the continuity assumption of $\boldsymbol{b}$ and Assumption A.1. Then

$$\mathcal{Z}_1 = \int_\Omega e^{-U_1(x)} \frac{\int_{t_-}^{t_+} \mathcal{F}^{(0)}_{-t}(x)\,\mathrm{d}t}{\int_{t_-}^{t_+} \mathcal{F}^{(0)}_{-s}(x)\,\mathrm{d}s}\,\mathrm{d}x$$

$$= \int_{t_-}^{t_+} \int_\Omega e^{-U_1(x)} \frac{\mathcal{F}^{(0)}_{-t}(x)}{\int_{t_-}^{t_+} \mathcal{F}^{(0)}_{-s}(x)\,\mathrm{d}s}\,\mathrm{d}x\,\mathrm{d}t.$$

By the change of variables $x = \boldsymbol{X}_t(\widetilde{x})$, we have

$$
\begin{aligned}
\mathcal{Z}_1 &= \int_{t_-}^{t_+} \int_\Omega e^{-U_1\big(\boldsymbol{X}_t(\widetilde{x})\big)} \frac{\mathcal{F}_{-t}^{(0)}(\boldsymbol{X}_t(\widetilde{x}))}{\int_{t_-}^{t_+} \mathcal{F}_{-s}^{(0)}(\boldsymbol{X}_t(\widetilde{x}))\,\mathrm{d}s} \mathcal{J}_t(\widetilde{x})\,\mathrm{d}\widetilde{x}\,\mathrm{d}t \\
&\overset{(25)}{=} \int_{t_-}^{t_+} \int_\Omega e^{-U_1\big(\boldsymbol{X}_t(\widetilde{x})\big)} \frac{\mathcal{F}_0^{(0)}(\widetilde{x})}{\int_{t_-}^{t_+} \mathcal{F}_{t-s}^{(0)}(\widetilde{x})\,\mathrm{d}s} \mathcal{J}_t(\widetilde{x})\,\mathrm{d}\widetilde{x}\,\mathrm{d}t \\
&= \int_{t_-}^{t_+} \int_\Omega e^{-U_1\big(\boldsymbol{X}_t(\widetilde{x})\big)} \mathcal{J}_t(\widetilde{x}) \frac{\rho_0(\widetilde{x})}{\int_{t_-}^{t_+} \mathcal{F}_{t-s}^{(0)}(\widetilde{x})\,\mathrm{d}s}\,\mathrm{d}\widetilde{x}\,\mathrm{d}t \\
&= \mathbb{E}_0\Big[ \int_{t_-}^{t_+} \frac{\mathcal{F}_t^{(1)}(\cdot)}{\int_{t-t_+}^{t-t_-} \mathcal{F}_s^{(0)}(\cdot)\,\mathrm{d}s}\,\mathrm{d}t \Big] \equiv \mathbb{E}_0\big[\mathcal{A}_{t_-,t_+}\big].
\end{aligned}
\tag{32}
$$

Note that as the integrand is non-negative, switching the order of time integration and space integration is justified by Fubini–Tonelli theorem.

The proof of Proposition 2.1 for the infinite-time case is essentially the same as the finite-time case, except the followings:

- We need to replace the domain $\Omega$ in (32) by $\mho(\boldsymbol{b})$ defined in Definition A.4.

- We need the continuity of $x \mapsto \int_{-\infty}^{\infty} \mathcal{F}_t^{(0)}(x)\,\mathrm{d}t$ in order to use Theorem 2 in [20] to get the first line in (32). A generalization with discontinuity should be possible, e.g., by considering piecewise continuous $\boldsymbol{b}$. However, we shall not explore this further in this work. As a remark, the map $x \mapsto \boldsymbol{X}_t(x)$ being bijective (due to the nature of ODE flows) on $\overline{\mho(\boldsymbol{b})}$ was proved in Lemma A.6 (iii), when applying [20, Theorem 2].

## B.2  Proof of Proposition 2.2

Consider

$$
\begin{aligned}
\frac{\mathrm{d}}{\mathrm{d}t}\boldsymbol{X}_t(x) &= \boldsymbol{b}\big(\boldsymbol{X}_t(x)\big), \qquad \boldsymbol{X}_0(x) = x; \\
\frac{\mathrm{d}}{\mathrm{d}t}\boldsymbol{Z}_t(x) &= \alpha\big(\boldsymbol{Z}_t(x)\big)\boldsymbol{b}\big(\boldsymbol{Z}_t(x)\big), \qquad \boldsymbol{Z}_0(x) = x.
\end{aligned}
$$

To prove the first conclusion, we need to verify that the trajectory $\big\{\boldsymbol{X}_t(x)\big\}_{t\in\mathbb{R}} = \big\{\boldsymbol{Z}_t(x)\big\}_{t\in\mathbb{R}}$. From now on, let us fix $x$ and introduce a scalar-valued function $\theta$ by $\theta_t := \int_0^t \alpha\big(\boldsymbol{Z}_s(x)\big)\,\mathrm{d}s$ (i.e., $\frac{\mathrm{d}}{\mathrm{d}t}\theta_t = \alpha\big(\boldsymbol{Z}_t(x)\big)$. By taking the time derivative, it is not hard to verify that $\boldsymbol{Z}_t(x) = \boldsymbol{X}_{\theta_t}(x)$ as both satisfy the same ODE. Of course, $\theta$ also depends on $x$ but we shall not explicitly specify this dependence for simplicity of notations. Thus, the trajectory $t \mapsto \boldsymbol{Z}_t(x)$ is the same as $t \mapsto \boldsymbol{X}_t(x)$ under time rescaling specified by $\theta$.

Next we shall prove the following lemma, which immediately leads into the second result in Proposition 2.2.

**Lemma B.1.** *Suppose $g : \Omega \to \mathbb{R}$ is a non-negative continuous function. For any $x \in \Omega$,*

$$
\int_{-\infty}^{\infty} g\big(\boldsymbol{Z}_t(x)\big) \exp\Big( \int_0^t \nabla \cdot (\alpha\boldsymbol{b})\big(\boldsymbol{Z}_s(x)\big)\,\mathrm{d}s \Big)\,\mathrm{d}t
$$
$$
= \frac{1}{\alpha(x)} \int_{-\infty}^{\infty} g\big(\boldsymbol{X}_t(x)\big) \exp\Big( \int_0^t (\nabla \cdot \boldsymbol{b})\big(\boldsymbol{X}_s(x)\big)\,\mathrm{d}s \Big)\,\mathrm{d}t.
$$

*Proof.* Because $\alpha$ is strictly positive, $\theta. : \mathbb{R} \to \mathbb{R}$ is bijective, and by the inverse function theorem

$$
\frac{\mathrm{d}}{\mathrm{d}t}\theta_t^{-1} = \frac{1}{\alpha\big(\boldsymbol{Z}_{\theta_t^{-1}}(x)\big)} = \frac{1}{\alpha\big(\boldsymbol{X}_t(x)\big)}.
$$

Then by the change of variables $\widetilde{t} = \theta_t$ and $\widetilde{s} = \theta_s$,

$$\int_{-\infty}^{\infty} g\big(\boldsymbol{Z}_t(x)\big) \exp\Big( \int_0^t \nabla \cdot (\alpha \boldsymbol{b})\big(\boldsymbol{Z}_s(x)\big) \, \mathrm{d}s \Big) \, \mathrm{d}t$$

$$= \int_{-\infty}^{\infty} g\big(\boldsymbol{X}_{\theta_t}(x)\big) \exp\Big( \int_0^t \nabla \cdot (\alpha \boldsymbol{b})\big(\boldsymbol{X}_{\theta_s}(x)\big) \, \mathrm{d}s \Big) \, \mathrm{d}t$$

$$= \int_{-\infty}^{\infty} g\big(\boldsymbol{X}_{\widetilde{t}}(x)\big) \frac{1}{\alpha\big(\boldsymbol{X}_{\widetilde{t}}(x)\big)} \exp\Big( \int_0^{\widetilde{t}} \nabla \cdot (\alpha \boldsymbol{b})\big(\boldsymbol{X}_{\widetilde{s}}(x)\big) \frac{1}{\alpha\big(\boldsymbol{X}_{\widetilde{s}}(x)\big)} \, \mathrm{d}\widetilde{s} \Big) \, \mathrm{d}\widetilde{t}.$$

It is then sufficient to show that $\psi_1(t) = \psi_2(t)$ for any $t \in \mathbb{R}$, where

$$\psi_1(t) := \frac{1}{\alpha\big(\boldsymbol{X}_t(x)\big)} \exp\Big( \int_0^t \nabla \cdot (\alpha \boldsymbol{b})\big(\boldsymbol{X}_s(x)\big) \frac{1}{\alpha\big(\boldsymbol{X}_s(x)\big)} \, \mathrm{d}s \Big),$$

$$\psi_2(t) := \frac{1}{\alpha(x)} \exp\Big( \int_0^t (\nabla \cdot \boldsymbol{b})\big(\boldsymbol{X}_s(x)\big) \, \mathrm{d}s \Big).$$

It is easy to observe that $\psi_1(0) = \psi_2(0)$. Let us consider the time derivative of $\psi_1$

$$\frac{\mathrm{d}}{\mathrm{d}t}\psi_1(t) = \psi_1(t)\Big( -\frac{1}{\alpha(\boldsymbol{X}_t(x))} \big\langle \nabla \alpha(\boldsymbol{X}_t(x)), \boldsymbol{b}(\boldsymbol{X}_t(x)) \big\rangle + \nabla \cdot (\alpha \boldsymbol{b})\big(\boldsymbol{X}_t(x)\big) \frac{1}{\alpha\big(\boldsymbol{X}_t(x)\big)} \Big)$$

$$= \psi_1(t)\big(\nabla \cdot \boldsymbol{b}\big)\big(\boldsymbol{X}_t(x)\big).$$

It is clear that $\psi_2(t)$ satisfies the same ODE and thus $\psi_1 = \psi_2$. $\qquad \square$

## B.3 Remarks on the discrete-time analogy of (7)

Let us briefly explain how (7) connects to the method in [42] by time discretization. Suppose $N_- = t_-/h$ and $N_+ = t_+/h$, where $h \ll 1$ is the time step size; for simplicity of notation, let us assume that $N_-$ and $N_+$ are simply integers. By discretizing the time integration for the finite-time NEIS scheme in (7),

$$\mathbb{E}_{x \sim \rho_0}\Big[ \int_{t_-}^{t_+} \frac{\mathcal{F}_t^{(1)}(x)}{\int_{t-T_+}^{t-T_-} \mathcal{F}_s^{(0)}(x) \, \mathrm{d}s} \, \mathrm{d}t \Big] \approx \mathbb{E}_{x \sim \rho_0}\Big[ \sum_{k=N_-}^{N_+} \frac{e^{-U_1\big(\boldsymbol{X}_{kh}(x)\big)} \mathcal{J}_{kh}(x)}{\sum_{j=k-N_+}^{k-N_-} e^{-U_0\big(\boldsymbol{X}_{jh}(x)\big)} \mathcal{J}_{jh}(x)} \Big]$$

$$= \mathbb{E}_{x \sim \rho_0}\Big[ \sum_{k=N_-}^{N_+} \frac{e^{-(U_1-U_0)\big(\boldsymbol{T}^{-k}(x)\big)} \rho_0\big(\boldsymbol{T}^{-k}(x)\big) \mathcal{J}_{\boldsymbol{T}^{-k}}(x)}{\sum_{j=k-N_+}^{k-N_-} \rho_0\big(\boldsymbol{T}^{-j}(x)\big) \mathcal{J}_{\boldsymbol{T}^{-j}}(x)} \Big],$$

where we denote $\boldsymbol{T}(\cdot) := \boldsymbol{X}_{-h}(\cdot)$, and $\mathcal{J}_{\boldsymbol{T}^{-j}}(x) := \big| \det\big( \nabla \boldsymbol{T}^{-j}(x) \big) \big|$ is the Jacobian for the map $\boldsymbol{T}^{-j}$. Then by choosing $N_- = -K$ and $N_+ = 0$, we have

$$\mathbb{E}_{x \sim \rho_0}\Big[ \sum_{k=0}^{K} e^{-(U_1-U_0)\big(\boldsymbol{T}^k(x)\big)} w_k(x) \Big], \qquad w_k(x) = \frac{\rho_0\big(\boldsymbol{T}^k(x)\big) \mathcal{J}_{\boldsymbol{T}^k}(x)}{\sum_{j=-k}^{-k+K} \rho_0\big(\boldsymbol{T}^{-j}(x)\big) \mathcal{J}_{\boldsymbol{T}^{-j}}(x)},$$

which are Eqs. (8) and (10) in arXiv v1 of [42].

## B.4 Remark on the relation between finite-time and infinite-time NEIS

In what follows, we briefly elaborate on the relation between the finite-time and infinite-time NEIS schemes. Suppose we fix $-\infty < t_- < 0 < t_+ < \infty$ and consider a fixed valid flow $\boldsymbol{b}$ for the infinite-time NEIS, i.e.,

$$\mathcal{A}^{\boldsymbol{b}}(x) \equiv \frac{\int_{\mathbb{R}} \exp\big( -U_1(\boldsymbol{X}_t(x)) + \int_0^t \nabla \cdot \boldsymbol{b}(\boldsymbol{X}_r(x)) \, \mathrm{d}r \big) \, \mathrm{d}t}{\int_{\mathbb{R}} \exp\big( -U_0(\boldsymbol{X}_t(x)) + \int_0^t \nabla \cdot \boldsymbol{b}(\boldsymbol{X}_r(x)) \, \mathrm{d}r \big) \, \mathrm{d}t}$$

is well-defined for almost all $x \in \Omega$, where the state $\boldsymbol{X}_t(x)$ solves the ODE $\frac{\mathrm{d}}{\mathrm{d}t}\boldsymbol{X}_t(x) = \boldsymbol{b}(\boldsymbol{X}_t(x))$ for any $x \in \Omega$. The superscript in $\mathcal{A}^{\boldsymbol{b}}$ is used to emphasize that it is the estimator for this particular flow $\boldsymbol{b}$ and similar notations are used below.

Next we consider a family of rescaled flow $\boldsymbol{b}^\alpha$ parameterized by $\alpha > 0$, defined as

$$\boldsymbol{b}^\alpha(x) = \alpha \boldsymbol{b}(x), \qquad \forall x \in \Omega.$$

Let us study its estimator for the finite-time NEIS:

$$\mathcal{A}^{\boldsymbol{b}^\alpha}_{t_-,t_+}(x) \overset{(8)}{=} \int_{t_-}^{t_+} \frac{\exp\left(-U_1(\boldsymbol{Z}_t(x)) + \int_0^t \nabla \cdot \boldsymbol{b}^\alpha(\boldsymbol{Z}_r(x))\,\mathrm{d}r\right)}{\int_{t-t_+}^{t-t_-} \exp\left(-U_0(\boldsymbol{Z}_s(x)) + \int_0^s \nabla \cdot \boldsymbol{b}^\alpha(\boldsymbol{Z}_r(x))\,\mathrm{d}r\right)\mathrm{d}s}\,\mathrm{d}t, \qquad (33)$$

where $\boldsymbol{Z}_t(x)$ solves the ODE $\frac{\mathrm{d}}{\mathrm{d}t}\boldsymbol{Z}_t(x) = \boldsymbol{b}^\alpha(\boldsymbol{Z}_t(x))$ for any $x \in \Omega$. From Appendix B.2, we already know that $\boldsymbol{Z}_t(x) = \boldsymbol{X}_{\alpha t}(x)$ for any $x \in \Omega$ and $t \in \mathbb{R}$. By change of time variables $t = \widetilde{t}/\alpha$, $s = \widetilde{s}/\alpha$ and $r = \widetilde{r}/\alpha$, we have

$$\mathcal{A}^{\boldsymbol{b}^\alpha}_{t_-,t_+}(x) = \int_{\alpha t_-}^{\alpha t_+} \frac{\exp\left(-U_1(\boldsymbol{X}_{\widetilde{t}}(x)) + \int_0^{\widetilde{t}} \nabla \cdot \boldsymbol{b}(\boldsymbol{X}_{\widetilde{r}}(x))\,\mathrm{d}\widetilde{r}\right)}{\int_{\widetilde{t}-\alpha t_+}^{\widetilde{t}-\alpha t_-} \exp\left(-U_0(\boldsymbol{X}_{\widetilde{s}}(x)) + \int_0^{\widetilde{s}} \nabla \cdot \boldsymbol{b}(\boldsymbol{X}_{\widetilde{r}}(x))\,\mathrm{d}\widetilde{r}\right)\mathrm{d}\widetilde{s}}\,\mathrm{d}\widetilde{t}$$

$$= \int_{\mathbb{R}} \frac{\chi_{[\alpha t_-,\alpha t_+]}(\widetilde{t}) \exp\left(-U_1(\boldsymbol{X}_{\widetilde{t}}(x)) + \int_0^{\widetilde{t}} \nabla \cdot \boldsymbol{b}(\boldsymbol{X}_{\widetilde{r}}(x))\,\mathrm{d}\widetilde{r}\right)}{\int_{\widetilde{t}-\alpha t_+}^{\widetilde{t}-\alpha t_-} \exp\left(-U_0(\boldsymbol{X}_{\widetilde{s}}(x)) + \int_0^{\widetilde{s}} \nabla \cdot \boldsymbol{b}(\boldsymbol{X}_{\widetilde{r}}(x))\,\mathrm{d}\widetilde{r}\right)\mathrm{d}\widetilde{s}}\,\mathrm{d}\widetilde{t}.$$

For any $\widetilde{t} \in \mathbb{R}$, as $\alpha \to \infty$, the integrand in the last equation converges pointwise to

$$\frac{\exp\left(-U_1(\boldsymbol{X}_{\widetilde{t}}(x)) + \int_0^{\widetilde{t}} \nabla \cdot \boldsymbol{b}(\boldsymbol{X}_{\widetilde{r}}(x))\,\mathrm{d}\widetilde{r}\right)}{\int_{\mathbb{R}} \exp\left(-U_0(\boldsymbol{X}_{\widetilde{s}}(x)) + \int_0^{\widetilde{s}} \nabla \cdot \boldsymbol{b}(\boldsymbol{X}_{\widetilde{r}}(x))\,\mathrm{d}\widetilde{r}\right)\mathrm{d}\widetilde{s}}.$$

If we heuristically swap the order of taking the limit $\alpha \to \infty$ and the integral with respect to $\widetilde{t}$ (which should generally hold for most examples), we end up with an identity:

$$\lim_{\alpha \to \infty} \mathcal{A}^{\boldsymbol{b}^\alpha}_{t_-,t_+}(x) = \int_{\mathbb{R}} \frac{\exp\left(-U_1(\boldsymbol{X}_{\widetilde{t}}(x)) + \int_0^{\widetilde{t}} \nabla \cdot \boldsymbol{b}(\boldsymbol{X}_{\widetilde{r}}(x))\,\mathrm{d}\widetilde{r}\right)}{\int_{\mathbb{R}} \exp\left(-U_0(\boldsymbol{X}_{\widetilde{s}}(x)) + \int_0^{\widetilde{s}} \nabla \cdot \boldsymbol{b}(\boldsymbol{X}_{\widetilde{r}}(x))\,\mathrm{d}\widetilde{r}\right)\mathrm{d}\widetilde{s}}\,\mathrm{d}\widetilde{t} \equiv \mathcal{A}^{\boldsymbol{b}}(x).$$

The above relation heuristically justifies that due to the space-time rescaling, in the limit of large magnitude of the flow (i.e., $\alpha \to \infty$ above), it does not matter how $t_-, t_+$ are chosen as long as $t_- < 0 < t_+$. In particular, if the flow $\boldsymbol{b}$ is a zero-variance dynamics, i.e., $\mathcal{A}^{\boldsymbol{b}}(x) = \mathcal{Z}_1$ for $x \in \Omega$ almost surely, then in the finite-time NEIS, the flow $\boldsymbol{b}^\alpha = \alpha\boldsymbol{b}$ (with $\alpha \gg 1$) should be approximately a zero-variance dynamics for the finite-time scheme. The finite-time NEIS scheme may not have explicit analytical results about zero-variance dynamics in the same way as the infinite-time NEIS scheme; however, due to the above discussed relation, the finite-time version still possesses the ability to handle and learn an approximately zero-variance dynamics, which is good enough in practice, e.g., during training the optimal flow in Section 5.

## C  The first-order perturbation of the variance for the finite-time scheme

Here we study how the variance (or equivalently, the second moment) of the estimator depends on $\boldsymbol{b}$, since the performance of the finite-time NEIS scheme (7) largely depends on this choice. More specifically, in the following proposition, we study how the second moment changes under a small perturbation of $\boldsymbol{b}$. The expression (34) below will be useful for training optimal dynamics in Section 5 (see also Appendix I for details).

**Proposition C.1.** *Suppose $\boldsymbol{b} \in \mathfrak{B}$ and for any perturbation $\delta\boldsymbol{b} \in C_c^\infty(\Omega, \mathbb{R}^d)$, denote $\boldsymbol{b}^\epsilon := \boldsymbol{b} + \epsilon\delta\boldsymbol{b}$. Then,*

$$\frac{\mathrm{d}}{\mathrm{d}\epsilon}\mathcal{M}_{t_-,t_+}(\boldsymbol{b} + \epsilon\delta\boldsymbol{b})\Big|_{\epsilon=0}$$

$$= 2\mathbb{E}_{x \sim \rho_0}\left[\mathcal{A}_{t_-,t_+}(x)\left(\int_{t_-}^{t_+} \frac{\mathcal{G}_t^{(1)}(x)\int_{t-t_+}^{t-t_-}\mathcal{F}_s^{(0)}(x)\,\mathrm{d}s - \mathcal{F}_t^{(1)}(x)\int_{t-t_+}^{t-t_-}\mathcal{G}_s^{(0)}(x)\,\mathrm{d}s}{\left(\int_{t-t_+}^{t-t_-}\mathcal{F}_s^{(0)}(x)\,\mathrm{d}s\right)^2}\,\mathrm{d}t\right)\right], \qquad (34)$$

*where for $k \in \{0, 1\}$, we define*

$$\mathcal{G}_t^{(k)}(x) := \mathcal{F}_t^{(k)}(x) \left( \begin{array}{l} \left\langle -\nabla U_k\big(\boldsymbol{X}_t(x)\big), \boldsymbol{Y}_t(x) \right\rangle + \int_0^t \left\langle \nabla(\nabla \cdot \boldsymbol{b})\big(\boldsymbol{X}_s(x)\big), \boldsymbol{Y}_s(x) \right\rangle \mathrm{d}s \\ \\ + \int_0^t (\nabla \cdot \delta\boldsymbol{b})\big(\boldsymbol{X}_s(x)\big) \mathrm{d}s \end{array} \right), \quad (35)$$

*and $\boldsymbol{Y}_t(x)$ is the solution of the following ODE:*

$$\frac{\mathrm{d}}{\mathrm{d}t} \boldsymbol{Y}_t(x) = \nabla\boldsymbol{b}\big(\boldsymbol{X}_t(x)\big)\boldsymbol{Y}_t(x) + \delta\boldsymbol{b}\big(\boldsymbol{X}_t(x)\big), \qquad \boldsymbol{Y}_0(x) = 0. \quad (36)$$

The expression of the functional derivative $\frac{\delta\mathcal{M}_{t_-,t_+}(\boldsymbol{b})}{\delta\boldsymbol{b}}$ (not presented in this work) for the finite-time case can be derived in the same way as Lemma D.8 below for the infinite-time case. However, it appears that the expression of $\frac{\delta\mathcal{M}_{t_-,t_+}(\boldsymbol{b})}{\delta\boldsymbol{b}}$ is too complicated to provide useful analytical results.

*Proof.* Let us perturb $\boldsymbol{b}$ by $\epsilon\delta\boldsymbol{b}$ where $\epsilon \ll 1$. Let us consider

$$\frac{\mathrm{d}}{\mathrm{d}t} \boldsymbol{X}_t(x) = \boldsymbol{b}\big(\boldsymbol{X}_t(x)\big),$$

$$\frac{\mathrm{d}}{\mathrm{d}t} \boldsymbol{X}_t^\epsilon(x) = \boldsymbol{b}^\epsilon\big(\boldsymbol{X}_t^\epsilon(x)\big) \equiv (\boldsymbol{b} + \epsilon\delta\boldsymbol{b})\big(\boldsymbol{X}_t^\epsilon(x)\big),$$

with a fixed initial condition $\boldsymbol{X}_0(x) = \boldsymbol{X}_0^\epsilon(x) = x$. For a small $\epsilon$, we can expect that $\boldsymbol{X}_t(x) \approx \boldsymbol{X}_t^\epsilon(x)$ and also we know $\boldsymbol{X}_t^0(x) \equiv \boldsymbol{X}_t(x)$. Define $\mathcal{F}_t^{(k,\epsilon)}(x)$ for the dynamics $\boldsymbol{b}^\epsilon$ in the same way as in (6), namely,

$$\mathcal{F}_t^{(k,\epsilon)}(x) := \exp\left( -U_k\big(\boldsymbol{X}_t^\epsilon(x)\big) + \int_0^t (\nabla \cdot \boldsymbol{b}^\epsilon)\big(\boldsymbol{X}_s^\epsilon(x)\big) \mathrm{d}s \right).$$

By these notations,

$$\mathcal{M}_{t_-,t_+}(\boldsymbol{b}^\epsilon) = \mathbb{E}_{x\sim\rho_0}\left[ \left( \int_{t_-}^{t_+} \frac{\mathcal{F}_t^{(1,\epsilon)}(x)}{\int_{t-t_+}^{t-t_-} \mathcal{F}_s^{(0,\epsilon)}(x) \mathrm{d}s} \mathrm{d}t \right)^2 \right].$$

Then we take the derivative of $\mathcal{M}_{t_-,t_+}(\boldsymbol{b}^\epsilon)$ with respect to $\epsilon$:

$$\frac{\mathrm{d}}{\mathrm{d}\epsilon} \mathcal{M}_{t_-,t_+}(\boldsymbol{b}^\epsilon)$$

$$= 2\mathbb{E}_{x\sim\rho_0} \left[ \begin{array}{l} \left( \int_{t_-}^{t_+} \frac{\mathcal{F}_t^{(1,\epsilon)}(x)}{\int_{t-t_+}^{t-t_-} \mathcal{F}_s^{(0,\epsilon)}(x) \mathrm{d}s} \mathrm{d}t \right) \\ \\ \times \left( \int_{t_-}^{t_+} \frac{\frac{\mathrm{d}}{\mathrm{d}\epsilon}\mathcal{F}_t^{(1,\epsilon)}(x) \int_{t-t_+}^{t-t_-} \mathcal{F}_s^{(0,\epsilon)}(x) \mathrm{d}s - \mathcal{F}_t^{(1,\epsilon)}(x) \int_{t-t_+}^{t-t_-} \frac{\mathrm{d}}{\mathrm{d}\epsilon}\mathcal{F}_s^{(0,\epsilon)}(x) \mathrm{d}s}{\left( \int_{t-t_+}^{t-t_-} \mathcal{F}_s^{(0,\epsilon)}(x) \mathrm{d}s \right)^2} \mathrm{d}t \right) \end{array} \right].$$

Next, we need to compute $\frac{\mathrm{d}}{\mathrm{d}\epsilon}\mathcal{F}_t^{(k,\epsilon)}(x)$. Let us first consider the perturbation to the trajectory. Let $\boldsymbol{Y}_t^\epsilon(x) := \frac{\mathrm{d}}{\mathrm{d}\epsilon}\big(\boldsymbol{X}_t^\epsilon(x)\big)$ and then

$$\frac{\mathrm{d}}{\mathrm{d}t} \boldsymbol{Y}_t^\epsilon(x) = \frac{\mathrm{d}}{\mathrm{d}\epsilon}\left( (\boldsymbol{b} + \epsilon\delta\boldsymbol{b})\big(\boldsymbol{X}_t^\epsilon(x)\big) \right)$$

$$= \nabla\boldsymbol{b}\big(\boldsymbol{X}_t^\epsilon(x)\big)\boldsymbol{Y}_t^\epsilon(x) + \delta\boldsymbol{b}\big(\boldsymbol{X}_t^\epsilon(x)\big) + \epsilon\left(\nabla\delta\boldsymbol{b}\big(\boldsymbol{X}_t^\epsilon(x)\big)\right)\boldsymbol{Y}_t^\epsilon(x),$$

$$\boldsymbol{Y}_0^\epsilon(x) = 0.$$

When $\epsilon = 0$, $\boldsymbol{Y}_t^0(x)$ is the solution to

$$\frac{\mathrm{d}}{\mathrm{d}t} \boldsymbol{Y}_t^0(x) = \nabla\boldsymbol{b}\big(\boldsymbol{X}_t(x)\big)\boldsymbol{Y}_t^0(x) + \delta\boldsymbol{b}\big(\boldsymbol{X}_t(x)\big), \qquad \boldsymbol{Y}_0^0(x) = 0.$$

Now we are ready to explicitly write down $\frac{\mathrm{d}}{\mathrm{d}\epsilon}\mathcal{F}_t^{(k,\epsilon)}(x)$. It is straightforward to derive that

$$\frac{\mathrm{d}}{\mathrm{d}\epsilon}\mathcal{F}_t^{(k,\epsilon)}(x)$$

$$= \mathcal{F}_t^{(k,\epsilon)}(x)\frac{\mathrm{d}}{\mathrm{d}\epsilon}\left(-U_k\big(\boldsymbol{X}_t^{\epsilon}(x)\big) + \int_0^t \nabla \cdot (\boldsymbol{b}+\epsilon\delta\boldsymbol{b})\big(\boldsymbol{X}_s^{\epsilon}(x)\big)\,\mathrm{d}s\right)$$

$$= \mathcal{F}_t^{(k,\epsilon)}(x)\left(\begin{array}{l} \big\langle -\nabla U_k\big(\boldsymbol{X}_t^{\epsilon}(x)\big), \boldsymbol{Y}_t^{\epsilon}(x)\big\rangle \\[2mm] + \int_0^t \big\langle \nabla(\nabla\cdot\boldsymbol{b})\big(\boldsymbol{X}_s^{\epsilon}(x)\big), \boldsymbol{Y}_s^{\epsilon}(x)\big\rangle + (\nabla\cdot\delta\boldsymbol{b})\big(\boldsymbol{X}_s^{\epsilon}(x)\big)\,\mathrm{d}s \\[2mm] + \epsilon\int_0^t \big\langle \nabla(\nabla\cdot\delta\boldsymbol{b})\big(\boldsymbol{X}_s^{\epsilon}(x)\big), \boldsymbol{Y}_s^{\epsilon}(x)\big\rangle\,\mathrm{d}s \end{array}\right).$$

When we let $\epsilon = 0$, we have

$$\mathcal{G}_t^{(k)}(x) := \frac{\mathrm{d}}{\mathrm{d}\epsilon}\mathcal{F}_t^{(k,\epsilon)}(x)\Big|_{\epsilon=0}$$

$$= \mathcal{F}_t^{(k)}(x)\left(\begin{array}{l} \big\langle -\nabla U_k\big(\boldsymbol{X}_t(x)\big), \boldsymbol{Y}_t^0(x)\big\rangle \\[2mm] + \int_0^t \big\langle \nabla(\nabla\cdot\boldsymbol{b})\big(\boldsymbol{X}_s(x)\big), \boldsymbol{Y}_s^0(x)\big\rangle + (\nabla\cdot\delta\boldsymbol{b})\big(\boldsymbol{X}_s(x)\big)\,\mathrm{d}s \end{array}\right).$$

Finally, we arrive at the conclusion by combining previous results and dropping the superscript in $\boldsymbol{Y}_t^0(x)$ for simplicity. $\qquad\square$

## D The first-order perturbation of the variance for the infinite-time scheme

The goal of this section is to derive the functional derivative of $\mathcal{M}(\boldsymbol{b})$ with respect to $\boldsymbol{b}$, denoted as $\frac{\delta\mathcal{M}(\boldsymbol{b})}{\delta\boldsymbol{b}}$, defined as follows: for any $\delta\boldsymbol{b} \in C_c^{\infty}(\Omega, \mathbb{R}^d)$ such that $\boldsymbol{b} + \epsilon\delta\boldsymbol{b} \in \mathfrak{B}_{\infty}$ for small enough $\epsilon$, we have

$$\frac{\mathrm{d}}{\mathrm{d}\epsilon}\mathcal{M}(\boldsymbol{b}+\epsilon\delta\boldsymbol{b})\Big|_{\epsilon=0} = \int_{\Omega}\Big\langle \frac{\delta\mathcal{M}(\boldsymbol{b})}{\delta\boldsymbol{b}}, \delta\boldsymbol{b}\Big\rangle. \tag{37}$$

Since $\mathcal{M}$ and $\mathrm{Var}$ only differ by a constant (which is independent of $\boldsymbol{b}$), it is apparent that $\frac{\delta\mathrm{Var}(\boldsymbol{b})}{\delta\boldsymbol{b}} \equiv \frac{\delta\mathcal{M}(\boldsymbol{b})}{\delta\boldsymbol{b}}$.

**Proposition D.1.** *The functional derivative* $\frac{\delta\mathcal{M}(\boldsymbol{b})}{\delta\boldsymbol{b}} : \Omega \to \mathbb{R}^d$ *has the following form*

$$\frac{\delta\mathrm{Var}(\boldsymbol{b})}{\delta\boldsymbol{b}}(x) \equiv \frac{\delta\mathcal{M}(\boldsymbol{b})}{\delta\boldsymbol{b}}(x)$$

$$= \frac{2\nabla\mathcal{A}(x)}{\mathcal{B}(x)}\left(\int_0^{\infty}\mathcal{F}_t^{(0)}(x)\,\mathrm{d}t\int_{-\infty}^0\mathcal{F}_t^{(1)}(x)\,\mathrm{d}t - \int_{-\infty}^0\mathcal{F}_t^{(0)}(x)\,\mathrm{d}t\int_0^{\infty}\mathcal{F}_t^{(1)}(x)\,\mathrm{d}t\right). \tag{38}$$

*Remark* D.2. The proof of the last formula is slightly formal: for instance, conditions on $\boldsymbol{b}$ to ensure the existence of $\nabla\mathcal{A}$ are not discussed.

Recall the expression of $\mathcal{B}$ from (24). The proof of Proposition D.1 is given in Appendix D.2: it relies on a few explicit formulas, that we state first.

### D.1 Some explicit formulas

We need some explicit formulas for $\boldsymbol{Y}_t(x)$ (36) and $\mathcal{G}_t^{(k)}(x)$ (35). We notice that $\mathcal{G}^{(k)}$ depends on $\boldsymbol{Y}.(\cdot)$ and $\delta\boldsymbol{b}$ linearly, and $\boldsymbol{Y}_t(x)$ also depends on $\delta\boldsymbol{b}$ linearly. Therefore, the first step is to rewrite the expression of $\boldsymbol{Y}_t(x)$ more explicitly in terms of $\delta\boldsymbol{b}$.

**Lemma D.3.** *Suppose the dynamics* $\boldsymbol{b} \in \mathfrak{B}$ *and* $\delta\boldsymbol{b} \in C_c^{\infty}(\Omega, \mathbb{R}^d)$. *Then we have*

$$\boldsymbol{Y}_t(x) = \int_0^t \boldsymbol{C}_{t,s}(x)\,\delta\boldsymbol{b}\big(\boldsymbol{X}_s(x)\big)\,\mathrm{d}s, \qquad \forall x \in \Omega. \tag{39}$$

*The kernel $C_{t,s}(x) \in \mathbb{R}^{d \times d}$ has the following form*

$$C_{t,s}(x) = \begin{cases} \exp_{\mathcal{T}_{\leftarrow}} \left( \int_s^t \nabla b(X_r(x)) \, \mathrm{d}r \right), & \text{if } t \geq s \geq 0, \\ \left( \exp_{\mathcal{T}_{\leftarrow}} \left( \int_t^s \nabla b(X_r(x)) \, \mathrm{d}r \right) \right)^{-1}, & \text{if } t \leq s \leq 0, \end{cases} \tag{40}$$

*where $\exp_{\mathcal{T}_{\leftarrow}}$ is the chronological time-ordered operator exponential.*

*Proof.* By plugging the ansatz (39) into (36), we immediately know that $C_{s,s}(x) = \mathbb{I}_d$ for all $s \in \mathbb{R}$ and

$$\partial_t C_{t,s}(x) = \nabla b(X_t(x)) C_{t,s}(x). \tag{41}$$

This linear ODE has an explicit solution as in (40), by introducing the time-ordered operator exponential. $\qquad \square$

Next, we shall rewrite $\mathcal{G}_t^{(k)}(x)$.

**Lemma D.4.** *We can rewrite $\mathcal{G}_t^{(k)}(x)$ as follows*

$$\mathcal{G}_t^{(k)}(x) = \mathcal{F}_t^{(k)}(x) \left( \int_0^t \left\langle V_{t,s}^{(k)}(x), \delta b(X_s(x)) \right\rangle + (\nabla \cdot \delta b)(X_s(x)) \, \mathrm{d}s \right), \tag{42}$$

*where $V_{t,s}^{(k)}(x)$ is defined as*

$$V_{t,s}^{(k)}(x) := -C_{t,s}(x)^T \nabla U_k(X_t(x)) + \int_s^t C_{r,s}(x)^T \nabla (\nabla \cdot b)(X_r(x)) \, \mathrm{d}r. \tag{43}$$

*Proof.* Recall the expression of $\mathcal{G}_t^{(k)}(x)$ from (35). After plugging (39), we have

$$\mathcal{G}_t^{(k)}(x) \overset{(35)}{=} \mathcal{F}_t^{(k)}(x) \begin{pmatrix} \left\langle -\nabla U_k(X_t(x)), Y_t(x) \right\rangle + \int_0^t \left\langle \nabla(\nabla \cdot b)(X_s(x)), Y_s(x) \right\rangle \mathrm{d}s \\ + \int_0^t (\nabla \cdot \delta b)(X_s(x)) \, \mathrm{d}s \end{pmatrix}$$

$$\overset{(39)}{=} \mathcal{F}_t^{(k)}(x) \begin{pmatrix} -\int_0^t \left\langle C_{t,s}(x)^T \nabla U_k(X_t(x)), \delta b(X_s(x)) \right\rangle \mathrm{d}s + \int_0^t (\nabla \cdot \delta b)(X_s(x)) \, \mathrm{d}s \\ + \int_0^t \int_0^s \left\langle C_{s,r}(x)^T \nabla(\nabla \cdot b)(X_s(x)), \delta b(X_r(x)) \right\rangle \mathrm{d}r \, \mathrm{d}s \end{pmatrix}$$

$$= \mathcal{F}_t^{(k)}(x) \begin{pmatrix} -\int_0^t \left\langle C_{t,s}(x)^T \nabla U_k(X_t(x)), \delta b(X_s(x)) \right\rangle \mathrm{d}s + \int_0^t (\nabla \cdot \delta b)(X_s(x)) \, \mathrm{d}s \\ + \int_0^t \left\langle \int_s^t C_{r,s}(x)^T \nabla(\nabla \cdot b)(X_r(x)) \, \mathrm{d}r, \delta b(X_s(x)) \right\rangle \mathrm{d}s \end{pmatrix}$$

$$= \mathcal{F}_t^{(k)}(x) \left( \int_0^t \left\langle V_{t,s}^{(k)}(x), \delta b(X_s(x)) \right\rangle + (\nabla \cdot \delta b)(X_s(x)) \, \mathrm{d}s \right).$$

$\square$

Then we present a few properties, which will be useful when computing the functional derivative $\frac{\delta \mathcal{M}(b)}{\delta b}$. The following lemma shows how $C_{t,-s}(\cdot)$ and $V_{t,-s}^{(k)}(\cdot)$ change under the dynamical evolution $X_s(\cdot)$.

**Lemma D.5.** *When $t \leq -s \leq 0$ or $t \geq -s \geq 0$,*

$$C_{t,-s}(X_s(x)) = C_{t+s,0}(x), \qquad V_{t,-s}^{(k)}(X_s(x)) = V_{t+s,0}^{(k)}(x). \tag{44}$$

*Proof.* We first consider the term $C_{t,-s}(X_s(x))$. When $t \geq -s \geq 0$,

$$C_{t,-s}(X_s(x)) = \exp_{\mathcal{T}_\leftarrow}\left(\int_{-s}^t \nabla b(X_r(X_s(x)))\,\mathrm{d}r\right)$$

$$= \exp_{\mathcal{T}_\leftarrow}\left(\int_0^{t+s} \nabla b(X_r(x))\,\mathrm{d}r\right) = C_{t+s,0}(x).$$

The case for $t \leq -s \leq 0$ can be similarly verified.

Recall from (43) that

$$V_{t,-s}^{(k)}(X_s(x))$$

$$\overset{(43)}{=} -C_{t,-s}(X_s(x))^T \nabla U_k\left(X_t(X_s(x))\right) + \int_{-s}^t C_{r,-s}(X_s(x))^T \nabla(\nabla \cdot b)\left(X_r(X_s(x))\right)\mathrm{d}r$$

$$= -C_{t+s,0}(x)^T \nabla U_k(X_{t+s}(x)) + \int_0^{t+s} C_{r,0}(x)^T \nabla(\nabla \cdot b)(X_r(x))\,\mathrm{d}r$$

$$\overset{(43)}{=} V_{t+s,0}^{(k)}(x),$$

where to get the third line, we use the above formula (44) about $C_{t,-s}(X_s(x))$. $\qquad\square$

The following lemma connects $C_{t,0}$ and $V_{t,0}^{(k)}$ with gradients.

**Lemma D.6.** *For any $x \in \Omega$ and $t \in \mathbb{R}$, we have*

$$\nabla_x X_t(x) := \left[\frac{\partial(X_t(x))_i}{\partial x_j}\right]_{i,j} = C_{t,0}(x), \tag{45}$$

$$\mathcal{F}_t^{(k)}(x) V_{t,0}^{(k)}(x) = \nabla_x \mathcal{F}_t^{(k)}(x), \qquad \text{for } k \in \{0,1\}. \tag{46}$$

*As a consequence,* $\det(C_{t,0}(x)) = \mathcal{J}_t(x)$.

*Proof.* We fix an index $1 \leq j \leq d$ and consider the dynamics $\frac{\mathrm{d}}{\mathrm{d}t}\widetilde{X}_t^\epsilon(x) = b(\widetilde{X}_t^\epsilon(x))$, $\widetilde{X}_0^\epsilon(x) = x + \epsilon e_j$ where $e_j$ is a vector with the $j^{\text{th}}$ element to be one, and all other elements to be zero. Clearly, $\widetilde{X}_t^0(x) = X_t(x)$.

Let $\widetilde{Y}_t^\epsilon(x) := \frac{\mathrm{d}}{\mathrm{d}\epsilon}\widetilde{X}_t^\epsilon(x)$. Then

$$\frac{\mathrm{d}}{\mathrm{d}t}\widetilde{Y}_t^\epsilon(x) = \frac{\mathrm{d}}{\mathrm{d}\epsilon}b(\widetilde{X}_t^\epsilon(x)) = \nabla b(\widetilde{X}_t^\epsilon(x))\widetilde{Y}_t^\epsilon(x), \qquad \widetilde{Y}_0^\epsilon(x) = e_j.$$

Moreover, when $\epsilon = 0$, we have

$$\frac{\mathrm{d}}{\mathrm{d}t}\widetilde{Y}_t^0(x) = \nabla b(X_t(x))\widetilde{Y}_t^0(x), \qquad \widetilde{Y}_0^0(x) = e_j,$$

whose solution is simply the $j^{\text{th}}$ column of $C_{t,0}(x)$. Besides, the $j^{\text{th}}$ column of $\nabla_x X_t(x)$ is given by

$$\lim_{\epsilon \to 0}\frac{X_t(x + \epsilon e_j) - X_t(x)}{\epsilon} = \lim_{\epsilon \to 0}\frac{\widetilde{X}_t^\epsilon(x) - \widetilde{X}_t^0(x)}{\epsilon} = \frac{\mathrm{d}}{\mathrm{d}\epsilon}\widetilde{X}_t^\epsilon(x)\Big|_{\epsilon=0} = \widetilde{Y}_t^0(x).$$

By combining above results, we easily know that $\nabla_x X_t(x) = C_{t,0}(x)$.

Next, for $k \in \{0,1\}$ and any $x \in \Omega$,

$$\nabla \mathcal{F}_t^{(k)}(x) = \mathcal{F}_t^{(k)}(x)\left(-(\nabla_x X_t(x))^T \nabla U_k(X_t(x)) + \int_0^t (\nabla_x X_s(x))^T \nabla(\nabla \cdot b)(X_s(x))\,\mathrm{d}s\right)$$

$$= \mathcal{F}_t^{(k)}(x)\left(-C_{t,0}(x)^T \nabla U_k(X_t(x)) + \int_0^t C_{s,0}(x)^T \nabla(\nabla \cdot b)(X_s(x))\,\mathrm{d}s\right)$$

$$\overset{(43)}{=} \mathcal{F}_t^{(k)}(x) V_{t,0}^{(k)}(x).$$

$\qquad\square$

## D.2 Proof of Proposition D.1

We list without proof the following result for the infinite-time case, which can be derived in the same way as Proposition C.1.

**Lemma D.7.** *Let $\boldsymbol{b}^\epsilon := \boldsymbol{b} + \epsilon\delta\boldsymbol{b}$. Suppose that for small enough $\epsilon$, we have $\boldsymbol{b}^\epsilon \in \mathfrak{B}_\infty$. Then*

$$\frac{\mathrm{d}}{\mathrm{d}\epsilon}\mathcal{M}(\boldsymbol{b}+\epsilon\delta\boldsymbol{b})\Big|_{\epsilon=0} = 2\mathbb{E}_{x\sim\rho_0}\left[\frac{\mathcal{A}(x)}{\mathcal{B}(x)}\left(\int_{-\infty}^\infty \mathcal{G}_t^{(1)}(x)\,\mathrm{d}t - \mathcal{A}(x)\int_{-\infty}^\infty \mathcal{G}_t^{(0)}(x)\,\mathrm{d}t\right)\right],$$

*where $\mathcal{G}_t^{(k)}(x)$ is defined in (35) for $k = 0, 1$.*

**Lemma D.8.** *The functional derivative of the second moment for the infinite-time case is*

$$\frac{\delta\mathcal{M}(\boldsymbol{b})}{\delta\boldsymbol{b}}(x) = 2\left(\int_{-\infty}^\infty \mathcal{F}_s^{(0)}(x)\boldsymbol{S}_{-s}^\infty\big(\boldsymbol{X}_s(x)\big) - \nabla_x\Big(\mathcal{F}_s^{(0)}(x)\mathcal{G}_{-s}^\infty\big(\boldsymbol{X}_s(x)\big)\Big)\,\mathrm{d}s\right), \tag{47}$$

*where*

$$\boldsymbol{S}_s^\infty(x) := \begin{cases} \dfrac{\mathcal{A}(x)}{\mathcal{B}(x)}\displaystyle\int_s^\infty \mathcal{F}_t^{(1)}(x)\boldsymbol{V}_{t,s}^{(1)}(x)\,\mathrm{d}t - \dfrac{\big(\mathcal{A}(x)\big)^2}{\mathcal{B}(x)}\displaystyle\int_s^\infty \mathcal{F}_t^{(0)}(x)\boldsymbol{V}_{t,s}^{(0)}(x)\,\mathrm{d}t, & \text{if } s > 0; \\[4mm] -\dfrac{\mathcal{A}(x)}{\mathcal{B}(x)}\displaystyle\int_{-\infty}^s \mathcal{F}_t^{(1)}(x)\boldsymbol{V}_{t,s}^{(1)}(x)\,\mathrm{d}t + \dfrac{\big(\mathcal{A}(x)\big)^2}{\mathcal{B}(x)}\displaystyle\int_{-\infty}^s \mathcal{F}_t^{(0)}(x)\boldsymbol{V}_{t,s}^{(0)}(x)\,\mathrm{d}t, & \text{if } s < 0. \end{cases}$$

$$\mathcal{G}_s^\infty(x) := \begin{cases} \dfrac{\mathcal{A}(x)}{\mathcal{B}(x)}\displaystyle\int_s^\infty \mathcal{F}_t^{(1)}(x)\,\mathrm{d}t - \dfrac{\big(\mathcal{A}(x)\big)^2}{\mathcal{B}(x)}\displaystyle\int_s^\infty \mathcal{F}_t^{(0)}(x)\,\mathrm{d}t, & \text{if } s > 0; \\[4mm] -\dfrac{\mathcal{A}(x)}{\mathcal{B}(x)}\displaystyle\int_{-\infty}^s \mathcal{F}_t^{(1)}(x)\,\mathrm{d}t + \dfrac{\big(\mathcal{A}(x)\big)^2}{\mathcal{B}(x)}\displaystyle\int_{-\infty}^s \mathcal{F}_t^{(0)}(x)\,\mathrm{d}t, & \text{if } s < 0. \end{cases}$$

When $s = 0$, $\boldsymbol{S}_0^\infty(\cdot), \mathcal{G}_0^\infty(\cdot)$ are not specified above, because they will not affect the functional derivative $\frac{\delta\mathcal{M}(\boldsymbol{b})}{\delta\boldsymbol{b}}$ by changing values at a single point.

*Proof.* In Lemma D.7, we need to simplify the term $\int_{-\infty}^\infty \mathcal{G}_t^{(k)}(x)\,\mathrm{d}t$. By plugging the formula of $\mathcal{G}^{(k)}$ from (42), we have

$$\int_{-\infty}^\infty \mathcal{G}_t^{(k)}(x)\,\mathrm{d}t$$

$$\overset{(42)}{=} \int_{-\infty}^\infty \mathcal{F}_t^{(k)}(x)\left(\int_0^t \Big\langle \boldsymbol{V}_{t,s}^{(k)}(x), \delta\boldsymbol{b}\big(\boldsymbol{X}_s(x)\big)\Big\rangle + (\nabla\cdot\delta\boldsymbol{b})\big(\boldsymbol{X}_s(x)\big)\,\mathrm{d}s\right)\mathrm{d}t$$

$$= \int_0^\infty \int_s^\infty \mathcal{F}_t^{(k)}(x)\Big\langle \boldsymbol{V}_{t,s}^{(k)}(x), \delta\boldsymbol{b}\big(\boldsymbol{X}_s(x)\big)\Big\rangle + \mathcal{F}_t^{(k)}(x)(\nabla\cdot\delta\boldsymbol{b})\big(\boldsymbol{X}_s(x)\big)\,\mathrm{d}t\,\mathrm{d}s$$

$$- \int_{-\infty}^0 \int_{-\infty}^s \mathcal{F}_t^{(k)}(x)\Big\langle \boldsymbol{V}_{t,s}^{(k)}(x), \delta\boldsymbol{b}\big(\boldsymbol{X}_s(x)\big)\Big\rangle + \mathcal{F}_t^{(k)}(x)(\nabla\cdot\delta\boldsymbol{b})\big(\boldsymbol{X}_s(x)\big)\,\mathrm{d}t\,\mathrm{d}s.$$

By plugging the last equation into Lemma D.7 and with straightforward simplification, we can verify that

$$\frac{\mathrm{d}}{\mathrm{d}\epsilon}\mathcal{M}(\boldsymbol{b}+\epsilon\delta\boldsymbol{b})\Big|_{\epsilon=0}$$

$$= 2\mathbb{E}_{x\sim\rho_0}\left[\int_{-\infty}^\infty \Big\langle \boldsymbol{S}_s^\infty(x), \delta\boldsymbol{b}\big(\boldsymbol{X}_s(x)\big)\Big\rangle\,\mathrm{d}s + \int_{-\infty}^\infty \mathcal{G}_s^\infty(x)(\nabla\cdot\delta\boldsymbol{b})\big(\boldsymbol{X}_s(x)\big)\,\mathrm{d}s\right]$$

$$= 2\int_{-\infty}^\infty \int_\Omega \rho_0(x)\Big(\Big\langle \boldsymbol{S}_s^\infty(x), \delta\boldsymbol{b}\big(\boldsymbol{X}_s(x)\big)\Big\rangle + \mathcal{G}_s^\infty(x)(\nabla\cdot\delta\boldsymbol{b})\big(\boldsymbol{X}_s(x)\big)\Big)\,\mathrm{d}x\,\mathrm{d}s$$

$$\overset{\widetilde{x}=\boldsymbol{X}_s(x)}{=} 2\int_{-\infty}^\infty \int_\Omega \Big(\rho_0\big(\boldsymbol{X}_{-s}(\widetilde{x})\big)\Big\langle \boldsymbol{S}_s^\infty\big(\boldsymbol{X}_{-s}(\widetilde{x})\big), \delta\boldsymbol{b}(\widetilde{x})\Big\rangle \mathcal{J}_{-s}(\widetilde{x})$$

$$+ \rho_0\big(\boldsymbol{X}_{-s}(\widetilde{x})\big)\mathcal{G}_s^\infty\big(\boldsymbol{X}_{-s}(\widetilde{x})\big)(\nabla\cdot\delta\boldsymbol{b})(\widetilde{x})\mathcal{J}_{-s}(\widetilde{x})\Big)\,\mathrm{d}\widetilde{x}\,\mathrm{d}s$$

$$\overset{(6)}{=} 2 \int_{-\infty}^{\infty} \int_{\Omega} \mathcal{F}_{-s}^{(0)}(x) \Big\langle \boldsymbol{S}_s^{\infty}\big(\boldsymbol{X}_{-s}(x)\big), \delta\boldsymbol{b}(x) \Big\rangle - \Big\langle \nabla\Big( \mathcal{F}_{-s}^{(0)}(x) \mathcal{G}_s^{\infty}\big(\boldsymbol{X}_{-s}(x)\big) \Big), \delta\boldsymbol{b}(x) \Big\rangle \, \mathrm{d}x \, \mathrm{d}s.$$

The integration by parts in the last line is valid because $\delta\boldsymbol{b}$ vanishes at the boundary $\partial\Omega$. By comparing the last equation with (37), we can immediately obtain (47) after straightforward simplifications. $\quad\square$

Then we need to simplify $\boldsymbol{S}_{-s}^{\infty}\big(\boldsymbol{X}_s(x)\big)$ and $\mathcal{G}_{-s}^{\infty}\big(\boldsymbol{X}_s(x)\big)$.

**Lemma D.9.** $\boldsymbol{S}_{-s}^{\infty}\big(\boldsymbol{X}_s(x)\big)$ and $\mathcal{G}_{-s}^{\infty}\big(\boldsymbol{X}_s(x)\big)$ have the following form

$$\boldsymbol{S}_{-s}^{\infty}\big(\boldsymbol{X}_s(x)\big) = \begin{cases} -\dfrac{\mathcal{A}(x)}{\mathcal{B}(x)} \displaystyle\int_{-\infty}^{0} \nabla\mathcal{F}_t^{(1)}(x) \, \mathrm{d}t + \dfrac{\big(\mathcal{A}(x)\big)^2}{\mathcal{B}(x)} \int_{-\infty}^{0} \nabla\mathcal{F}_t^{(0)}(x) \, \mathrm{d}t, & \text{if } s > 0, \\[4mm] \dfrac{\mathcal{A}(x)}{\mathcal{B}(x)} \displaystyle\int_{0}^{\infty} \nabla\mathcal{F}_t^{(1)}(x) \, \mathrm{d}t - \dfrac{\big(\mathcal{A}(x)\big)^2}{\mathcal{B}(x)} \int_{0}^{\infty} \nabla\mathcal{F}_t^{(0)}(x) \, \mathrm{d}t, & \text{if } s < 0. \end{cases}$$

$$\mathcal{G}_{-s}^{\infty}\big(\boldsymbol{X}_s(x)\big) = \begin{cases} -\dfrac{\mathcal{A}(x)}{\mathcal{B}(x)} \displaystyle\int_{-\infty}^{0} \mathcal{F}_t^{(1)}(x) \, \mathrm{d}t + \dfrac{\big(\mathcal{A}(x)\big)^2}{\mathcal{B}(x)} \int_{-\infty}^{0} \mathcal{F}_t^{(0)}(x) \, \mathrm{d}t, & \text{if } s > 0, \\[4mm] \dfrac{\mathcal{A}(x)}{\mathcal{B}(x)} \displaystyle\int_{0}^{\infty} \mathcal{F}_t^{(1)}(x) \, \mathrm{d}t - \dfrac{\big(\mathcal{A}(x)\big)^2}{\mathcal{B}(x)} \int_{0}^{\infty} \mathcal{F}_t^{(0)}(x) \, \mathrm{d}t, & \text{if } s < 0. \end{cases}$$

*Proof.* When $s > 0$,

$$\boldsymbol{S}_{-s}^{\infty}\big(\boldsymbol{X}_s(x)\big)$$
$$= -\frac{\mathcal{A}\big(\boldsymbol{X}_s(x)\big)}{\mathcal{B}\big(\boldsymbol{X}_s(x)\big)} \int_{-\infty}^{-s} \mathcal{F}_t^{(1)}\big(\boldsymbol{X}_s(x)\big) \boldsymbol{V}_{t,-s}^{(1)}\big(\boldsymbol{X}_s(x)\big) \, \mathrm{d}t$$

$$+ \frac{\big(\mathcal{A}\big(\boldsymbol{X}_s(x)\big)\big)^2}{\mathcal{B}\big(\boldsymbol{X}_s(x)\big)} \int_{-\infty}^{-s} \mathcal{F}_t^{(0)}\big(\boldsymbol{X}_s(x)\big) \boldsymbol{V}_{t,-s}^{(0)}\big(\boldsymbol{X}_s(x)\big) \, \mathrm{d}t$$

$$\overset{(25),(26),(44)}{=} -\frac{\mathcal{A}(x)}{\mathcal{B}(x)} \int_{-\infty}^{-s} \mathcal{F}_{t+s}^{(1)}(x) \boldsymbol{V}_{t+s,0}^{(1)}(x) \, \mathrm{d}t + \frac{\big(\mathcal{A}(x)\big)^2}{\mathcal{B}(x)} \int_{-\infty}^{-s} \mathcal{F}_{t+s}^{(0)}(x) \boldsymbol{V}_{t+s,0}^{(0)}(x) \, \mathrm{d}t$$

$$= -\frac{\mathcal{A}(x)}{\mathcal{B}(x)} \int_{-\infty}^{0} \mathcal{F}_t^{(1)}(x) \boldsymbol{V}_{t,0}^{(1)}(x) \, \mathrm{d}t + \frac{\big(\mathcal{A}(x)\big)^2}{\mathcal{B}(x)} \int_{-\infty}^{0} \mathcal{F}_t^{(0)}(x) \boldsymbol{V}_{t,0}^{(0)}(x) \, \mathrm{d}t$$

$$\overset{(46)}{=} -\frac{\mathcal{A}(x)}{\mathcal{B}(x)} \int_{-\infty}^{0} \nabla\mathcal{F}_t^{(1)}(x) \, \mathrm{d}t + \frac{\big(\mathcal{A}(x)\big)^2}{\mathcal{B}(x)} \int_{-\infty}^{0} \nabla\mathcal{F}_t^{(0)}(x) \, \mathrm{d}t,$$

which is clearly independent of $s$ from this expression. Similarly, when $s < 0$,

$$\boldsymbol{S}_{-s}^{\infty}\big(\boldsymbol{X}_s(x)\big)$$
$$= \frac{\mathcal{A}\big(\boldsymbol{X}_s(x)\big)}{\mathcal{B}\big(\boldsymbol{X}_s(x)\big)} \int_{-s}^{\infty} \mathcal{F}_t^{(1)}\big(\boldsymbol{X}_s(x)\big) \boldsymbol{V}_{t,-s}^{(1)}\big(\boldsymbol{X}_s(x)\big) \, \mathrm{d}t$$

$$- \frac{\big(\mathcal{A}\big(\boldsymbol{X}_s(x)\big)\big)^2}{\mathcal{B}\big(\boldsymbol{X}_s(x)\big)} \int_{-s}^{\infty} \mathcal{F}_t^{(0)}\big(\boldsymbol{X}_s(x)\big) \boldsymbol{V}_{t,-s}^{(0)}\big(\boldsymbol{X}_s(x)\big) \, \mathrm{d}t$$

$$= \frac{\mathcal{A}(x)}{\mathcal{B}(x)} \int_{-s}^{\infty} \mathcal{F}_{t+s}^{(1)}(x) \boldsymbol{V}_{t+s,0}^{(1)}(x) \, \mathrm{d}t - \frac{\big(\mathcal{A}(x)\big)^2}{\mathcal{B}(x)} \int_{-s}^{\infty} \mathcal{F}_{t+s}^{(0)}(x) \boldsymbol{V}_{t+s,0}^{(0)}(x) \, \mathrm{d}t$$

$$= \frac{\mathcal{A}(x)}{\mathcal{B}(x)} \int_{0}^{\infty} \mathcal{F}_t^{(1)}(x) \boldsymbol{V}_{t,0}^{(1)}(x) \, \mathrm{d}t - \frac{\big(\mathcal{A}(x)\big)^2}{\mathcal{B}(x)} \int_{0}^{\infty} \mathcal{F}_t^{(0)}(x) \boldsymbol{V}_{t,0}^{(0)}(x) \, \mathrm{d}t$$

$$= \frac{\mathcal{A}(x)}{\mathcal{B}(x)} \int_{0}^{\infty} \nabla\mathcal{F}_t^{(1)}(x) \, \mathrm{d}t - \frac{\big(\mathcal{A}(x)\big)^2}{\mathcal{B}(x)} \int_{0}^{\infty} \nabla\mathcal{F}_t^{(0)}(x) \, \mathrm{d}t,$$

which is again independent of $s$. We can similarly simplify $\mathcal{G}_{-s}^{\infty}\big(\boldsymbol{X}_s(x)\big)$. $\quad\square$

By plugging the formula in Lemma D.9 into (47),

$$\int_{-\infty}^{\infty} \mathcal{F}_s^{(0)}(x)\mathcal{G}_{-s}^{\infty}\big(\boldsymbol{X}_s(x)\big)\,\mathrm{d}s$$

$$=\int_0^{\infty}\mathcal{F}_s^{(0)}(x)\,\mathrm{d}s\,\Big(-\frac{\mathcal{A}(x)}{\mathcal{B}(x)}\int_{-\infty}^0\mathcal{F}_t^{(1)}(x)\,\mathrm{d}t+\frac{\big(\mathcal{A}(x)\big)^2}{\mathcal{B}(x)}\int_{-\infty}^0\mathcal{F}_t^{(0)}(x)\,\mathrm{d}t\Big)$$

$$+\int_{-\infty}^0\mathcal{F}_s^{(0)}(x)\,\mathrm{d}s\,\Big(\frac{\mathcal{A}(x)}{\mathcal{B}(x)}\int_0^{\infty}\mathcal{F}_t^{(1)}(x)\,\mathrm{d}t-\frac{\big(\mathcal{A}(x)\big)^2}{\mathcal{B}(x)}\int_0^{\infty}\mathcal{F}_t^{(0)}(x)\,\mathrm{d}t\Big)$$

$$=\frac{\mathcal{A}(x)}{\mathcal{B}(x)}\Big(\int_{-\infty}^0\mathcal{F}_t^{(0)}(x)\,\mathrm{d}t\int_0^{\infty}\mathcal{F}_t^{(1)}(x)\,\mathrm{d}t-\int_0^{\infty}\mathcal{F}_t^{(0)}(x)\,\mathrm{d}t\int_{-\infty}^0\mathcal{F}_t^{(1)}(x)\,\mathrm{d}t\Big).$$

Besides,

$$\int_{-\infty}^{\infty}\mathcal{F}_s^{(0)}(x)\boldsymbol{S}_{-s}^{\infty}\big(\boldsymbol{X}_s(x)\big)\,\mathrm{d}s$$

$$=-\frac{\mathcal{A}(x)}{\mathcal{B}(x)}\int_0^{\infty}\mathcal{F}_s^{(0)}(x)\,\mathrm{d}s\int_{-\infty}^0\nabla\mathcal{F}_t^{(1)}(x)\,\mathrm{d}t+\frac{\big(\mathcal{A}(x)\big)^2}{\mathcal{B}(x)}\int_0^{\infty}\mathcal{F}_s^{(0)}(x)\,\mathrm{d}s\int_{-\infty}^0\nabla\mathcal{F}_t^{(0)}(x)\,\mathrm{d}t$$

$$+\frac{\mathcal{A}(x)}{\mathcal{B}(x)}\int_{-\infty}^0\mathcal{F}_s^{(0)}(x)\,\mathrm{d}s\int_0^{\infty}\nabla\mathcal{F}_t^{(1)}(x)\,\mathrm{d}t-\frac{\big(\mathcal{A}(x)\big)^2}{\mathcal{B}(x)}\int_{-\infty}^0\mathcal{F}_s^{(0)}(x)\,\mathrm{d}s\int_0^{\infty}\nabla\mathcal{F}_t^{(0)}(x)\,\mathrm{d}t.$$

By combining previous results,

$$\frac{1}{2}\times\frac{\delta\mathcal{M}(\boldsymbol{b})}{\delta\boldsymbol{b}}(x)$$

$$\overset{(47)}{=}\int_{-\infty}^{\infty}\mathcal{F}_s^{(0)}(x)\boldsymbol{S}_{-s}^{\infty}\big(\boldsymbol{X}_s(x)\big)\,\mathrm{d}s-\nabla\Big(\int_{-\infty}^{\infty}\mathcal{F}_s^{(0)}(x)\mathcal{G}_{-s}^{\infty}\big(\boldsymbol{X}_s(x)\big)\,\mathrm{d}s\Big)$$

$$=-\frac{\mathcal{A}(x)}{\mathcal{B}(x)}\int_0^{\infty}\mathcal{F}_t^{(0)}(x)\,\mathrm{d}t\int_{-\infty}^0\nabla\mathcal{F}_t^{(1)}(x)\,\mathrm{d}t+\frac{\big(\mathcal{A}(x)\big)^2}{\mathcal{B}(x)}\int_0^{\infty}\mathcal{F}_t^{(0)}(x)\,\mathrm{d}t\int_{-\infty}^0\nabla\mathcal{F}_t^{(0)}(x)\,\mathrm{d}t$$

$$+\frac{\mathcal{A}(x)}{\mathcal{B}(x)}\int_{-\infty}^0\mathcal{F}_t^{(0)}(x)\,\mathrm{d}t\int_0^{\infty}\nabla\mathcal{F}_t^{(1)}(x)\,\mathrm{d}t-\frac{\big(\mathcal{A}(x)\big)^2}{\mathcal{B}(x)}\int_{-\infty}^0\mathcal{F}_t^{(0)}(x)\,\mathrm{d}t\int_0^{\infty}\nabla\mathcal{F}_t^{(0)}(x)\,\mathrm{d}t$$

$$-\nabla\Big(\frac{\mathcal{A}(x)}{\mathcal{B}(x)}\Big)\Big(\int_{-\infty}^0\mathcal{F}_t^{(0)}(x)\,\mathrm{d}t\int_0^{\infty}\mathcal{F}_t^{(1)}(x)\,\mathrm{d}t-\int_0^{\infty}\mathcal{F}_t^{(0)}(x)\,\mathrm{d}t\int_{-\infty}^0\mathcal{F}_t^{(1)}(x)\,\mathrm{d}t\Big)$$

$$-\frac{\mathcal{A}(x)}{\mathcal{B}(x)}\nabla\Big(\int_{-\infty}^0\mathcal{F}_t^{(0)}(x)\,\mathrm{d}t\int_0^{\infty}\mathcal{F}_t^{(1)}(x)\,\mathrm{d}t-\int_0^{\infty}\mathcal{F}_t^{(0)}(x)\,\mathrm{d}t\int_{-\infty}^0\mathcal{F}_t^{(1)}(x)\,\mathrm{d}t\Big)$$

$$=\frac{\big(\mathcal{A}(x)\big)^2}{\mathcal{B}(x)}\Big(\int_0^{\infty}\mathcal{F}_t^{(0)}(x)\,\mathrm{d}t\int_{-\infty}^0\nabla\mathcal{F}_t^{(0)}(x)\,\mathrm{d}t-\int_{-\infty}^0\mathcal{F}_t^{(0)}(x)\,\mathrm{d}t\int_0^{\infty}\nabla\mathcal{F}_t^{(0)}(x)\,\mathrm{d}t\Big)$$

$$-\nabla\Big(\frac{\mathcal{A}(x)}{\mathcal{B}(x)}\Big)\Big(\int_{-\infty}^0\mathcal{F}_t^{(0)}(x)\,\mathrm{d}t\int_0^{\infty}\mathcal{F}_t^{(1)}(x)\,\mathrm{d}t-\int_0^{\infty}\mathcal{F}_t^{(0)}(x)\,\mathrm{d}t\int_{-\infty}^0\mathcal{F}_t^{(1)}(x)\,\mathrm{d}t\Big)$$

$$+\frac{\mathcal{A}(x)}{\mathcal{B}(x)}\Big(\int_0^{\infty}\nabla\mathcal{F}_t^{(0)}(x)\,\mathrm{d}t\int_{-\infty}^0\mathcal{F}_t^{(1)}(x)\,\mathrm{d}t-\int_{-\infty}^0\nabla\mathcal{F}_t^{(0)}(x)\,\mathrm{d}t\int_0^{\infty}\mathcal{F}_t^{(1)}(x)\,\mathrm{d}t\Big).$$

It can be directly verify that

$$\nabla\big(\frac{\mathcal{A}}{\mathcal{B}}\big)(x)=\frac{\mathcal{A}(x)}{\mathcal{B}(x)}\Big(\frac{\int_{-\infty}^{\infty}\nabla\mathcal{F}_t^{(1)}(x)\,\mathrm{d}t}{\int_{-\infty}^{\infty}\mathcal{F}_t^{(1)}(x)\,\mathrm{d}t}-2\frac{\int_{-\infty}^{\infty}\nabla\mathcal{F}_t^{(0)}(x)\,\mathrm{d}t}{\int_{-\infty}^{\infty}\mathcal{F}_t^{(0)}(x)\,\mathrm{d}t}\Big).$$

By plugging this into the expression of the functional derivative and dividing both sides by $\frac{\mathcal{A}}{\mathcal{B}}$,

$$\frac{\mathcal{B}(x)}{2\mathcal{A}(x)}\times\frac{\delta\mathcal{M}(\boldsymbol{b})}{\delta\boldsymbol{b}}(x)$$

$$= \frac{\int_{-\infty}^{\infty} \mathcal{F}_t^{(1)}(x)\,\mathrm{d}t}{\mathcal{B}(x)} \left( \int_0^{\infty} \mathcal{F}_t^{(0)}(x)\,\mathrm{d}t \int_{-\infty}^0 \nabla \mathcal{F}_t^{(0)}(x)\,\mathrm{d}t - \int_{-\infty}^0 \mathcal{F}_t^{(0)}(x)\,\mathrm{d}t \int_0^{\infty} \nabla \mathcal{F}_t^{(0)}(x)\,\mathrm{d}t \right)$$

$$- \frac{\int_{-\infty}^{\infty} \nabla \mathcal{F}_t^{(1)}(x)\,\mathrm{d}t}{\int_{-\infty}^{\infty} \mathcal{F}_t^{(1)}(x)\,\mathrm{d}t} \left( \int_{-\infty}^0 \mathcal{F}_t^{(0)}(x)\,\mathrm{d}t \int_0^{\infty} \mathcal{F}_t^{(1)}(x)\,\mathrm{d}t - \int_0^{\infty} \mathcal{F}_t^{(0)}(x)\,\mathrm{d}t \int_{-\infty}^0 \mathcal{F}_t^{(1)}\,\mathrm{d}t \right)$$

$$+ 2\frac{\int_{-\infty}^{\infty} \nabla \mathcal{F}_t^{(0)}(x)\,\mathrm{d}t}{\int_{-\infty}^{\infty} \mathcal{F}_t^{(0)}(x)\,\mathrm{d}t} \left( \int_{-\infty}^0 \mathcal{F}_t^{(0)}(x)\,\mathrm{d}t \int_0^{\infty} \mathcal{F}_t^{(1)}(x)\,\mathrm{d}t - \int_0^{\infty} \mathcal{F}_t^{(0)}(x)\,\mathrm{d}t \int_{-\infty}^0 \mathcal{F}_t^{(1)}(x)\,\mathrm{d}t \right)$$

$$+ \left( \int_0^{\infty} \nabla \mathcal{F}_t^{(0)}(x)\,\mathrm{d}t \int_{-\infty}^0 \mathcal{F}_t^{(1)}(x)\,\mathrm{d}t - \int_{-\infty}^0 \nabla \mathcal{F}_t^{(0)}(x)\,\mathrm{d}t \int_0^{\infty} \mathcal{F}_t^{(1)}(x)\,\mathrm{d}t \right).$$

We keep the terms involving $\int \nabla \mathcal{F}_t^{(1)}(x)\,\mathrm{d}t$ untouched and we only try to simplify terms involving $\int \nabla \mathcal{F}_t^{(0)}(x)\,\mathrm{d}t$. The coefficient for $\int_0^{\infty} \nabla \mathcal{F}_t^{(0)}(x)\,\mathrm{d}t$ is

$$\frac{1}{\mathcal{B}(x)} \left( \begin{array}{l} -\int_{-\infty}^{\infty} \mathcal{F}_t^{(1)}(x)\,\mathrm{d}t \int_{-\infty}^0 \mathcal{F}_t^{(0)}(x)\,\mathrm{d}t + 2\int_{-\infty}^0 \mathcal{F}_t^{(0)}(x)\,\mathrm{d}t \int_0^{\infty} \mathcal{F}_t^{(1)}(x)\,\mathrm{d}t \\ -2\int_0^{\infty} \mathcal{F}_t^{(0)}(x)\,\mathrm{d}t \int_{-\infty}^0 \mathcal{F}_t^{(1)}(x)\,\mathrm{d}t + \int_{-\infty}^{\infty} \mathcal{F}_t^{(0)}(x)\,\mathrm{d}t \int_{-\infty}^0 \mathcal{F}_t^{(1)}(x)\,\mathrm{d}t \end{array} \right)$$

$$= \frac{1}{\mathcal{B}(x)} \left( \int_{-\infty}^0 \mathcal{F}_t^{(0)}(x)\,\mathrm{d}t \int_0^{\infty} \mathcal{F}_t^{(1)}(x)\,\mathrm{d}t - \int_0^{\infty} \mathcal{F}_t^{(0)}(x)\,\mathrm{d}t \int_{-\infty}^0 \mathcal{F}_t^{(1)}(x)\,\mathrm{d}t \right).$$

Similarly, the coefficient for $\int_{-\infty}^0 \nabla \mathcal{F}_t^{(0)}(x)\,\mathrm{d}t$ is

$$\frac{1}{\mathcal{B}(x)} \left( \begin{array}{l} \int_{-\infty}^{\infty} \mathcal{F}_t^{(1)}(x)\,\mathrm{d}t \int_0^{\infty} \mathcal{F}_t^{(0)}(x)\,\mathrm{d}t + 2\int_{-\infty}^0 \mathcal{F}_t^{(0)}(x)\,\mathrm{d}t \int_0^{\infty} \mathcal{F}_t^{(1)}(x)\,\mathrm{d}t \\ -2\int_0^{\infty} \mathcal{F}_t^{(0)}(x)\,\mathrm{d}t \int_{-\infty}^0 \mathcal{F}_t^{(1)}(x)\,\mathrm{d}t - \int_{-\infty}^{\infty} \mathcal{F}_t^{(0)}(x)\,\mathrm{d}t \int_0^{\infty} \mathcal{F}_t^{(1)}(x)\,\mathrm{d}t \end{array} \right)$$

$$= \frac{1}{\mathcal{B}(x)} \left( \int_{-\infty}^0 \mathcal{F}_t^{(0)}(x)\,\mathrm{d}t \int_0^{\infty} \mathcal{F}_t^{(1)}(x)\,\mathrm{d}t - \int_0^{\infty} \mathcal{F}_t^{(0)}(x)\,\mathrm{d}t \int_{-\infty}^0 \mathcal{F}_t^{(1)}(x)\,\mathrm{d}t \right).$$

Hence,

$$\frac{\mathcal{B}(x)}{2\mathcal{A}(x)} \times \frac{\delta \mathcal{M}(\boldsymbol{b})}{\delta \boldsymbol{b}}(x)$$

$$= \left( \int_{-\infty}^0 \mathcal{F}_t^{(0)}(x)\,\mathrm{d}t \int_0^{\infty} \mathcal{F}_t^{(1)}(x)\,\mathrm{d}t - \int_0^{\infty} \mathcal{F}_t^{(0)}(x)\,\mathrm{d}t \int_{-\infty}^0 \mathcal{F}_t^{(1)}(x)\,\mathrm{d}t \right) \times$$

$$\left( \frac{\int_{-\infty}^{\infty} \nabla \mathcal{F}_t^{(0)}(x)\,\mathrm{d}t}{\int_{-\infty}^{\infty} \mathcal{F}_t^{(0)}(x)\,\mathrm{d}t} - \frac{\int_{-\infty}^{\infty} \nabla \mathcal{F}_t^{(1)}(x)\,\mathrm{d}t}{\int_{-\infty}^{\infty} \mathcal{F}_t^{(1)}(x)\,\mathrm{d}t} \right)$$

$$= \left( \int_{-\infty}^0 \mathcal{F}_t^{(0)}(x)\,\mathrm{d}t \int_0^{\infty} \mathcal{F}_t^{(1)}(x)\,\mathrm{d}t - \int_0^{\infty} \mathcal{F}_t^{(0)}(x)\,\mathrm{d}t \int_{-\infty}^0 \mathcal{F}_t^{(1)}(x)\,\mathrm{d}t \right) \left( -\nabla \ln(\mathcal{A})(x) \right).$$

After straightforward simplification, we obtain (38). $\qquad\square$

## E  Proof of Proposition 3.1 and discussion about its assumptions and implications

In this section, we prove Proposition 3.1, and discuss its assumptions (in particular the Morse function condition) as well as some of its implications (including the settings that go beyond the ones in the proposition). In particular, we solve the Poisson equation (11) when $\rho_0$ is the standard Gaussian density in $\mathbb{R}^d$ and $\rho_1$ is the density of a Gaussian mixture distribution.

### E.1 Proof of Proposition 3.1 when $\mathcal{D} = 1$

We proceed in three steps:

**Step 1:** We shall first establish the limiting behavior of the dynamics.

More specifically, the trajectory $t \mapsto \boldsymbol{X}_t(x)$ will converge to a local maximum of $V$ in the forward direction and converge to a local minimum in the backward direction, except at a set of points with measure zero.

**Lemma E.1.** *For any $x \in \Omega$, we have $\lim_{|t| \to \infty} |\boldsymbol{b}(\boldsymbol{X}_t(x))| = 0$.*

*Proof.* We only need to show one direction $t \to \infty$ and the other case follows similarly. Assume that the conclusion does not hold, then there exists $\epsilon > 0$ and a monotone increasing sequence $\{t_k\}_{k=1}^\infty$ such that $\left|\boldsymbol{b}\big(\boldsymbol{X}_{t_k}(x)\big)\right| \geq \epsilon$ and $\lim_{k \to \infty} t_k = \infty$. Consider

$$
\frac{\mathrm{d}}{\mathrm{d}t}\left|\boldsymbol{b}\big(\boldsymbol{X}_t(x)\big)\right|^2 = \frac{\mathrm{d}}{\mathrm{d}t}\left|\nabla V\big(\boldsymbol{X}_t(x)\big)\right|^2
$$
$$
= 2\Big\langle \nabla V\big(\boldsymbol{X}_t(x)\big), \nabla^2 V\big(\boldsymbol{X}_t(x)\big)\nabla V\big(\boldsymbol{X}_t(x)\big)\Big\rangle \geq -2C\left|\boldsymbol{b}\big(\boldsymbol{X}_t(x)\big)\right|^2,
$$

where $C := \sup_{x \in \Omega}\left\|\nabla^2 V(x)\right\| < \infty$. By Grönwall's inequality,

$$
\left|\boldsymbol{b}\big(\boldsymbol{X}_t(x)\big)\right| \geq \left|\boldsymbol{b}\big(\boldsymbol{X}_s(x)\big)\right| e^{-C(t-s)},
$$

for all $t \geq s \geq 0$. Without loss of generality, we can ensure that $t_k - t_{k-1} \geq 1$. Hence,

$$
V\big(\boldsymbol{X}_t(x)\big) - V\big(\boldsymbol{X}_0(x)\big) = \int_0^t \left|\boldsymbol{b}\big(\boldsymbol{X}_s(x)\big)\right|^2 \mathrm{d}s \geq \sum_{k=1}^\infty \chi_{[0,t]}(t_{k+1}) \int_{t_k}^{t_k+1} \epsilon e^{-C(t-t_k)}\, \mathrm{d}t
$$
$$
= \sup\{k : t_{k+1} \leq t\}\frac{\epsilon\big(1 - e^{-C}\big)}{C},
$$

which will diverge to infinity as $t \to \infty$. This contradicts with the boundedness of $V$ and thus the assumption does not hold. $\square$

**Lemma E.2.** *Suppose $x$ is not in the stable nor unstable manifold of a saddle point of $V$. Then the trajectory $t \mapsto \boldsymbol{X}_t(x)$ must converge to a local maximum of $V$ in the forward direction and a local minimum of $V$ in the backward direction.*

*Proof.* Since the torus $\Omega = [0,1]^d$ is bounded, the trajectory $\{\boldsymbol{X}_t(x)\}_{t \geq 0}$ must be bounded and there exists an increasing sequence $\{t_k\}_{k=1}^\infty$ such that $\big\{\boldsymbol{X}_{t_k}(x)\big\}_{k=1}^\infty$ is convergent by Bolzano-Weierstrass theorem and let us denote the limit as $x^\star$. By Lemma E.1, we know $\boldsymbol{b}(x^\star) = \boldsymbol{0}_d$ and thus $x^\star$ is a critical point. By the assumption, $x^\star$ is not a saddle point nor a local minimum, that is, $x^\star$ must be a local maximum of $V$. By the assumption that $V$ is a Morse function, the critical point $x^\star$ has a non-degenerate Hessian. After the trajectory enters its basin of attraction (containing an open ball around $x^\star$), the trajectory $t \mapsto \boldsymbol{X}_t(x)$ will eventually converge to $x^\star$. The backward direction can be proved in a similar way. $\square$

**Lemma E.3.** *Under the same assumption as in Lemma E.2, we know $\int_0^\infty \mathcal{F}_t^{(k)}(x)\, \mathrm{d}t < \infty$ and $\int_{-\infty}^0 \mathcal{F}_t^{(k)}(x)\, \mathrm{d}t < \infty$. In particular, $\lim_{|t| \to \infty} \mathcal{F}_t^{(k)}(x) = 0$ and $\lim_{|t| \to \infty} \mathcal{J}_t(x) = 0$.*

*Proof.* Without loss of generality, we only consider the forward branch. Since $V$ is assumed to be a Morse function, the Hessian $\nabla^2 V(x^\star) < 0$ is non-degenerate and $0 > \mathrm{tr}\big(\nabla^2 V(x^\star)\big) = \Delta V(x^\star)$. Therefore,

$$
\lim_{t \to \infty} \boldsymbol{\nabla} \cdot \boldsymbol{b}\big(\boldsymbol{X}_t(x)\big) = \boldsymbol{\nabla} \cdot \boldsymbol{b}(x^\star) = \Delta V(x^\star) < 0. \tag{48}
$$

Since $\rho_k = e^{-U_k}/\mathcal{Z}_k$ is bounded on the torus, we know

$$
\int_0^\infty \mathcal{F}_t^{(k)}(x)\, \mathrm{d}t = \int_0^\infty e^{-U_k\big(\boldsymbol{X}_t(x)\big)} \mathcal{J}_t(x)\, \mathrm{d}t \leq C\int_0^\infty \mathcal{J}_t(x)\, \mathrm{d}t = C\int_0^\infty e^{\int_0^t \boldsymbol{\nabla} \cdot \boldsymbol{b}\big(\boldsymbol{X}_s(x)\big)\, \mathrm{d}s}\, \mathrm{d}t,
$$

where $C := \sup_{x \in \Omega} \max\{e^{-U_0(x)}, e^{-U_1(x)}\} < \infty$ herein. From (48), there exists $\beta > 0$ and $\tau > 0$ such that $\boldsymbol{\nabla} \cdot \boldsymbol{b}(\boldsymbol{X}_s(x)) \le -\beta$ for all $s \ge \tau$. Then if $t \ge \tau$,

$$\int_0^t \boldsymbol{\nabla} \cdot \boldsymbol{b}(\boldsymbol{X}_s(x)) \, \mathrm{d}s \le \int_0^\tau \boldsymbol{\nabla} \cdot \boldsymbol{b}(\boldsymbol{X}_s(x)) \, \mathrm{d}s - \beta(t - \tau),$$

and therefore,

$$\int_0^\infty \mathcal{F}_t^{(k)}(x) \, \mathrm{d}t \le C\left( \int_0^\tau e^{\int_0^t \boldsymbol{\nabla} \cdot \boldsymbol{b}(\boldsymbol{X}_s(x))} \, \mathrm{d}t + \int_\tau^\infty e^{\int_0^\tau \boldsymbol{\nabla} \cdot \boldsymbol{b}(\boldsymbol{X}_s(x)) \, \mathrm{d}s} e^{-\beta(t-\tau)} \, \mathrm{d}t \right) < \infty.$$

In particular, when $t \ge \tau$,

$$\mathcal{J}_t(x) \le e^{\int_0^\tau \boldsymbol{\nabla} \cdot \boldsymbol{b}(\boldsymbol{X}_s(x)) \, \mathrm{d}s} e^{-\beta(t-\tau)},$$

which converges to zero exponentially fast as $t \to \infty$. The same conclusion holds for $\mathcal{F}_t^{(k)}(x) \equiv e^{-U_k(\boldsymbol{X}_t(x))} \mathcal{J}_t(x)$ when $t \to \infty$. $\qquad\square$

**Step 2:** We verify that $\boldsymbol{b} = \nabla V$ is a zero-variance dynamics.

We need to show that $\int_{-\infty}^\infty (\rho_1 - \rho_0)(\boldsymbol{X}_t(x)) \mathcal{J}_t(x) \, \mathrm{d}t = 0$ almost everywhere on $\Omega$.

Under the same assumption as Lemma E.2, let us consider

$$\int_{-\infty}^\infty (\rho_1 - \rho_0)(\boldsymbol{X}_t(x)) \mathcal{J}_t(x) \, \mathrm{d}t = \int_{-\infty}^\infty \Delta V(\boldsymbol{X}_t(x)) e^{\int_0^t \Delta V(\boldsymbol{X}_s(x)) \, \mathrm{d}s} \, \mathrm{d}t$$

$$= \int_{-\infty}^\infty \frac{\mathrm{d}}{\mathrm{d}t} \mathcal{J}_t(x) \, \mathrm{d}t$$

$$= \lim_{t \to \infty} \mathcal{J}_t(x) - \lim_{t \to -\infty} \mathcal{J}_t(x) = 0.$$

The last line comes from Lemma E.3. The validity of the above equation almost everywhere on $\Omega$ will be explained in Step 3.

**Step 3:** We prove that $\boldsymbol{b} = \nabla V \in \mathfrak{B}_\infty$ in the sense of Definition A.4, i.e., such a gradient ascent dynamics is a valid one for the infinite-time NEIS scheme.

If we can find two open subsets $D_1, D_2$ such that $x \mapsto \int_0^\infty \mathcal{F}_t^{(k)}(x) \, \mathrm{d}t$ is continuous on $D_1$, $x \mapsto \int_{-\infty}^0 \mathcal{F}_t^{(k)}(x) \, \mathrm{d}t$ is continuous on $D_2$, and both $\Omega \backslash D_1$ and $\Omega \backslash D_2$ have Lebesgue measure zero, then clearly $\mho(\boldsymbol{b}) \supset D_1 \cap D_2$ and $\boldsymbol{b} \in \mathfrak{B}_\infty$.

Due to the symmetric role of forward and backward branches of trajectories, it is then sufficient to prove the following lemma.

**Lemma E.4.** *There exists an open subset $D \subset \Omega$ such that $x \mapsto \int_0^\infty \mathcal{F}_t^{(k)}(x)$ is continuous and $\Omega \backslash D$ has measure zero.*

*Proof.* Let us denote the local maxima of $V$ as $\mathscr{X}_1, \mathscr{X}_2, \cdots, \mathscr{X}_r$. The index $r < \infty$ because $V$ is a Morse function and $\Omega$ is compact. Since $\nabla^2 V(\mathscr{X}_i) < 0$ for $1 \le i \le r$, there exists a local neighborhood $B_{\delta_i}(\mathscr{X}_i)$ such that $\lim_{t \to \infty} \boldsymbol{X}_t(z) = \mathscr{X}_i$ if $z \in B_{\delta_i}(\mathscr{X}_i)$. Hence, it is not hard to characterize the basin of attraction of $\mathscr{X}_i$

$$O_i = \left\{ \boldsymbol{X}_t(x) : x \in B_{\delta_i}(\mathscr{X}_i), \, t \le 0 \right\}$$

which is open. Then define an open subset $D := \cup_{i=1}^r O_i$. By Lemma E.2, we know $\Omega \backslash D$ has Lebesgue measure zero, since there is only a finite number of critical points and the stable/unstable manifold of saddle points has measure zero. Next, we still need to verify $z \mapsto \int_0^\infty \mathcal{F}_t^{(k)}(z) \, \mathrm{d}t$ is continuous at an arbitrary point $x \in D$. Since $D = \cup_{i=1}^r O_i$ and $O_i$ are open and disjoint, it is sufficient to verify this conclusion for $x \in O_i$ for an arbitrary index $i$.

By the smoothness of $V$, there exists a local neighborhood $O_\delta := \{z \in O_i : V(\mathscr{X}_i) - \delta < V(z) \le V(\mathscr{X}_i)\}$ such that $\frac{(\boldsymbol{\nabla} \cdot \boldsymbol{b})(z)}{(\boldsymbol{\nabla} \cdot \boldsymbol{b})(\mathscr{X}_i)} \in (\frac{1}{2}, \frac{3}{2})$ for every $z \in O_\delta$. Let $M = \sup \{ \max\{e^{-U_1(x)}, e^{-U_0(x)}\} :$

$x \in \Omega$}. We know $M < \infty$ because $\Omega$ is compact and $U_0, U_1$ are smooth. Define $\tau := \inf\{t \geq 1 : \boldsymbol{X}_t(x) \in O_\delta\}$. Due to the smoothness of $\boldsymbol{b}$, for any integer $j \geq 2$, we can choose a small neighborhood $B_{\delta_j}(x)$ such that $\boldsymbol{X}_t(z) \in O_\delta$ for all $t \geq j\tau$ and for all $z \in B_{\delta_j}(x)$ (note that $O_\delta$ is automatically a trapping region of $\boldsymbol{b}$ by construction). Then when $z \in B_{\delta_j}(x)$,

$$
\begin{aligned}
\int_{j\tau}^{\infty} \mathcal{F}_t^{(k)}(z)\,\mathrm{d}t &\leq \int_{j\tau}^{\infty} M \mathcal{J}_t(z)\,\mathrm{d}t \\
&\leq \int_{j\tau}^{\infty} M \mathcal{J}_{j\tau}(z) e^{(\boldsymbol{\nabla}\cdot\boldsymbol{b})(\mathscr{X}_i)\frac{1}{2}(t-j\tau)}\,\mathrm{d}t \\
&\leq 2M \mathcal{J}_{j\tau}(z) \frac{1}{-(\boldsymbol{\nabla}\cdot\boldsymbol{b})(\mathscr{X}_i)} \\
&= \frac{2M}{-(\boldsymbol{\nabla}\cdot\boldsymbol{b})(\mathscr{X}_i)} \Big( \mathcal{J}_{j\tau}(z) - \mathcal{J}_{j\tau}(x) + \mathcal{J}_{j\tau}(x) \Big).
\end{aligned}
$$

For an arbitrary $\epsilon > 0$, by Lemma E.3, we can pick $j$ large enough such that $\mathcal{J}_{j\tau}(x) < \frac{-(\boldsymbol{\nabla}\cdot\boldsymbol{b})(\mathscr{X}_i)}{2M}\frac{\epsilon}{6}$. Next we can accordingly pick $\delta_j$ small enough such that $\mathcal{J}_{j\tau}(z) - \mathcal{J}_{j\tau}(x) < \frac{-(\boldsymbol{\nabla}\cdot\boldsymbol{b})(\mathscr{X}_i)}{2M}\frac{\epsilon}{6}$ for all $z \in B_{\delta_j}(x)$. In this way, we can ensure that

$$
\int_{j\tau}^{\infty} \mathcal{F}_t^{(k)}(z) \leq \frac{\epsilon}{6} + \frac{\epsilon}{6} = \frac{\epsilon}{3}, \qquad \forall z \in B_{\delta_j}(x). \tag{49}
$$

Furthermore, due to the smoothness of $\boldsymbol{b}$, we can choose $\delta_j$ (possibly even smaller) so that

$$
\Big| \int_0^{j\tau} \mathcal{F}_t^{(k)}(z)\,\mathrm{d}t - \int_0^{j\tau} \mathcal{F}_t^{(k)}(x)\,\mathrm{d}t \Big| \leq \frac{\epsilon}{3}, \qquad \forall z \in B_{\delta_j}(x). \tag{50}
$$

The continuity of $z \mapsto \int_0^{j\tau} \mathcal{F}_t^{(k)}(z)\,\mathrm{d}t$ can be easily established due to the differentiability of $z \mapsto \mathcal{F}_t^{(k)}(z)$. By combining previous results, for each $\epsilon > 0$, we can find a $\delta_j$ such that for any $z \in B_{\delta_j}(x)$,

$$
\begin{aligned}
&\Big| \int_0^{\infty} \mathcal{F}_t^{(k)}(z)\,\mathrm{d}t - \int_0^{\infty} \mathcal{F}_t^{(k)}(x)\,\mathrm{d}t \Big| \\
&\leq \Big| \int_0^{j\tau} \mathcal{F}_t^{(k)}(z)\,\mathrm{d}t - \int_0^{j\tau} \mathcal{F}_t^{(k)}(x)\,\mathrm{d}t \Big| + \int_{j\tau}^{\infty} \mathcal{F}_t^{(k)}(z)\,\mathrm{d}t + \int_{j\tau}^{\infty} \mathcal{F}_t^{(k)}(x)\,\mathrm{d}t \\
&\overset{(49),(50)}{\leq} \frac{\epsilon}{3} + \frac{\epsilon}{3} + \frac{\epsilon}{3} = \epsilon.
\end{aligned}
$$

This proves the continuity of $z \mapsto \int_0^{\infty} \mathcal{F}_t^{(k)}(z)\,\mathrm{d}t$ at the point $x$. $\qquad\square$

### E.2 A remark about the general case

By Proposition 2.2, to prove that $\nabla V$ is a zero-variance dynamics, it is equivalent to prove that $\boldsymbol{b} = \mathcal{D}\nabla V$ is a zero-variance dynamics where $V$ solves (11).

Notice that $\mathcal{J}_t(x) = e^{\int_0^t \boldsymbol{\nabla}\cdot(\mathcal{D}V)\big(\boldsymbol{X}_s(x)\big)\,\mathrm{d}s}$ and

$$
\begin{aligned}
\int_{-\infty}^{\infty} \big( \rho_1(\boldsymbol{X}_t(x)) - \rho_0(\boldsymbol{X}_t(x)) \big) \mathcal{J}_t(x)\,\mathrm{d}t &= \int_{-\infty}^{\infty} \boldsymbol{\nabla}\cdot(\mathcal{D}\nabla V)(\boldsymbol{X}_t(x)) \mathcal{J}_t(x)\,\mathrm{d}t \\
&= \int_{-\infty}^{\infty} \frac{\mathrm{d}}{\mathrm{d}t}\mathcal{J}_t(x)\,\mathrm{d}t = \lim_{t\to\infty} \mathcal{J}_t(x) - \lim_{t\to-\infty} \mathcal{J}_t(x).
\end{aligned}
$$

As long as $\mathcal{J}_t(x)$ vanishes when $|t| \to \infty$, such a dynamics $\boldsymbol{b} = \mathcal{D}\nabla V$ is indeed a zero-variance dynamics.

For a point $x \in \Omega$, suppose the gradient ascent trajectory under $\nabla V$ will converge to a (non-degenerate) local maximum of $V$, denoted as $x^\star$; by Proposition 2.2, the trajectory initiated from $x$ under $\boldsymbol{b} = \mathcal{D}\nabla V$ is the same and $\boldsymbol{X}_t(x) \to x^\star$ as $t \to \infty$ under the flow $\boldsymbol{b}$. Since $\mathcal{D}$ is strictly

positive, it also does not change the concavity of local extreme points: when $x^\star$ is a local maximum of $V$ (with $\nabla V(x^\star) = \mathbf{0}_d$ and $\nabla^2 V(x^\star) < 0$), then

$$\nabla \boldsymbol{b}(x^\star) = \nabla V(x^\star)\nabla\mathcal{D}(x^\star)^T + \mathcal{D}(x^\star)\nabla^2 V(x^\star) = \mathcal{D}(x^\star)\nabla^2 V(x^\star) < 0,$$

which implies that as $t \to \infty$,

$$\nabla \cdot \boldsymbol{b}\big(\boldsymbol{X}_t(x)\big) \to \nabla \cdot \boldsymbol{b}(x^\star) = \mathrm{tr}\left(\nabla\boldsymbol{b}(x^\star)\right) < 0.$$

By the same argument as in the case $\mathcal{D} = 1$ (i.e., Lemma E.3), we can establish the validity that $\mathcal{J}_t(x) \to 0$ as $|t| \to \infty$.

### E.3 A remark about the existence of Morse function

In Poisson's equation (11), a Morse function $V$ does not always exist for an arbitrary smooth density function $\rho_1$, e.g., when $\rho_1 = \rho_0$, $\mathcal{D} = 1$, we know $V = 0$ is the solution of (11) but $V = 0$ is not a Morse function. However, since Morse functions are dense in $C^\infty(\Omega, \mathbb{R})$ [4], we can always find a Morse function such that the dynamics $\boldsymbol{b} = \nabla V$ behaves almost like a zero-variance dynamics, which is summarized in the next proposition.

**Proposition E.5.** *Suppose $\Omega = [0,1]^d$ is a torus and $U_0, U_1 \in C^\infty(\Omega, \mathbb{R})$. Without loss of generality, assume $\mathcal{Z}_0 = \mathcal{Z}_1 = 1$. For any $\epsilon \in (0,1)$, there exists a Morse function $V$ such that the dynamics $\boldsymbol{b} = \nabla V$ provides an estimator $1 - \epsilon \leq \mathcal{A}(x) \leq 1 + \epsilon$ for almost all $x \in \Omega$ in the infinite-time NEIS method. Consequently, the variance $Var(\boldsymbol{b}) \leq \epsilon^2$.*

*Proof.* Denote $\theta := \inf\{\rho_0(x) : x \in \Omega\} \equiv e^{-\sup\{U_0(x): x \in \Omega\}} > 0$ since $U_0$ is smooth and $\Omega$ is compact. Since both $\rho_k = e^{-U_k}$ are smooth for $k = 0, 1$, one could approximate $\rho_1 - \rho_0$ by trigonometric polynomials $T_N(x) = \sum_{|\mu|_\infty \leq N, \mu \neq \mathbf{0}_d} a_\mu e^{i2\pi\langle\mu, x\rangle}$ such that

$$\|(\rho_1 - \rho_0) - T_N\|_{C^0(\Omega)} < \frac{\epsilon\theta}{2},$$

where $a_\mu = \int_\Omega e^{-i2\pi\langle\mu, x\rangle}(\rho_1 - \rho_0)(x)\,\mathrm{d}x \in \mathbb{C}$ are Fourier coefficients, $\mu \in \mathbb{Z}^d$ and $N \in \mathbb{N}$; see [7, Theorem 16]. Let $\Psi_N(x) = \sum_{|\mu|_\infty \leq N, \mu \neq \mathbf{0}_d} \frac{a_\mu}{-4\pi^2|\mu|^2} e^{i2\pi\langle\mu, x\rangle} \in C^\infty(\Omega, \mathbb{R})$. It is clear that $\Delta\Psi_N = T_N$. As Morse functions are dense, we can find a Morse function $V$ such that $\|V - \Psi_N\|_{C^2(\Omega)} < \frac{\epsilon\theta}{2}$ [4, Proposition 1.2.4], and in particular, $\|\Delta V - \Delta\Psi_N\|_{C^0(\Omega)} < \frac{\epsilon\theta}{2}$. Therefore,

$$\|\Delta V - (\rho_1 - \rho_0)\|_{C^0(\Omega)} \leq \|\Delta V - \Delta\Psi_N\|_{C^0(\Omega)} + \|\Delta\Psi_N - (\rho_1 - \rho_0)\|_{C^0(\Omega)} \leq \epsilon\theta. \tag{51}$$

By Proposition 3.1, we know that $\boldsymbol{b} = \nabla V$ is a zero-variance dynamics for $\widetilde{\rho}_1 := \rho_0 + \Delta V$. The Proposition 3.1 is proved under the assumption that densities are positive smooth functions for convenience and it is straightforward to verify that it also holds if $\rho_1$ is an arbitrary smooth function in Proposition 3.1. In particular, using the same argument in Appendix E.1 Step 2, we have for almost all $x \sim \rho_0$,

$$\frac{\int_\mathbb{R} \widetilde{\rho}_1(\boldsymbol{X}_t(x))\mathcal{J}_t(x)\,\mathrm{d}t}{\int_\mathbb{R} \rho_0(\boldsymbol{X}_t(x))\mathcal{J}_t(x)\,\mathrm{d}t} = 1 + \frac{\int_\mathbb{R} \Delta V(\boldsymbol{X}_t(x))\mathcal{J}_t(x)\,\mathrm{d}t}{\int_\mathbb{R} \rho_0(\boldsymbol{X}_t(x))\mathcal{J}_t(x)\,\mathrm{d}t} = 1 + \frac{\mathcal{J}_t(x)|_{t=-\infty}^{t=\infty}}{\int_\mathbb{R} \rho_0(\boldsymbol{X}_t(x))\mathcal{J}_t(x)\,\mathrm{d}t} = 1. \tag{52}$$

Hence,

$$\begin{aligned}
|\mathcal{A}(x) - 1| &\stackrel{(9)}{=} \left|\frac{\int_\mathbb{R} \rho_1(\boldsymbol{X}_t(x))\mathcal{J}_t(x)\,\mathrm{d}t}{\int_\mathbb{R} \rho_0(\boldsymbol{X}_t(x))\mathcal{J}_t(x)\,\mathrm{d}t} - 1\right| \\
&= \left|\frac{\int_\mathbb{R} \widetilde{\rho}_1(\boldsymbol{X}_t(x))\mathcal{J}_t(x)\,\mathrm{d}t}{\int_\mathbb{R} \rho_0(\boldsymbol{X}_t(x))\mathcal{J}_t(x)\,\mathrm{d}t} + \frac{\int_\mathbb{R}(\rho_1 - \widetilde{\rho}_1)(\boldsymbol{X}_t(x))\mathcal{J}_t(x)\,\mathrm{d}t}{\int_\mathbb{R} \rho_0(\boldsymbol{X}_t(x))\mathcal{J}_t(x)\,\mathrm{d}t} - 1\right| \\
&\stackrel{(52)}{=} \left|\frac{\int_\mathbb{R}(\rho_1 - \widetilde{\rho}_1)(\boldsymbol{X}_t(x))\mathcal{J}_t(x)\,\mathrm{d}t}{\int_\mathbb{R} \rho_0(\boldsymbol{X}_t(x))\mathcal{J}_t(x)\,\mathrm{d}t}\right| \\
&\leq \frac{\int_\mathbb{R}|(\rho_1 - \widetilde{\rho}_1)(\boldsymbol{X}_t(x))|\mathcal{J}_t(x)\,\mathrm{d}t}{\int_\mathbb{R} \rho_0(\boldsymbol{X}_t(x))\mathcal{J}_t(x)\,\mathrm{d}t}
\end{aligned}$$

$$\overset{(51)}{\leq} \frac{\int_{\mathbb{R}} \epsilon \theta \mathcal{J}_t(x)\,\mathrm{d}t}{\int_{\mathbb{R}} \theta \mathcal{J}_t(x)\,\mathrm{d}t} = \epsilon.$$

Since we assumed $\mathcal{Z}_1 = 1$,

$$\mathrm{Var}(\boldsymbol{b}) = \mathbb{E}_0[|\mathcal{A}|^2] - (\mathcal{Z}_1)^2 = \mathbb{E}_0|\mathcal{A} - \mathcal{Z}_1|^2 \leq \epsilon^2.$$

$\square$

### E.4 Solution of Poisson's equation (11) for Gaussian mixtures

**Lemma E.6.** *One solution of the Poisson's equation* $\Delta V = Ce^{-\frac{|x-\mu|^2}{2\sigma^2}}$ *with* $d \geq 2$ *on* $\Omega = \mathbb{R}^d$ *is* $V(x) = f(|x-\mu|)$, *where the function* $f : \mathbb{R}^+ \to \mathbb{R}$ *has the derivative*

$$f'(r) = C2^{d/2-1}\sigma^d r^{1-d} \int_0^{\frac{r^2}{2\sigma^2}} t^{d/2-1}e^{-t}\,\mathrm{d}t \equiv C2^{d/2-1}\sigma^d r^{1-d}\Gamma\left(d/2, \frac{r^2}{2\sigma^2}\right),$$

*where* $\Gamma(a, x) := \int_0^x t^{a-1}e^{-t}\,\mathrm{d}t$ *is the lower incomplete gamma function.*

*Proof.* Without loss of generality, let $\mu = \boldsymbol{0}_d$. Then a natural radial solution is given as $V(x) = f(|x|)$ for some scalar-valued function $f$. The above Poisson's equation becomes

$$f''(|x|) + f'(|x|)\frac{d-1}{|x|} = Ce^{-\frac{|x|^2}{2\sigma^2}}.$$

By some straightforward computation,

$$f'(r) = Cr^{-(d-1)}\int_0^r s^{d-1}e^{-s^2/(2\sigma^2)}\,\mathrm{d}s = C2^{d/2-1}\sigma^d r^{1-d}\underbrace{\int_0^{\frac{r^2}{2\sigma^2}} t^{d/2-1}e^{-t}\,\mathrm{d}t}_{\equiv \Gamma\left(d/2, \frac{|r|^2}{2\sigma^2}\right)}.$$

$\square$

**Proposition E.7.** *Suppose* $V$ *solves the following Poisson's equation on* $\Omega = \mathbb{R}^d$

$$\Delta V = \rho_1 - \rho_0$$

*where*

$$\rho_0(x) = \frac{1}{\sqrt{2\pi}^d}\exp\left(-\frac{|x|^2}{2}\right),$$

$$\rho_1(x) = \sum_{i=1}^n \omega_i \frac{1}{\sqrt{2\pi\sigma_i^2}^d}\exp\left(-\frac{|x-\mu_i|^2}{2\sigma_i^2}\right), \qquad n \in \mathbb{N},\ \mu_i \in \mathbb{R}^d, \sigma_i \in \mathbb{R}^+, \forall 1 \leq i \leq n,$$

*and* $\mu_i \neq \mu_j$ *if* $i \neq j$. *Then one solution for the gradient flow dynamics* $\boldsymbol{b} = \nabla V$ *is given as*

$$\boldsymbol{b}(x) = 2^{-1}\pi^{-d/2}\left(\sum_{i=1}^n \omega_i |x-\mu_i|^{-d}\Gamma\left(d/2, \frac{|x-\mu_i|^2}{2\sigma_i^2}\right)(x-\mu_i) - |x|^{-d}\Gamma\left(d/2, \frac{|x|^2}{2}\right)x\right). \quad (53)$$

*Proof.* We just need to apply the last lemma and the formula $\nabla V(x) = \frac{f'(|x|)}{|x|}x$ if $V(x) = f(|x|)$.

$\square$

**Proposition E.8.** *Suppose* $\boldsymbol{b}$ *is given in* (53). *Then*

$$\lim_{\left(\max_{i=1}^n \sigma_i\right)\to 0} \boldsymbol{b}(x)$$

$$= \begin{cases} 2^{-1}\pi^{-d/2}\left(\sum_{i\neq j}\omega_i|x-\mu_i|^{-d}\Gamma(d/2)(x-\mu_i) - |x|^{-d}\Gamma\left(d/2, \frac{|x|^2}{2}\right)x\right), & \text{if } x = \mu_j, \\[3mm] 2^{-1}\pi^{-d/2}\left(\sum_i\omega_i|x-\mu_i|^{-d}\Gamma(d/2)(x-\mu_i) - |x|^{-d}\Gamma\left(d/2, \frac{|x|^2}{2}\right)x\right), & \text{otherwise.} \end{cases}$$

*The limiting dynamics* $x \mapsto \lim_{\left(\max_{i=1}^n \sigma_i\right)\to 0} \boldsymbol{b}(x)$ *is continuous on the region* $\mathbb{R}^d \setminus \{\mu_i\}_{i=1}^n$.

*Proof.* If $x = \mu_j$, then

$$\boldsymbol{b}(x) = \frac{1}{2\pi^{d/2}}\left(\sum_{i \neq j}\omega_i|x - \mu_i|^{-d}\Gamma\left(d/2, \frac{|x - \mu_i|^2}{2\sigma_i^2}\right)(x - \mu_i) - |x|^{-d}\Gamma\left(d/2, \frac{|x|^2}{2}\right)x\right).$$

When $\sigma_i \to 0$ for all $i$, we know $\Gamma\left(d/2, \frac{|\mu_j - \mu_i|^2}{2\sigma_i^2}\right) \to \Gamma(d/2)$ when $i \neq j$ and hence we have the above result. Similarly, we can obtain the expression when $x \neq \mu_j$ for any $j$. $\qquad\square$

### E.5 Example: Poisson's equation yields a zero-variance dynamics

The example for Figure 1 is

$$\rho_0(x) = e^{-U_0(x)} = 1,$$
$$\rho_1(x) \propto e^{-U_1(x)} = \frac{\phi\left(x - \left[\begin{smallmatrix}0.3\\0.3\end{smallmatrix}\right]\right) + \phi\left(x - \left[\begin{smallmatrix}0.7\\0.3\end{smallmatrix}\right]\right) + \phi\left(x - \left[\begin{smallmatrix}0.3\\0.7\end{smallmatrix}\right]\right)}{3}, \tag{54}$$
$$\phi(x) = e^{2\cos(2\pi x_1) + 2\cos(2\pi x_2)}.$$

The periodic boundary condition in Proposition 3.1 helps to ease the technicalities in proving that the gradient dynamics $\boldsymbol{b} = \nabla V$ from solving the Poisson's equation (11) is a zero-variance dynamics, by removing the effect from the boundary $\partial\Omega$. The same conclusion, however, should hold if $V$ solves the Poisson's equation with Neumann boundary condition:

$$\Delta V = \rho_1 - \rho_0, \qquad \nabla V \cdot \boldsymbol{n} = 0 \text{ on } \partial\Omega, \tag{55}$$

where $\boldsymbol{n}$ is the normal vector of the boundary $\partial\Omega$. We consider the same model (54) and $\Omega = (0,1)^2$. The potential $V$ and flowlines of $\boldsymbol{b} = \nabla V$ are visualized in Figure 3 and we can numerically verify that $\mathcal{A}(x) = \mathcal{Z}_1$ for almost all $x \in \Omega$.

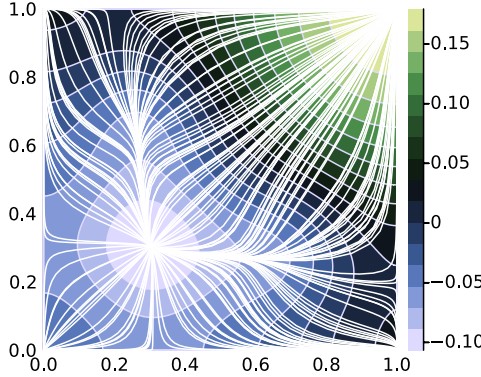

Figure 3: Contour plot of $V$ and flowlines of $\boldsymbol{b} = \nabla V$ for the model (54) on the domain $\Omega = (0,1)^2$ with Neumann boundary condition.

### E.6 Non-uniqueness of zero-variance dynamics

Recall from Proposition 2.2 that there are certain degrees of freedom to choose the dynamics: for a given $\boldsymbol{b}_1$, if we choose $\boldsymbol{b}_2 = \alpha\boldsymbol{b}_1$ where $\alpha \in C^\infty(\Omega, \mathbb{R})$ is strictly positive, then this function $\alpha$ can be absorbed into the time rescaling and it does not affect the variance of the sampling scheme. However, even if we remove this parameterization redundancy, zero-variance dynamics may still not be unique, e.g., due to the geometric rotational symmetry.

**Proposition E.9** (Non-uniqueness). *For given $\rho_0$ and $\rho_1$, there might exist more than one zero-variance dynamics (let us say $\boldsymbol{b}_1, \boldsymbol{b}_2$) but there is no scalar-valued function $\alpha$ such that $\boldsymbol{b}_2 = \alpha\boldsymbol{b}_1$.*

*Proof.* We construct an example to prove the non-uniqueness: let $d = 2$, $U_0(x) = |x|^2/2 + \ln(2\pi)$ and $U_1$ be given by

$$\exp\big(-U_1(x)\big) = \exp\big(-U_0(x)\big) + \frac{1}{2\pi}x_1x_2\exp\big(-x_1^4 - x_2^4\big).$$

We can easily verify that $|x|e^{-x^4} < \frac{3}{4}e^{-x^2/2}$ for any $x \in \mathbb{R}$. Then we know $\frac{7}{16}e^{-U_0(x)} < e^{-U_1(x)} < \frac{25}{16}e^{-U_0(x)}$. Therefore, $U_1$ is clearly well-defined and $\mathcal{Z}_1 = 1$. For the dynamics $\boldsymbol{b}(x) = \begin{bmatrix} v_1 \\ v_2 \end{bmatrix}$ with $v_1^2 + v_2^2 > 0$, we have $\mathcal{J}_t(x) = 1$ for any $x \in \Omega, t \in \mathbb{R}$, and

$$\int_{-\infty}^{\infty} e^{-U_1\big(\boldsymbol{X}_t(x)\big)} \mathcal{J}_t(x)\,\mathrm{d}t$$

$$= \int_{-\infty}^{\infty} e^{-U_0\big(\boldsymbol{X}_t(x)\big)} \mathcal{J}_t(x)\,\mathrm{d}t$$

$$+ \frac{1}{2\pi}\int_{-\infty}^{\infty} (x_1 + v_1 t)(x_2 + v_2 t)\exp\big(-(x_1 + v_1 t)^4 - (x_2 + v_2 t)^4\big)\,\mathrm{d}t.$$

When either $v_1 = 0$ or $v_2 = 0$, we can easily verify that

$$\frac{\int_{-\infty}^{\infty} e^{-U_1\big(\boldsymbol{X}_t(x)\big)} \mathcal{J}_t(x)\,\mathrm{d}t}{\int_{-\infty}^{\infty} e^{-U_0\big(\boldsymbol{X}_t(x)\big)} \mathcal{J}_t(x)\,\mathrm{d}t} = 1, \qquad \forall x \in \mathbb{R}^2.$$

Therefore, the variance is zero for two dynamics with orthogonal directions $\boldsymbol{b} = \begin{bmatrix} 1 \\ 0 \end{bmatrix}$ and $\boldsymbol{b} = \begin{bmatrix} 0 \\ 1 \end{bmatrix}$. It is clear that there is no scalar-valued function $\alpha$ such that $\boldsymbol{b}_2 = \alpha\boldsymbol{b}_1$, and thus the non-uniqueness is established. $\square$

### E.7   Connection to the Beckmann's problem.

The Poisson's equation (11) with $\mathcal{D} = 1$ is the Euler-Lagrange equation associated with

$$\min \int_{\Omega} |\boldsymbol{b}(x)|^p \,\mathrm{d}x \quad \text{subject to} \quad \boldsymbol{\nabla} \cdot \boldsymbol{b} = \rho_1 - \rho_0, \tag{56}$$

when $p = 2$. The variational problem in (56) is known as Beckman's problem of continuous transportation [5]; when $p = 1$, it is also related to optimal transport in $W_1$ Wasserstein distance [35, 36].

## F   Explicitly solvable zero-variance dynamics

In this section, we provide some examples with explicitly solvable zero-variance dynamics. Throughout this section, we consider $\Omega = \mathbb{R}^d$.

Table 1: Examples with explicitly solvable zero-variance dynamics for the infinite-time case with the domain $\Omega = \mathbb{R}^d$. By Proposition 2.2, given a zero-variance dynamics $\boldsymbol{b}$, any dynamics of the form $\alpha\boldsymbol{b}$ for some scalar-valued positive function $\alpha$ is also a zero-variance dynamics. In this table, we have removed such a degree of freedom.

| Dimension | $U_0$ and $U_1$ | $\boldsymbol{b}$ | Details |
|---|---|---|---|
| $d = 1$ | arbitrary | $\boldsymbol{b}(x) = 1$ | Appendix F.2 |
| general $d$ | $U_0(x) = |x|^2/2 + \frac{d}{2}\ln(2\pi)$ $U_1(x) = (x - \varpi)^T\Sigma^{-1}(x - \varpi)/2$ | $\boldsymbol{b}(x) = \Lambda x + v$ with $\Lambda = \ln\big(\Sigma^{-1/2}\big)$, $v = -\big(\mathbb{I}_d - \Sigma^{1/2}\big)^{-1}\ln\big(\Sigma^{-1/2}\big)\varpi.$ | Appendix F.3 |
| general $d$ | $\rho_0$ and $\rho_1$ have the same marginal distribution on the orthogonal subspace of $\{cv : c \in \mathbb{R}\}$ | $\boldsymbol{b}(x) = v$ | Appendix F.4 |

## F.1 Some general properties

Given a $b \in \mathfrak{B}$ and a distribution $\rho_0$, we study the family of $U_1$ such that $b$ is a zero-variance dynamics. Given an ODE flow map $X_\tau(\cdot)$ based on the dynamics $b$, let us introduce

$$U^\tau(x) := U_0\big(X_{-\tau}(x)\big) - \log\big(\mathcal{J}_{-\tau}(x)\big). \tag{57}$$

Then $\rho \propto e^{-U^\tau}$ is the push-forward distribution of the flow map $X_\tau(\cdot)$, i.e., $\rho = \big(X_\tau(\cdot)\big)\#\rho_0$. The family of distributions that can be written as a linear combination of such pushforward distributions can be characterized by

$$\mathfrak{F} := \Big\{U : e^{-U} \in \mathrm{Span}\{e^{-U^\tau}\}_{\tau \in \mathbb{R}}\Big\}.$$

We have:

**Proposition F.1.** *For every $U_1 \in \mathfrak{F}$, the variance $\mathrm{Var}(b)$ (if well-defined) is exactly zero, i.e., if the distribution $\rho_1 \propto e^{-U_1}$ is a linear combination of push-forward distributions by the ODE flows maps, then $\mathcal{Z}_1$ can be estimated with zero-variance by the infinite-time NEIS scheme.*

This proposition is proven in Appendix F.5 below. In words it says that, if we can learn a perfect neural ODE such that $\rho_1 = X_\tau(\cdot)\#\rho_0$ for some $\tau$, then such a dynamics is also the optimal one (i.e., zero-variance dynamics) for the infinite-time NEIS scheme. Conversely, if $U \notin \mathfrak{F}$, then is it still possible that $\mathrm{Var}(b) = 0$? The answer is positive:

**Proposition F.2.** *The exists a dynamics $b \in \mathfrak{B}$ and a $\rho_1 \propto e^{-U_1}$ such that $\mathcal{Z}_1$ can be computed with zero-variance but $\rho_1$ does not need to be a linear combination of $\big\{X_\tau(\cdot)\#\rho_0\big\}_{\tau \in \mathbb{R}}$ (namely, $U_1 \notin \mathfrak{F}$).*

This proposition is proven in Appendix F.6 below.

## F.2 Flows for the 1D case

Let us consider $b = 1$. Then we can compute $\mathcal{Z}_1$ via the infinite-time NEIS scheme with zero-variance for arbitrary potentials $U_0$ and $U_1$. This could be verified via direct computation: $\mathcal{J}_t(x) = 1$ and $X_t(x) = x + t$ for any $t, x \in \mathbb{R}$, and thus for an arbitrary $x$,

$$\mathcal{A}(x) = \frac{\int_{-\infty}^\infty e^{-U_1\big(X_t(x)\big)} \mathcal{J}_t(x)\, dt}{\int_{-\infty}^\infty e^{-U_0\big(X_t(x)\big)} \mathcal{J}_t(x)\, dt} = \frac{\int_{-\infty}^\infty e^{-U_1(x+t)}\, dt}{\int_{-\infty}^\infty e^{-U_0(x+t)}\, dt} = \mathcal{Z}_1.$$

Therefore, the variance is exactly zero.

Another perspective to understand this comes from Proposition F.1. For $b = 1$, we know $e^{-U^\tau(x)} = e^{-U_0(x-\tau)}$ in (57). By Proposition F.1, if the potential $U_1$ can be expressed as follows

$$e^{-U_1} = \int_{-\infty}^\infty f(\tau)e^{-U^\tau}\, d\tau = (e^{-U_0} * f),$$

then $\mathcal{Z}_1$ can be computed with zero-variance, where $f$ is a tempered distribution and $*$ means the convolution. Then it is sufficient to show the existence of such a $f$ for a generic $U_1$.

For a given potential $U_1$, we can solve the above equation for $f$ using Fourier transform; more specifically,

$$f = \mathscr{F}^{-1}\Big(\mathscr{F}(e^{-U_1})/\mathscr{F}(e^{-U_0})\Big),$$

where $\mathscr{F}^{(-1)}$ are (inverse) Fourier transform.

## F.3 Linear flows for Gaussian distributions

**Proposition F.3.** *Suppose $U_0(x) = |x|^2/2 + \frac{d}{2}\ln(2\pi)$ and $U_1(x) = (x - \varpi)^T \Sigma^{-1}(x - \varpi)/2$, where the covariance matrix $\Sigma$ is non-degenerate. A zero-variance linear dynamics is $b(x) = \Lambda x + v$ with*

$$\Lambda = \ln\Big(\Sigma^{-1/2}\Big), \qquad v = -\Big(\mathbb{I}_d - \Sigma^{1/2}\Big)^{-1} \ln\Big(\Sigma^{-1/2}\Big)\varpi. \tag{58}$$

Before presenting detailed proofs, let us make a few remarks about zero-variance dynamics in (58):

- If we further let $\Sigma = (1 - \epsilon)\mathbb{I}_d$ where $\epsilon \ll 1$ is an asymptotic parameter, then the above choice (58) can be approximated as follows:

$$\frac{\mathrm{d}}{\mathrm{d}t}\boldsymbol{X}_t(x) = \Lambda\boldsymbol{X}_t(x) + v \approx \frac{\epsilon}{2}\boldsymbol{X}_t(x) - \left(1 + \frac{\epsilon}{4}\right)\varpi + \mathcal{O}(\epsilon^2).$$

The leading order dynamics $\frac{\mathrm{d}}{\mathrm{d}t}\boldsymbol{X}_t(x) \approx -\varpi$ is consistent with the parallel velocity case below in Appendix F.4.

- For the above Gaussian case, the dynamics (58) can be regarded as the gradient flow dynamics of the following quadratic potential

$$V(x) = \frac{1}{2}\left(x - \left(\mathbb{I}_d - \Sigma^{1/2}\right)^{-1}\varpi\right)^T \ln\left(\Sigma^{-1/2}\right)\left(x - \left(\mathbb{I}_d - \Sigma^{1/2}\right)^{-1}\varpi\right).$$

Indeed, it is not hard to guess that the optimal dynamics might have a linear form in order to transport Gaussian distributions. It is natural to guess that $\boldsymbol{b} = -\nabla U_1$ or $\boldsymbol{b} = -\nabla(U_1 - U_0)$ might be zero-variance dynamics, but it could be verified that neither of them are zero-variance dynamics.

*Proof of Proposition F.3.* Suppose we consider a family of linear dynamics

$$\frac{\mathrm{d}}{\mathrm{d}t}\boldsymbol{X}_t(x) = \Lambda\boldsymbol{X}_t(x) + v, \qquad \boldsymbol{X}_0(x) = x.$$

Then $\boldsymbol{X}_t(x) = e^{\Lambda t}x + \Lambda^{-1}\left(e^{\Lambda t} - \mathbb{I}_d\right)v$ and $(\nabla \cdot \boldsymbol{b})(x) = \mathrm{tr}(\Lambda)$. Therefore, $U^{-\tau}$ defined in (57) has the following form for any $\tau \in \mathbb{R}$,

$$U^{-\tau}(x) = U_0\left(e^{\Lambda\tau}x + \Lambda^{-1}(e^{\Lambda\tau} - \mathbb{I}_d)v\right) - \mathrm{tr}(\Lambda)\tau.$$

By Proposition F.1, the variance is zero if

$$U_1(x) = U^{-1}(x) + C,$$

where $C$ is some constant. This condition can be simplified as

$$(x - \varpi)^T\Sigma^{-1}(x - \varpi)/2 = U_0\left(e^{\Lambda}x + \Lambda^{-1}(e^{\Lambda} - \mathbb{I}_d)v\right) - \mathrm{tr}(\Lambda) + C.$$

Thus, we just need to ensure

$$-\Lambda^{-1}(\mathbb{I}_d - e^{-\Lambda})v = \varpi, \qquad e^{\Lambda^T}e^{\Lambda} = \Sigma^{-1},$$

by matching the order of $x$ and, more specifically, we can choose $\Lambda$ and $v$ as in (58). $\qquad \square$

### F.4 Flows with parallel velocity

As we have mentioned, in the 1D case, the choice $\boldsymbol{b} = 1$ gives a zero-variance estimator for the infinite-time NEIS scheme. A straightforward generalization is to consider the following parallel velocity case

$$\boldsymbol{b}(x) = \alpha(x)v,$$

where $v \in \mathbb{R}^d$ and $\alpha$ is a positive scalar-valued function. Due to Proposition 2.2, it suffices to consider $\boldsymbol{b}(x) = v$. As we can always rotate the coordinate without affecting partition functions, without loss of generality, let us assume $v = \boldsymbol{e}_1$ for simplicity, where $\boldsymbol{e}_1$ is a vector with the first element to be 1 and zeros otherwise. For an arbitrary initial proposal $x$, only the first coordinate $x_1$ is changing under the dynamics $\boldsymbol{b} = \boldsymbol{e}_1$. The estimator essentially works like the 1D case above:

$$\mathcal{A}(x) = \frac{\int_{-\infty}^{\infty} e^{-U_1\left(\boldsymbol{X}_t(x)\right)}\mathcal{J}_t(x)\,\mathrm{d}t}{\int_{-\infty}^{\infty} e^{-U_0\left(\boldsymbol{X}_t(x)\right)}\mathcal{J}_t(x)\,\mathrm{d}t} = \frac{\int_{-\infty}^{\infty} e^{-U_1(q,x_2,x_3,\cdots,x_d)}\,\mathrm{d}q}{\int_{-\infty}^{\infty} e^{-U_0(q,x_2,x_3,\cdots,x_d)}\,\mathrm{d}q} = \mathcal{Z}_1 \times \frac{\widetilde{\rho}_1(x_2, x_3, \cdots, x_d)}{\widetilde{\rho}_0(x_2, x_3, \cdots, x_d)},$$

where $\widetilde{\rho}_k$ is the marginal distribution of $\rho_k$ for the subspace $\mathbb{R}^{d-1}$ by tracing out the first coordinate. Therefore, the dynamics $\boldsymbol{b} = \boldsymbol{e}_1$ is a zero-variance dynamics iff $\widetilde{\rho}_0 = \widetilde{\rho}_1$.

**Proposition F.4.** *Suppose $\boldsymbol{b}(x) = \alpha(x)v$ where $\alpha \in C^\infty(\mathbb{R}^d, \mathbb{R})$ with $\inf_{x \in \Omega}\alpha(x) > 0$ and $v \in \mathbb{R}^d$. Such a dynamics $\boldsymbol{b}$ gives a zero-variance estimator iff $\rho_0$ and $\rho_1$ have the same marginal distribution in the orthogonal space of $\{cv : c \in \mathbb{R}\}$.*

## F.5 Proof of Proposition F.1

**Lemma F.5.** *Fix the potential $U_0$ and a dynamics $\boldsymbol{b} \in \mathfrak{B}$. If $\frac{\int_{-\infty}^{\infty} e^{-U_k\left(\boldsymbol{X}_t(x)\right)} \mathcal{J}_t(x)\,\mathrm{d}t}{\int_{-\infty}^{\infty} e^{-U_0\left(\boldsymbol{X}_t(x)\right)} \mathcal{J}_t(x)\,\mathrm{d}t} = C_k$ is a constant function for $k \in \{2, 3\}$, then any mixture of $U_2$ and $U_3$, given below, also ensures that $\frac{\int_{-\infty}^{\infty} e^{-U\left(\boldsymbol{X}_t(x)\right)} \mathcal{J}_t(x)\,\mathrm{d}t}{\int_{-\infty}^{\infty} e^{-U_0\left(\boldsymbol{X}_t(x)\right)} \mathcal{J}_t(x)\,\mathrm{d}t}$ is a constant function, as long as $U$ is a valid potential function:*

$$U = -\log\left(\omega_2 e^{-U_2} + \omega_3 e^{-U_3}\right), \qquad \omega_2, \omega_3 \in \mathbb{R}.$$

*Proof.* We can easily observe that

$$\int_{-\infty}^{\infty} e^{-U\left(\boldsymbol{X}_t(x)\right)} \mathcal{J}_t(x)\,\mathrm{d}t = \int_{-\infty}^{\infty} \omega_2 e^{-U_2(\boldsymbol{X}_t(x))} \mathcal{J}_t(x) + \omega_3 e^{-U_3(\boldsymbol{X}_t(x))} \mathcal{J}_t(x)\,\mathrm{d}t$$

$$= (C_2\omega_2 + C_3\omega_3)\int_{-\infty}^{\infty} e^{-U_0(\boldsymbol{X}_t(x))} \mathcal{J}_t(x)\,\mathrm{d}t.$$

$\square$

**Lemma F.6.** *For the distribution $\rho_1 \propto e^{-U^\tau + C}$, the partition function $\mathcal{Z}_1$ can be computed with zero-variance.*

*Proof.* We only need to verify the case $C = 0$. Then

$$\int_{-\infty}^{\infty} e^{-U^\tau\left(\boldsymbol{X}_t(x)\right)} \mathcal{J}_t(x)\,\mathrm{d}t = \int_{-\infty}^{\infty} e^{-U_0\left(\boldsymbol{X}_{t-\tau}(x)\right)} \mathcal{J}_{-\tau}\left(\boldsymbol{X}_t(x)\right)\mathcal{J}_t(x)\,\mathrm{d}t$$

$$\overset{(25)}{=} \int_{-\infty}^{\infty} e^{-U_0\left(\boldsymbol{X}_{t-\tau}(x)\right)} \frac{\mathcal{J}_{t-\tau}(x)}{\mathcal{J}_t(x)} \mathcal{J}_t(x)\,\mathrm{d}t$$

$$= \int_{-\infty}^{\infty} e^{-U_0\left(\boldsymbol{X}_{t-\tau}(x)\right)} \mathcal{J}_{t-\tau}(x)\,\mathrm{d}t$$

$$= \int_{-\infty}^{\infty} e^{-U_0\left(\boldsymbol{X}_t(x)\right)} \mathcal{J}_t(x)\,\mathrm{d}t.$$

This means that $\boldsymbol{b}$ is a zero-variance dynamics for the distribution $\rho_1$. $\square$

*Proof of Proposition F.1.* Combine the above two lemmas. $\square$

## F.6 Proof of Proposition F.2

Consider a 2D example with $U_0(x) = \frac{|x|^2}{2} + \ln(2\pi)$ and $\boldsymbol{b} = \begin{bmatrix} 1 \\ 0 \end{bmatrix}$. Then $\mathcal{J}_t(x) = 1$, and $U^\tau(x) = \frac{(x_1-\tau)^2 + x_2^2}{2} + \ln(2\pi)$ where $x = \begin{bmatrix} x_1 \\ x_2 \end{bmatrix}$. Let us consider

$$e^{-U_1(x)} := e^{-U^1(x)} + \epsilon x_1 e^{-x_1^4 - x_2^4},$$

where $\epsilon := \frac{1}{2\pi}\left(2\sup_{y\in\mathbb{R}} |y|e^{-y^4+(y-1)^2/2} \sup_{y\in\mathbb{R}} e^{-y^4+y^2/2}\right)^{-1} > 0$. It could be straightforwardly verified that $U_1$ is a well-defined potential and

$$\int_{-\infty}^{\infty} e^{-U_1\left(\boldsymbol{X}_t(x)\right)} \mathcal{J}_t(x)\,\mathrm{d}t = \int_{-\infty}^{\infty} e^{-U^1\left(\boldsymbol{X}_t(x)\right)} \mathcal{J}_t(x)\,\mathrm{d}t + \epsilon \int_{-\infty}^{\infty} (x_1 + t)e^{-(x_1+t)^4 - x_2^4}\,\mathrm{d}t$$

$$= \int_{-\infty}^{\infty} e^{-U^1\left(\boldsymbol{X}_t(x)\right)} \mathcal{J}_t(x)\,\mathrm{d}t$$

$$= \int_{-\infty}^{\infty} e^{-U_0\left(\boldsymbol{X}_t(x)\right)} \mathcal{J}_t(x)\,\mathrm{d}t.$$

The last equality holds by Lemma F.6. Therefore, $\boldsymbol{b}$ is a zero-variance dynamics for $\rho_0$ and $\rho_1$. However, $U_1 \notin \mathfrak{F}$ because $\int_{-\infty}^{\infty} f(\tau)e^{-U^\tau(x)}\,\mathrm{d}\tau$ must be a separable function, whereas $U_1$ is not.

# G Proof of Proposition 3.2 and more discussions

## G.1 Proof of Proposition 3.2

We proceed in three steps:

**Step 1:** Let us first assume the existence of $\varkappa \in C^1(D, \mathbb{R})$ satisfying (14), where $D$ is an open subset of $\Omega$ and $\Omega \backslash D$ has measure zero. We shall verify that $\boldsymbol{T} \# \rho_0 = \rho_1$ almost everywhere.

From (14), let us replace $x$ by $\boldsymbol{X}_\epsilon(x)$,

$$0 \stackrel{(25)}{=} \frac{1}{\mathcal{J}_\epsilon(x)} \left( \int_{-\infty}^0 \rho_0(\boldsymbol{X}_{s+\epsilon}(x)) \mathcal{J}_{s+\epsilon}(x) \, \mathrm{d}s - \int_{-\infty}^{\varkappa(\boldsymbol{X}_\epsilon(x))} \rho_1(\boldsymbol{X}_{s+\epsilon}(x)) \mathcal{J}_{s+\epsilon}(x) \, \mathrm{d}s \right).$$

By straightforward simplification,

$$0 = \int_{-\infty}^{0+\epsilon} \rho_0(\boldsymbol{X}_s(x)) \mathcal{J}_s(x) \, \mathrm{d}s - \int_{-\infty}^{\varkappa(\boldsymbol{X}_\epsilon(x))+\epsilon} \rho_1(\boldsymbol{X}_s(x)) \mathcal{J}_s(x) \, \mathrm{d}s.$$

By taking the derivative with respect to $\epsilon$ at $\epsilon = 0$,

$$\rho_0(x) = \rho_1(\boldsymbol{X}_{\varkappa(x)}(x)) \mathcal{J}_{\varkappa(x)}(x) \big( 1 + \langle \nabla \varkappa(x), \boldsymbol{b}(x) \rangle \big)$$
$$= \rho_1(\boldsymbol{T}(x)) \mathcal{J}_{\varkappa(x)}(x) \big( 1 + \langle \nabla \varkappa(x), \boldsymbol{b}(x) \rangle \big).$$

To show that $\boldsymbol{T} \# \rho_0 = \rho_1$, we need to verify that $\rho_0(x) = \rho_1(\boldsymbol{T}(x)) \mathcal{J}_{\boldsymbol{T}}(x)$ where $\mathcal{J}_{\boldsymbol{T}}(x) = \left| \det \big( \nabla_x \boldsymbol{X}_{\varkappa(x)}(x) \big) \right|$. Therefore, it remains to prove that

$$\det \big( \nabla_x \boldsymbol{X}_{\varkappa(x)}(x) \big) = \mathcal{J}_{\varkappa(x)}(x) \big( 1 + \langle \nabla \varkappa(x), \boldsymbol{b}(x) \rangle \big). \tag{59}$$

By direct computation,

$$\nabla_x \boldsymbol{X}_{\varkappa(x)}(x) \stackrel{(45)}{=} \boldsymbol{C}_{\varkappa(x),0}(x) + \boldsymbol{b}\big(\boldsymbol{X}_{\varkappa(x)}(x)\big) \big(\nabla \varkappa(x)\big)^T.$$

By the matrix determinant lemma,

$$\det \big( \nabla_x \boldsymbol{X}_{\varkappa(x)}(x) \big) = \left( 1 + \Big\langle \nabla \varkappa(x), \big(\boldsymbol{C}_{\varkappa(x),0}(x)\big)^{-1} \boldsymbol{b}\big(\boldsymbol{X}_{\varkappa(x)}(x)\big) \Big\rangle \right) \det \big( \boldsymbol{C}_{\varkappa(x),0}(x) \big)$$
$$= \left( 1 + \Big\langle \nabla \varkappa(x), \boldsymbol{b}(x) \Big\rangle \right) \mathcal{J}_{\varkappa(x)}(x),$$

where we used (40) and Lemma D.6 to get the second line. The last equation verifies (59) and therefore, $\boldsymbol{T} \# \rho_0 = \rho_1$ almost everywhere.

**Step 2:** We shall explain $D$ and establish the existence of $\varkappa \in C^1(D, \mathbb{R})$.

Suppose we denote the local minima of $V$ as $\mathscr{X}_1, \mathscr{X}_2, \cdots, \mathscr{X}_a$, and local maxima of $V$ as $\mathscr{Y}_1, \mathscr{Y}_2, \cdots, \mathscr{Y}_b$, where $a, b \in \mathbb{N}$. Then in the proof of Proposition 3.1, we have already mentioned that

$$D := \left\{ x \in \Omega \,\Big|\, \lim_{t \to -\infty} \boldsymbol{X}_t(x) = \mathscr{X}_i, \ \lim_{t \to \infty} \boldsymbol{X}_t(x) = \mathscr{Y}_j, \text{ for some } i, j \right\}$$

is an open subset of $\mho(\boldsymbol{b})$ and $\Omega \backslash D$ has measure zero, due to the assumption that $V$ is a Morse function.

Consider the following function $L(x, t) : D \times \mathbb{R} \to \mathbb{R}$, defined as

$$L(x, t) := \int_{-\infty}^0 \rho_0(\boldsymbol{X}_s(x)) \mathcal{J}_s(x) \, \mathrm{d}s - \int_{-\infty}^t \rho_1(\boldsymbol{X}_s(x)) \mathcal{J}_s(x) \, \mathrm{d}s.$$

We can observe that

- $\lim_{t \to -\infty} L(x, t) = \int_{-\infty}^0 \rho_0(\boldsymbol{X}_s(x)) \mathcal{J}_s(x) \, \mathrm{d}s > 0.$

- Besides,

$$\lim_{t \to \infty} L(x,t) = \int_{-\infty}^{0} \rho_0(\boldsymbol{X}_s(x))\mathcal{J}_s(x)\,\mathrm{d}s - \int_{-\infty}^{\infty} \rho_1(\boldsymbol{X}_s(x))\mathcal{J}_s(x)\,\mathrm{d}s$$

$$= -\int_{0}^{\infty} \rho_0(\boldsymbol{X}_s(x))\mathcal{J}_s(x)\,\mathrm{d}s < 0,$$

where the second equality comes from the fact that $\boldsymbol{b} = \nabla V$ is a zero-variance dynamics; see (12).

- With fixed $x$, the function $t \mapsto L(x,t)$ is continuously differentiable and is strictly monotonically decreasing.

These imply that for each $x \in D$, there exists a unique $\varkappa(x)$ such that $L\big(x, \varkappa(x)\big) = 0$ by the intermediate value theorem. Therefore, $\varkappa$ is well-defined via (14).

Next, we need to prove that such a function $\varkappa \in C^1(D, \mathbb{R})$, which can be immediately obtained by the implicit function theorem [34], provided that we can prove $L \in C^1(D \times \mathbb{R}, \mathbb{R})$. Due to the smoothness assumption on $\rho_0$ and $\rho_1$, it is clear that $\partial_t L(x,t) = -\rho_1(\boldsymbol{X}_t(x))\mathcal{J}_t(x) < 0$ exists and is continuous. Therefore, the task becomes to prove that $\nabla_x L(x,t)$ exists and is continuous. Since it is clear that $\int_0^t \rho_1(\boldsymbol{X}_s(x))\mathcal{J}_s(x)\,\mathrm{d}s$ is continuously differentiable with respect to $x$, it is then sufficient to prove that

$$G_k(x) := \int_{-\infty}^{0} \rho_k(\boldsymbol{X}_s(x))\mathcal{J}_s(x)\,\mathrm{d}s$$

is continuously differentiable for $k \in \{0,1\}$.

**Step 3:** Prove that $G_k \in C^1(D, \mathbb{R})$.

In Appendix E.1, we have proved that $G_k$ is continuous; see Lemma E.4 in particular. Next, we first verify that $G_k$ is differentiable and then verify that $\nabla G_k$ is also continuous.

*Part 1: $G_k$ is differentiable.*

We want to verify that $G_k$ is differentiable and in particular

$$\nabla_x G_k(x) = \int_{-\infty}^{0} \nabla_x \big(\rho_k(\boldsymbol{X}_s(x))\mathcal{J}_s(x)\big)\,\mathrm{d}s. \tag{60}$$

Let us consider an arbitrary $x \in D$. Without loss of generality, suppose its limit in the backward branch is $\mathscr{X}_1 = \lim_{t \to -\infty} \boldsymbol{X}_t(x)$. Let us focus on a local neighborhood $B_\delta(x)$ such that $\lim_{t \to -\infty} \boldsymbol{X}_t(z) = \mathscr{X}_1$ for all $z \in B_\delta(x)$. Such a small $\delta$ exists because $\mathscr{X}_1$ is a strict local minimum and $\boldsymbol{b} = \nabla V$ is smooth from the assumption that $V$ is a Morse function. In order to verify that $G_k$ is differentiable (i.e., (60)), by the Leibniz rule (see e.g., [18, Theorem 6.28]), it is sufficient to prove that there exists an integrable function $Q : (-\infty, 0] \to \mathbb{R}$ such that

$$\big|\nabla_z \big(\rho_k(\boldsymbol{X}_s(z))\mathcal{J}_s(z)\big)\big| \le Q(s), \qquad \forall s \in (-\infty, 0], \ \forall z \in B_\delta(x). \tag{61}$$

By direct computation,

$$\nabla_z \big(\rho_k(\boldsymbol{X}_s(z))\mathcal{J}_s(z)\big)$$

$$= \rho_k(\boldsymbol{X}_s(z))\mathcal{J}_s(z)\Big(-(\nabla \boldsymbol{X}_s(z))^T \nabla U_k(\boldsymbol{X}_s(z)) + \int_0^s (\nabla \boldsymbol{X}_r(z))^T \nabla(\boldsymbol{\nabla} \cdot \boldsymbol{b})(\boldsymbol{X}_r(z))\,\mathrm{d}r\Big) \tag{62}$$

$$\overset{(45)}{=} \rho_k(\boldsymbol{X}_s(z))\mathcal{J}_s(z)\Big(-(\boldsymbol{C}_{s,0}(z))^T \nabla U_k(\boldsymbol{X}_s(z)) + \int_0^s (\boldsymbol{C}_{r,0}(z))^T \nabla(\boldsymbol{\nabla} \cdot \boldsymbol{b})(\boldsymbol{X}_r(z))\,\mathrm{d}r\Big).$$

Since the domain $\Omega$ is assumed to be a torus, we know $\rho_k$, $\nabla U_k$ and $\nabla(\boldsymbol{\nabla} \cdot \boldsymbol{b})$ are uniformly bounded on the domain $\Omega$. Therefore,

$$\big|\nabla_z \big(\rho_k(\boldsymbol{X}_s(z))\mathcal{J}_s(z)\big)\big| \lesssim \mathcal{J}_s(z)\Big(\|\boldsymbol{C}_{s,0}(z)\| + \int_s^0 \|\boldsymbol{C}_{r,0}(z)\|\,\mathrm{d}r\Big). \tag{63}$$

Recall from (5) and (40) that

$$\partial_s \mathcal{J}_s(z) = (\boldsymbol{\nabla} \cdot \boldsymbol{b})(\boldsymbol{X}_s(z))\mathcal{J}_s(z), \qquad \mathcal{J}_0(z) = 1;$$
$$\partial_s \boldsymbol{C}_{s,0}(z) = \nabla \boldsymbol{b}(\boldsymbol{X}_s(z))\boldsymbol{C}_{s,0}(z), \qquad \boldsymbol{C}_{0,0}(z) = \mathbb{I}_d.$$

Recall that $\boldsymbol{X}_s(z) \to \mathscr{X}_1$ as $s \to -\infty$; the value $\boldsymbol{\nabla} \cdot \boldsymbol{b}(\mathscr{X}_1)$ and the hessian matrix $\nabla \boldsymbol{b}(\mathscr{X}_1)$ are both strictly positive, which imply that $\mathcal{J}_s(x)$ and $\|\boldsymbol{C}_{s,0}(z)\|$ are both decaying as $s \to -\infty$.

Let us denote $\upsilon > 0$ as the smallest eigenvalue of $\nabla \boldsymbol{b}(\mathscr{X}_1)$. For a given $\epsilon$ with $0 < \epsilon < \min\{\boldsymbol{\nabla} \cdot \boldsymbol{b}(\mathscr{X}_1), \upsilon\}$, let us define

$$\mathcal{E} = \{z \in \Omega : \boldsymbol{\nabla} \cdot \boldsymbol{b}(z) > \nabla \cdot \boldsymbol{b}(\mathscr{X}_1) - \epsilon, \ \nabla \boldsymbol{b}(z) > (\upsilon - \epsilon)\mathbb{I}_d\},$$

which is an open neighborhood of $\mathscr{X}_1$. We can find a subset of $\mathcal{E}$, denoted as $\mathcal{E}_{\text{trap}}$, such that $\mathcal{E}_{\text{trap}}$ is a trapping region of the dynamics $\boldsymbol{b}$ for the backward branch. Thus, we can find a negative time $\tau$ (which possibly depends on $\epsilon$) such that $\boldsymbol{X}_s(x) \in \mathcal{E}_{\text{trap}}$ for all $s \le \tau$. If we choose a $\delta$ small enough, then we can even ensure that

$$\boldsymbol{X}_s(z) \in \mathcal{E}_{\text{trap}}, \qquad \forall s \le \tau, \forall z \in B_\delta(x), \tag{64}$$

due to the smoothness of $\boldsymbol{b}$ and the construction that $\mathcal{E}_1$ is a trapping region. Therefore, when $s \le \tau$, $\mathcal{J}_s(z)$ decays to zero exponentially fast as $s \to -\infty$ with a rate at least $\boldsymbol{\nabla} \cdot \boldsymbol{b}(\mathscr{X}_1) - \epsilon$; similarly, the matrix norm $\|\boldsymbol{C}_{s,0}(z)\|$ also decays to zero exponentially fast as $s \to -\infty$ with a rate at least $\upsilon - \epsilon$. On the region $(-\infty, 0] \times B_\delta(x)$, we can readily obtain

$$\mathcal{J}_s(z) \lesssim e^{(\boldsymbol{\nabla} \cdot \boldsymbol{b}(\mathscr{X}_1) - \epsilon)s}, \qquad \|\boldsymbol{C}_{s,0}(z)\| \lesssim e^{(\upsilon - \epsilon)s}. \tag{65}$$

By plugging the above estimates into (63), we know that

$$\left|\nabla_z\big(\rho_k(\boldsymbol{X}_s(z))\mathcal{J}_s(z)\big)\right| \lesssim \mathcal{J}_s(z) \lesssim e^{(\boldsymbol{\nabla} \cdot \boldsymbol{b}(\mathscr{X}_1) - \epsilon)s}. \tag{66}$$

The function $e^{(\boldsymbol{\nabla} \cdot \boldsymbol{b}(\mathscr{X}_1) - \epsilon)s}$ is integrable on $(-\infty, 0]$ and this serves as the function $Q$ needed in (61) with some multiplicative constant.

*Part 2: $\nabla G_k$ is continuous.*

Let us denote $H(z, s) := \nabla_z\big(\rho_k(\boldsymbol{X}_s(z))\mathcal{J}_s(z)\big)$. From (62), we can observe that $\nabla_z H(z, s)$ is continuous with respect to $z$ with

$$\nabla_z H(z, s) = \frac{H(z, s)\big(H(z, s)\big)^T}{\rho_k(\boldsymbol{X}_s(z))\mathcal{J}_s(z)} +$$
$$\rho_k(\boldsymbol{X}_s(z))\mathcal{J}_s(z)\nabla_z\Big(-(\boldsymbol{C}_{s,0}(z))^T\nabla U_k(\boldsymbol{X}_s(z))\Big) +$$
$$\rho_k(\boldsymbol{X}_s(z))\mathcal{J}_s(z)\int_0^s \nabla_z\Big((\boldsymbol{C}_{r,0}(z))^T\nabla(\boldsymbol{\nabla} \cdot \boldsymbol{b})(\boldsymbol{X}_r(z))\Big)\,\mathrm{d}r.$$

**Lemma G.1.** *For a given smooth vector field $W \in C^\infty(\Omega, \mathbb{R}^d)$, there exists a constant $C$ such that for any $z \in B_\delta(x)$ and $s \in (-\infty, 0]$, we have*

$$\left\|\nabla_z\big(\boldsymbol{C}_{s,0}(z)^T W(\boldsymbol{X}_s(z))\big)\right\| \le Ce^{(\upsilon - \epsilon)s}.$$

By this lemma and the estimates in (66),

$$\|\nabla_z H(z, s)\| \lesssim \left\|\frac{H(z, s)\big(H(z, s)\big)^T}{\rho_k(\boldsymbol{X}_s(z))\mathcal{J}_s(z)}\right\| + \mathcal{J}_s(z)e^{(\upsilon - \epsilon)s} + \mathcal{J}_s(z)\int_s^0 e^{(\upsilon - \epsilon)r}\,\mathrm{d}r$$
$$\lesssim \mathcal{J}_s(z) \lesssim e^{(\boldsymbol{\nabla} \cdot \boldsymbol{b}(\mathscr{X}_1) - \epsilon)s},$$

which readily leads into the continuity of $\nabla G_k$ based on (60).

*Part 3: Proof of Lemma G.1*

By direct computation

$$\nabla_{z_j}\Big(\boldsymbol{C}_{s,0}(z)^T W(\boldsymbol{X}_s(z))\Big)_i$$

$$= \sum_l \nabla_{z_j}\Big(\big(\boldsymbol{C}_{s,0}(z)\big)_{l,i} W_l(\boldsymbol{X}_s(z))\Big)$$

$$\overset{(45)}{=} \sum_l \Big(\nabla_{z_j}\big(\boldsymbol{C}_{s,0}(z)\big)_{l,i}\Big) W_l(\boldsymbol{X}_s(z)) + \sum_{l,m}\big(\boldsymbol{C}_{s,0}(z)\big)_{l,i}\big(\nabla W(\boldsymbol{X}_s(z))\big)_{l,m}\big(\boldsymbol{C}_{s,0}(z)\big)_{m,j}.$$

Since $W$ is assumed to be smooth on the torus $\Omega$, we know $W$ and $\nabla W$ are uniformly bounded. Previously, we know that $\|\boldsymbol{C}_{s,0}(z)\| \lesssim e^{(\upsilon-\epsilon)s}$. Therefore, we only need to prove that for any $1 \le \ell \le d$,

$$\|\partial_{z_\ell}\boldsymbol{C}_{s,0}(z)\| \lesssim e^{(\upsilon-\epsilon)s}. \tag{67}$$

From (41), we have

$$\partial_s\big(\partial_{z_\ell}\boldsymbol{C}_{s,0}(z)\big) = \nabla\boldsymbol{b}(\boldsymbol{X}_s(z))\big(\partial_{z_\ell}\boldsymbol{C}_{s,0}(z)\big) + \boldsymbol{S}(s,z), \qquad \big(\partial_{z_\ell}\boldsymbol{C}_{0,0}(z)\big) = \boldsymbol{0}_{d\times d},$$

$$\big(\boldsymbol{S}(s,z)\big)_{i,j} = \sum_{n,m}(\partial_{z_i,z_n,z_m}V)(\boldsymbol{X}_s(z))\big(\boldsymbol{C}_{s,0}(z)\big)_{m,\ell}\big(\boldsymbol{C}_{s,0}(z)\big)_{n,j}. \tag{68}$$

Since $\|\boldsymbol{C}_{s,0}(z)\| \lesssim e^{(\upsilon-\epsilon)s}$ from (65), the source term $\|\boldsymbol{S}(s,z)\|$ also decays exponentially fast with rate $2(\upsilon-\epsilon)$ as $s \to -\infty$, namely,

$$\|\boldsymbol{S}(s,z)\| \lesssim e^{2(\nu-\epsilon)s}, \qquad \forall s \in (-\infty, 0], z \in B_\delta(x). \tag{69}$$

By rewriting (68) in the integral form and by (40), we have

$$\big(\partial_{z_\ell}\boldsymbol{C}_{s,0}(z)\big) = -\int_s^0 \boldsymbol{C}_{s,0}(z)\big(\boldsymbol{C}_{r,0}(z)\big)^{-1}\boldsymbol{S}(r,z)\,\mathrm{d}r = -\int_s^0 \boldsymbol{C}_{s,r}(z)\boldsymbol{S}(r,z)\,\mathrm{d}r.$$

To prove Lemma G.1, we only need to consider the case $s \ll 0$. Suppose we consider $s \le \tau$ only; recall the role of $\tau$ in (64). Then we could obtain that when $s \le r \le \tau$

$$\|\boldsymbol{C}_{s,r}(z)\| = \left\|\exp_{\mathcal{T}_\rightarrow}\Big(-\int_s^r \nabla\boldsymbol{b}(\boldsymbol{X}_u(z))\,\mathrm{d}u\Big)\right\| \overset{(64)}{\le} e^{-(\nu-\epsilon)(r-s)}, \tag{70}$$

where $\exp_{\mathcal{T}_\rightarrow}$ is the anti-chronological time-ordered operator exponential. We can separate the above integral form using $\tau$ and obtain the following estimates: when $s \le \tau$,

$$\|\partial_{z_\ell}\boldsymbol{C}_{s,0}(z)\| \le \left\|\int_s^\tau \boldsymbol{C}_{s,r}(z)\boldsymbol{S}(r,z)\,\mathrm{d}r\right\| + \left\|\int_\tau^0 \boldsymbol{C}_{s,r}(z)\boldsymbol{S}(r,z)\,\mathrm{d}r\right\|$$

$$\le \int_s^\tau \|\boldsymbol{C}_{s,r}(z)\boldsymbol{S}(r,z)\|\,\mathrm{d}r + \int_\tau^0 \left\|\boldsymbol{C}_{s,0}(z)\big(\boldsymbol{C}_{r,0}(z)\big)^{-1}\boldsymbol{S}(r,z)\right\|\,\mathrm{d}r$$

$$\le \int_s^\tau \underbrace{\|\boldsymbol{C}_{s,r}(z)\|}_{\text{use (70)}}\underbrace{\|\boldsymbol{S}(r,z)\|}_{\text{use (69)}}\,\mathrm{d}r + \underbrace{\|\boldsymbol{C}_{s,0}(z)\|}_{\text{use (65)}}\int_\tau^0 \underbrace{\left\|\big(\boldsymbol{C}_{r,0}(z)\big)^{-1}\right\|}_{=\mathcal{O}(1)}\cdot\underbrace{\|\boldsymbol{S}(r,z)\|}_{\text{use (69)}}\,\mathrm{d}r$$

$$\lesssim \int_s^\tau e^{-(\upsilon-\epsilon)(r-s)}e^{2(\upsilon-\epsilon)r}\,\mathrm{d}r + e^{(\upsilon-\epsilon)s}\underbrace{\int_\tau^0 e^{2(\upsilon-\epsilon)r}\,\mathrm{d}r}_{=\mathcal{O}(1)}$$

$$\lesssim e^{(\upsilon-\epsilon)s} + e^{(\upsilon-\epsilon)s} \lesssim e^{(\upsilon-\epsilon)s}.$$

This verifies (67) for any $z \in B_\delta(x)$ and $s \le \tau$. When $s \in [-\tau, 0]$, we can simply choose the prefactor large enough so that (67) holds, as $\tau$ is a finite value. Hence, Lemma G.1 is verified.

## G.2 Examples

We elaborate on Proposition 3.2 by concrete examples. For these examples, we estimate $\varkappa$ from (14) either analytically or numerically, and then we validate Proposition 3.2 by comparing $\rho_1$ and the empirical distribution of $\boldsymbol{X}_{\varkappa(x)}(x)$ where $x \sim \rho_0$.

### G.2.1 Gaussian examples in 1D

For the case $U_0(x) = \frac{|x|^2}{2} + \frac{1}{2}\ln(2\pi)$ and $U_1(x) = \frac{|x-\omega|^2}{2\sigma^2}$ with $\sigma < 1$, from Table 1, we already know that a zero-variance dynamics is $\boldsymbol{b}(x) = x - \frac{\omega}{1-\sigma}$. Then $\boldsymbol{X}_t(x) = e^t x - (e^t - 1)\frac{\omega}{1-\sigma}$ for any $t, x \in \mathbb{R}$ and $\mathcal{J}_t(x) = e^t$ for any $t \in \mathbb{R}$. By direct computation

$$\frac{\int_{-\infty}^{\theta} \rho_1(\boldsymbol{X}_s(x))\mathcal{J}_s(x)\mathrm{d}s}{\int_{-\infty}^{0} \rho_0(\boldsymbol{X}_s(x))\mathcal{J}_s(x)\mathrm{d}s} = -\frac{\mathrm{erf}(\frac{\omega}{\sqrt{2}(1-\sigma)}) + \mathrm{erf}\left(\frac{-e^{\theta}(\omega+x(\sigma-1)+\omega\sigma)}{\sqrt{2}\sigma(\sigma-1)}\right)}{\mathrm{erf}(\frac{x}{\sqrt{2}}) + \mathrm{erf}\left(\frac{\omega}{\sqrt{2}(\sigma-1)}\right)}.$$

By solving $\varkappa$ in (14) using the last equation, we have

$$\varkappa(x) = \log(\sigma), \qquad \forall x \neq \frac{\omega}{1-\sigma},$$

which is independent of the state $x$.

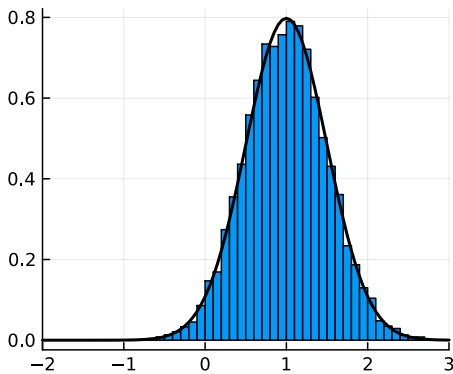

Figure 4: This figure shows the histogram of sample points of $\boldsymbol{X}_{\varkappa(x)}(x)$ (blue) and the distribution $\rho_1$ (black) for the 1D Gaussian example in Appendix G.2.1 with $\omega = 1$ and $\sigma = 0.5$.

### G.2.2 Three-mode mixture on a 2D torus

We consider the model (54) on the torus $\Omega = [0, 1]^2$ and recall that the zero-variance dynamics has been shown in Figure 1. Then the contour plot of $\varkappa$ is visualized in Figure 5. Moreover, in the same figure, the empirical distributions of $\boldsymbol{X}_{\varkappa(x)}(x)$ and contour plots of $\rho_1$ are provided, which numerically verifies Proposition 3.2.

### G.2.3 An example on $(0, 1)^2$ with Neumann boundary condition

We consider the model

$$V(x) = \gamma \cos(2\pi x_1) \cos(2\pi x_2), \qquad \rho_0(x) = 1, \qquad \rho_1(x) = \rho_0(x) + \Delta V(x),$$

$$x = \left[\begin{smallmatrix} x_1 \\ x_2 \end{smallmatrix}\right], \qquad \gamma = \frac{0.45}{4\pi^2}, \tag{71}$$

on the domain $\Omega = (0, 1)^2$ and the potential $V$ automatically solves the Poisson's equation with Neumann boundary condition (see (55)) by the above construction. Apart from verifying that $\boldsymbol{b} = \nabla V$ is a zero-variance dynamics numerically, we can also observe that Proposition 3.2 holds in this case; see Figure 6.

## H   Proof of Proposition 4.1

Below is the full version of Proposition 4.1.

**Proposition H.1** (Local minimum). *Assume that*

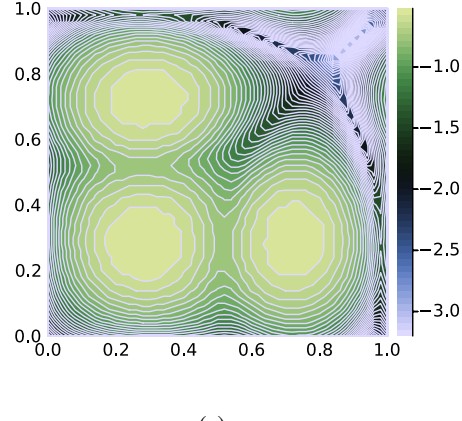

(a) $\varkappa$

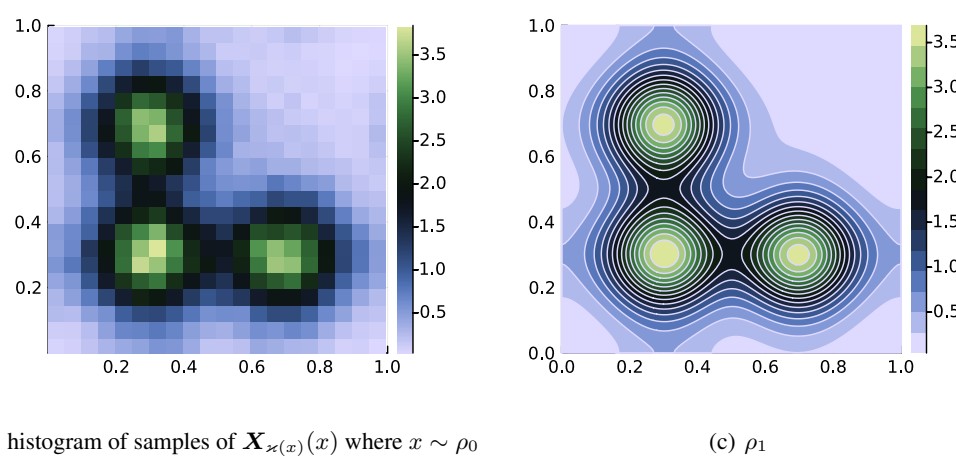

(b) histogram of samples of $\boldsymbol{X}_{\varkappa(x)}(x)$ where $x \sim \rho_0$           (c) $\rho_1$

Figure 5: The figure shows the time map $\varkappa$ defined via (14) and it also numerically verifies that $\rho_1$ is the distribution of $\boldsymbol{X}_{\varkappa(x)}(x)$ with $x \sim \rho_0$, for the model (54) on the torus $\Omega = [0, 1]^2$ (with periodic boundary condition).

(i) *(Local minimum).* $\boldsymbol{b} \in \mathfrak{B}_\infty$ *is a (non-trivial) local minimum of Var, namely, Var$(\boldsymbol{b}) <$ Var$^{(\max)}$ and if there is a perturbation $\delta\boldsymbol{b} \in C_c^\infty(\Omega, \mathbb{R}^d)$ such that $\boldsymbol{b} + \epsilon\delta\boldsymbol{b} \in \mathfrak{B}_\infty$ for sufficiently small $\epsilon$, then Var$(\boldsymbol{b} + \epsilon\delta\boldsymbol{b}) \geq$ Var$(\boldsymbol{b})$.*

(ii) *(Continuity assumption).* The functional derivative $\frac{\delta Var(\boldsymbol{b})}{\delta\boldsymbol{b}}$ is continuous and $\frac{\delta Var(\boldsymbol{b})}{\delta\boldsymbol{b}} = \boldsymbol{0}_d$ *on $\Omega$. In particular, $\nabla\mathcal{A}$ exists and is continuous on $\Omega$.*

(iii) *(Technical assumptions).* $\mho(\boldsymbol{b}) = \Omega$ *(see Definition A.4) and the set of $\boldsymbol{b}$-unstable points (see Definition A.8) has Lebesgue measure zero. Moreover, the set $\{x \in \Omega : \nabla(U_1 - U_0)(x) = \boldsymbol{0}_d\}$ has Lebesgue measure zero.*

*Then $\boldsymbol{b}$ is a zero-variance dynamics for the infinite-time NEIS scheme, i.e., Var$(\boldsymbol{b}) = 0$.*

Recall the formula of $\frac{\delta Var(\boldsymbol{b})}{\delta\boldsymbol{b}}$ from (38):

$$\frac{\delta \text{Var}(\boldsymbol{b})}{\delta\boldsymbol{b}}(x) = \frac{2\nabla\mathcal{A}(x)}{\mathcal{B}(x)} \left( \int_0^\infty \mathcal{F}_t^{(0)}(x)\,\mathrm{d}t \int_{-\infty}^0 \mathcal{F}_t^{(1)}(x)\,\mathrm{d}t - \int_{-\infty}^0 \mathcal{F}_t^{(0)}(x)\,\mathrm{d}t \int_0^\infty \mathcal{F}_t^{(1)}(x)\,\mathrm{d}t \right).$$

Here is a sketch of the main idea behind the proof. If $\nabla\mathcal{A} = \boldsymbol{0}_d$ on the domain $\Omega$, then $\mathcal{A}$ is a constant function. Hence, $\boldsymbol{b}$ provides a zero-variance estimator for the infinite-time NEIS scheme and it must be a global minimum of the functional Var as well. If the other term $\int_0^\infty \mathcal{F}_t^{(0)}(x)\,\mathrm{d}t \int_{-\infty}^0 \mathcal{F}_t^{(1)}(x)\,\mathrm{d}t -$

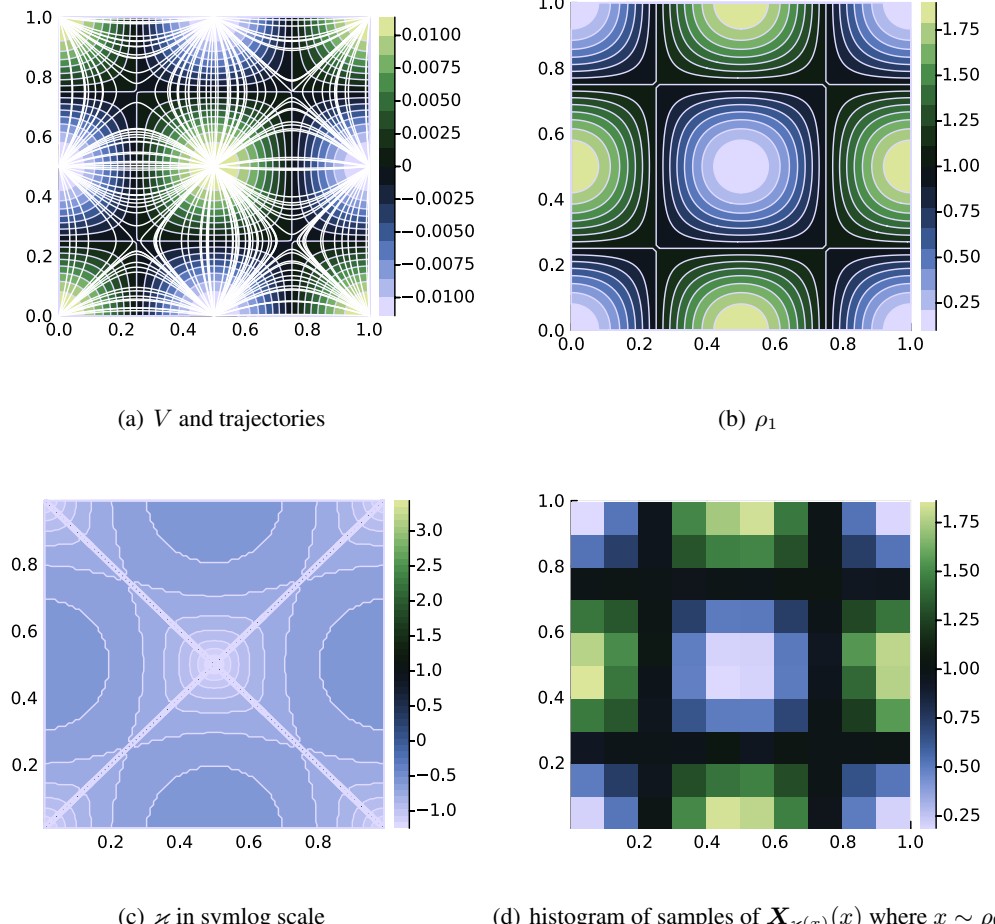

(a) $V$ and trajectories

(b) $\rho_1$

(c) $\varkappa$ in symlog scale

(d) histogram of samples of $\boldsymbol{X}_{\varkappa(x)}(x)$ where $x \sim \rho_0$

Figure 6: This figure visualizes the model (71) on the domain $\Omega = (0,1)^2$ (with Neumann boundary condition). In the panel (a), we show the potential $V$ and sample trajectories under the dynamics $\boldsymbol{b} = \nabla V$; in the panel (b), we show the distribution $\rho_1$. In the panel (c), we present the time map $\varphi \circ \varkappa$ where $\varkappa$ is defined via (14) and the rescaling function $\varphi(z) := \mathrm{sign}(z) \log(1 + |z|)$ is the symlog function. In the panel (d), we show a histogram of samples of $\boldsymbol{X}_{\varkappa(x)}(x)$ where $x \sim \rho_0$. The panel (d) resembles the panel (b), which numerically verifies that $\rho_1$ is the distribution of $\boldsymbol{X}_{\varkappa(x)}(x)$ with $x \sim \rho_0$.

$\int_{-\infty}^0 \mathcal{F}_t^{(0)}(x) \, \mathrm{d}t \int_0^\infty \mathcal{F}_t^{(1)}(x) \, \mathrm{d}t = 0$ locally on an open subset, then it could be shown that this is equivalent to $\langle \nabla(U_1 - U_0)(x), \boldsymbol{b}(x) \rangle = 0$ (see Lemma H.4 and Lemma H.5). Such a dynamics $\boldsymbol{b}$ should not be optimal, because $\boldsymbol{b}$ is perpendicular to the gradient of the potential difference and such a $\boldsymbol{b}$ does not explore the local structure (cf. Proposition H.3). This intuition leads into the following idea: if $\boldsymbol{b}$ is a local minimum of Var, then we should only have $\nabla \mathcal{A} = \boldsymbol{0}_d$ almost everywhere on $\Omega$; otherwise, we should be able to perturb $\boldsymbol{b}$ so that the dynamics $\boldsymbol{b}$ can better explore the landscapes of $U_0$ and $U_1$ and the variance can be further reduced; see Proposition H.6.

*Remark* H.2. As we work on the domain $\Omega$ only, we shall consider the topological space for the domain $\Omega$ instead of $\mathbb{R}^d$ from now on.

## H.1 A characterization of the global maximum

**Proposition H.3.** *If the dynamics $\boldsymbol{b} \in \mathfrak{B}_\infty$, then*

$$Var(\boldsymbol{b}) \leq Var^{(\max)}. \tag{72}$$

*The equality can be achieved iff*

$$\langle \boldsymbol{b}(x), \nabla(U_1 - U_0)(x) \rangle = 0, \qquad \forall x \in \Omega. \tag{73}$$

*Proof.* We only need to prove that $\mathcal{M}(\boldsymbol{b}) \leq \mathbb{E}_{\rho_0}[e^{-2(U_1 - U_0)}]$ and the equality is achieved iff (73) holds.

By Jensen's inequality,

$$\left( \frac{\int_{-\infty}^{\infty} \mathcal{F}_t^{(1)}(x) \, \mathrm{d}t}{\int_{-\infty}^{\infty} \mathcal{F}_t^{(0)}(x) \, \mathrm{d}t} \right)^2 = \left( \frac{\int_{-\infty}^{\infty} e^{-U_1\left(\boldsymbol{X}_t(x)\right) + U_0\left(\boldsymbol{X}_t(x)\right)} \mathcal{F}_t^{(0)}(x) \, \mathrm{d}t}{\int_{-\infty}^{\infty} \mathcal{F}_t^{(0)}(x) \, \mathrm{d}t} \right)^2$$

$$\leq \frac{\int_{-\infty}^{\infty} e^{-2(U_1 - U_0)\left(\boldsymbol{X}_t(x)\right)} \mathcal{F}_t^{(0)}(x) \, \mathrm{d}t}{\int_{-\infty}^{\infty} \mathcal{F}_t^{(0)}(x) \, \mathrm{d}t}.$$

By taking the expectation $\mathbb{E}_{x \sim \rho_0}[\,\cdot\,]$ for both sides and by (10) for new potentials $\widetilde{U}_1 = 2U_1 - U_0$ and $\widetilde{U}_0 = U_0$, we immediately have the inequality:

$$\mathcal{M}(\boldsymbol{b}) = \mathbb{E}_{x \sim \rho_0} \left[ \left( \frac{\int_{-\infty}^{\infty} \mathcal{F}_t^{(1)}(x) \, \mathrm{d}t}{\int_{-\infty}^{\infty} \mathcal{F}_t^{(0)}(x) \, \mathrm{d}t} \right)^2 \right]$$

$$\leq \mathbb{E}_{x \sim \rho_0} \left[ \frac{\int_{-\infty}^{\infty} e^{-2(U_1 - U_0)\left(\boldsymbol{X}_t(x)\right)} \mathcal{F}_t^{(0)}(x) \, \mathrm{d}t}{\int_{-\infty}^{\infty} \mathcal{F}_t^{(0)}(x) \, \mathrm{d}t} \right]$$

$$\overset{(10)}{=} \frac{\int_\Omega e^{-2U_1 + U_0}}{\int_\Omega e^{-U_0}} = \mathbb{E}_{x \sim \rho_0} \left[ e^{-2(U_1 - U_0)} \right].$$

This is essentially the inequality (72).

Next in order to achieve the equality in (72), we need the equality to hold in the above Jensen's inequality:

$$t \mapsto e^{-\left(U_1 - U_0\right)\left(\boldsymbol{X}_t(x)\right)} \text{ is a constant function.}$$

By taking derivative with respect to $t$, we immediately know that

$$\langle \nabla(U_1 - U_0)(x), \boldsymbol{b}(x) \rangle = 0, \qquad \rho_0\text{-almost surely.}$$

By the continuity assumption on $U_1, U_0$ and $\boldsymbol{b}$, we obtain (73).

Conversely, when (73) holds,

$$\frac{\mathrm{d}}{\mathrm{d}t}(U_1 - U_0)\left(\boldsymbol{X}_t(x)\right) = \langle \nabla(U_1 - U_0)(\boldsymbol{X}_t(x)), \boldsymbol{b}\left(\boldsymbol{X}_t(x)\right) \rangle = 0.$$

Hence, $U_1\left(\boldsymbol{X}_t(x)\right) - U_0\left(\boldsymbol{X}_t(x)\right) = U_1(x) - U_0(x)$ for any $t \in \mathbb{R}$. Then

$$\frac{\int_{-\infty}^{\infty} \mathcal{F}_t^{(1)}(x) \, \mathrm{d}t}{\int_{-\infty}^{\infty} \mathcal{F}_t^{(0)}(x) \, \mathrm{d}t} = e^{-U_1(x) + U_0(x)} \frac{\int_{-\infty}^{\infty} e^{-U_0(\boldsymbol{X}_t(x))} \mathcal{J}_t(x) \, \mathrm{d}t}{\int_{-\infty}^{\infty} e^{-U_0(\boldsymbol{X}_t(x))} \mathcal{J}_t(x) \, \mathrm{d}t} = e^{-U_1(x) + U_0(x)}.$$

Hence, the equality in (72) holds under the condition (73). $\qquad\square$

## H.2 Some observations about the functional derivative

We need some simplified understanding of the condition $\int_0^{\infty} \mathcal{F}_t^{(0)}(x) \, \mathrm{d}t \int_{-\infty}^{0} \mathcal{F}_t^{(1)}(x) \, \mathrm{d}t = \int_{-\infty}^{0} \mathcal{F}_t^{(0)}(x) \, \mathrm{d}t \int_0^{\infty} \mathcal{F}_t^{(1)}(x) \, \mathrm{d}t$ arising from $\frac{\delta \mathrm{Var}(\boldsymbol{b})}{\delta \boldsymbol{b}} = \boldsymbol{0}_d$, which are presented in the following two lemmas.

**Lemma H.4.** $\int_0^{\infty} \mathcal{F}_t^{(0)}(x) \, \mathrm{d}t \int_{-\infty}^{0} \mathcal{F}_t^{(1)}(x) \, \mathrm{d}t = \int_{-\infty}^{0} \mathcal{F}_t^{(0)}(x) \, \mathrm{d}t \int_0^{\infty} \mathcal{F}_t^{(1)}(x) \, \mathrm{d}t$ *is equivalent to*

$$\frac{\int_0^{\infty} \mathcal{F}_t^{(1)}(x) \, \mathrm{d}t}{\int_0^{\infty} \mathcal{F}_t^{(0)}(x) \, \mathrm{d}t} = \mathcal{A}(x). \tag{74}$$

*Proof.*

$$\int_0^\infty \mathcal{F}_t^{(0)}(x)\,\mathrm{d}t \int_{-\infty}^0 \mathcal{F}_t^{(1)}(x)\,\mathrm{d}t = \int_{-\infty}^0 \mathcal{F}_t^{(0)}(x)\,\mathrm{d}t \int_0^\infty \mathcal{F}_t^{(1)}(x)\,\mathrm{d}t$$

$$\iff \frac{\int_{-\infty}^0 \mathcal{F}_t^{(0)}(x)\,\mathrm{d}t}{\int_0^\infty \mathcal{F}_t^{(0)}(x)\,\mathrm{d}t} = \frac{\int_{-\infty}^0 \mathcal{F}_t^{(1)}(x)\,\mathrm{d}t}{\int_0^\infty \mathcal{F}_t^{(1)}(x)\,\mathrm{d}t} \qquad \text{(add 1 to both sides)}$$

$$\iff \frac{\int_{-\infty}^\infty \mathcal{F}_t^{(0)}(x)\,\mathrm{d}t}{\int_0^\infty \mathcal{F}_t^{(0)}(x)\,\mathrm{d}t} = \frac{\int_{-\infty}^\infty \mathcal{F}_t^{(1)}(x)\,\mathrm{d}t}{\int_0^\infty \mathcal{F}_t^{(1)}(x)\,\mathrm{d}t}$$

$$\iff \frac{\int_0^\infty \mathcal{F}_t^{(1)}(x)\,\mathrm{d}t}{\int_0^\infty \mathcal{F}_t^{(0)}(x)\,\mathrm{d}t} = \mathcal{A}(x).$$

$\square$

The following result provides a simplified characterization of the equality (74).

**Lemma H.5.**

    *(i) Suppose the condition in (74) holds for any $x$ in an open set $D$. Then*

$$\langle \nabla(U_1 - U_0)(x), \boldsymbol{b}(x) \rangle = 0, \qquad \forall x \in D. \tag{75}$$

    *(ii) Conversely, if (75) holds for $D = \Omega$, then $\boldsymbol{b}$ is a global maximum of $\mathcal{M}$ (as well as Var).*

*Proof.* Part (ii) immediately follows from Proposition H.3. Next we prove part (i). From previous results, the condition is that

$$\frac{\int_0^\infty \mathcal{F}_t^{(1)}(x)\,\mathrm{d}t}{\int_0^\infty \mathcal{F}_t^{(0)}(x)\,\mathrm{d}t} = \mathcal{A}(x), \qquad \forall x \in D.$$

After replacing $x$ by $\boldsymbol{X}_s(x) \in D$ in the above equation and by (25), we have

$$\mathcal{A}(x) \overset{(26)}{=} \mathcal{A}\big(\boldsymbol{X}_s(x)\big) = \frac{\int_0^\infty \mathcal{F}_t^{(1)}\big(\boldsymbol{X}_s(x)\big)\,\mathrm{d}t}{\int_0^\infty \mathcal{F}_t^{(0)}\big(\boldsymbol{X}_s(x)\big)\,\mathrm{d}t} = \frac{\int_s^\infty \mathcal{F}_t^{(1)}(x)\,\mathrm{d}t}{\int_s^\infty \mathcal{F}_t^{(0)}(x)\,\mathrm{d}t}, \ \forall x \in D, \ s \in \big(\tau_D^-(x), \tau_D^+(x)\big),$$

where $\tau_D^+(x)$ and $\tau_D^-(x)$ defined in (23) are hitting times for the forward and backward branches to the boundary of $D$. Note that the right hand side of the last equation depends on $s$, whereas the left hand side does not. Let us take the derivative with respect to $s$ and with straightforward simplifications, we obtain

$$\mathcal{F}_s^{(1)}(x)/\mathcal{F}_s^{(0)}(x) = \mathcal{A}(x), \qquad \forall x \in D, \ s \in \big(\tau_D^-(x), \tau_D^+(x)\big).$$

By (6), we know $\exp\Big(U_0\big(\boldsymbol{X}_s(x)\big) - U_1\big(\boldsymbol{X}_s(x)\big)\Big) = \mathcal{A}(x)$. Again, the left hand side depends on $s$ whereas the right hand side does not. So we take the derivative with respect to $s$ again and obtain $\big\langle \nabla(U_1 - U_0)\big(\boldsymbol{X}_s(x)\big), \boldsymbol{b}\big(\boldsymbol{X}_s(x)\big)\big\rangle = 0$ for any $x \in D$ and $s \in \big(\tau_D^-(x), \tau_D^+(x)\big)$. Then (75) follows immediately by choosing $s = 0$. $\square$

## H.3    A weaker version

Lemma H.5 leads into the following intuition: if there is a certain open subset $D$ on which $\langle \nabla(U_1 - U_0), \boldsymbol{b} \rangle = 0$, then such a dynamics $\boldsymbol{b}$ should not be a local minimum of Var, because such a $\boldsymbol{b}$ cannot explored the landscape structure of $U_1$ on $D$. This intuition is more rigorously formulated in the following proposition.

**Proposition H.6.** *Consider a dynamics $\boldsymbol{b} \in \mathfrak{B}_\infty$. Suppose $D \subset \mho(\boldsymbol{b})$ is nonempty and open. Let $K := \Omega \backslash D$. Assume that*

(i) *For any $x \in D$, we have $\langle \boldsymbol{b}(x), \nabla(U_1 - U_0)(x) \rangle = 0$.*

(ii) *For any $t \in \mathbb{R}$ and $x \in D$, we have $\boldsymbol{X}_t(x) \notin K^\circ$, i.e., trajectories from $D$ are confined inside $\overline{D}$.*

(iii) *There exists a $\boldsymbol{b}$-stable point $x^\star \in D$ such that $\nabla(U_1 - U_0)(x^\star) \neq \boldsymbol{0}_d$.*

*Then such a dynamics $\boldsymbol{b}$ must not be a local minimum of the second moment $\mathcal{M}$ (as well as the variance).*

*Proof.* We proceed in two steps.

**Step (I):** The first goal is to find a smooth function $\delta \boldsymbol{b} \in C_c^\infty(D, \mathbb{R}^d)$ and $\epsilon_0 > 0$ such that

$$\rho_0(E) > 0 \text{ where } E := \left\{ x \in D \mid \langle \delta \boldsymbol{b}(x), \nabla(U_1 - U_0)(x) \rangle \neq 0 \right\} \subset \text{supp}(\delta \boldsymbol{b}); \quad (76a)$$

$$\boldsymbol{b} + \epsilon \delta \boldsymbol{b} \in \mathfrak{B}_\infty, \qquad \forall \epsilon \in (0, \epsilon_0); \quad (76b)$$

$$\text{dist}(E, \partial D) > 0. \quad (76c)$$

By the assumption (iii) of this proposition and Proposition A.10, we know there is an open ball $B_\lambda(x^\star) \subset D$ such that for any $\delta \boldsymbol{b} \in C_c^\infty(B_\lambda(x^\star), \mathbb{R}^d)$, $\boldsymbol{b} + \epsilon \delta \boldsymbol{b} \in \mathfrak{B}_\infty$ for small enough $\epsilon$ and thus (76b) is satisfied. It is clear that we can easily choose $\lambda$ small enough so that (76c) holds.

Next, the task is to find a smooth function $\delta \boldsymbol{b}$ supported on $B_\lambda(x^\star)$ such that (76a) holds. Since $\nabla(U_1 - U_0)(x^\star) \neq \boldsymbol{0}_d$ and $U_0, U_1$ are assumed to be smooth, we can choose $\lambda$ small enough such that

$$|\nabla(U_1 - U_0)(x) - \nabla(U_1 - U_0)(x^\star)| \leq \frac{1}{4}|\nabla(U_1 - U_0)(x^\star)|, \qquad \forall x \in B_\lambda(x^\star).$$

It is well-known that

$$\varphi(x) := \begin{cases} e^{-\frac{1}{1 - |x|^2}}, & \text{if } |x| < 1; \\ 0, & \text{if } |x| \geq 1, \end{cases}$$

is a smooth function compactly supported on $B_1(0)$. Then let us consider

$$\delta \boldsymbol{b}_1(x) := \varphi\left(\frac{x - x^\star}{\lambda_1}\right) \nabla(U_1 - U_0)(x), \qquad \text{where } \lambda_1 \in (0, \lambda).$$

It is clear that $\delta \boldsymbol{b}_1$ is compactly supported on $B_{\lambda_1}(x^\star) \subset B_\lambda(x^\star)$ and for any $x \in B_{\lambda_1}(x^\star)$, we have $\langle \delta \boldsymbol{b}_1(x), \nabla(U_1 - U_0)(x) \rangle > 0$ so that (76a) clearly holds. Next, we still need to further smooth out $\delta \boldsymbol{b}_1$ (see e.g., Appendix C of [11]) by introducing

$$\delta \boldsymbol{b}_2(x) := \int_{\mathbb{R}^d} \varphi_\varepsilon(x - y) \delta \boldsymbol{b}_1(y) \, \mathrm{d}y,$$

where $\varphi_\varepsilon(x) := \frac{1}{\varepsilon^d} \varphi(x/\varepsilon)$. Note that we can easily extend $\delta \boldsymbol{b}_1$ to $\mathbb{R}^d$ by letting $\delta \boldsymbol{b}_1 = \boldsymbol{0}_d$ outside of $B_{\lambda_1}(x^\star)$ so that $\delta \boldsymbol{b}_1$ can be well-defined on $\mathbb{R}^d$. By choosing $0 < \varepsilon < \lambda - \lambda_1$, we can ensure that the smooth function $\delta \boldsymbol{b}_2$ is compactly supported on $B_\lambda(x^\star)$. It is also not hard to show that (76a) still holds for $\delta \boldsymbol{b}_2$: for any $x \in B_{\lambda_1}(x^\star)$,

$$\langle \delta \boldsymbol{b}_2(x), \nabla(U_1 - U_0)(x) \rangle = \int_{\mathbb{R}^d} \varphi_\varepsilon(x - y) \langle \delta \boldsymbol{b}_1(y), \nabla(U_1 - U_0)(x) \rangle \, \mathrm{d}y$$

$$= \int_{\mathbb{R}^d} \varphi_\varepsilon(x - y) \varphi\left(\frac{y - x^\star}{\lambda_1}\right) \langle \nabla(U_1 - U_0)(y), \nabla(U_1 - U_0)(x) \rangle \, \mathrm{d}y$$

$$\geq \frac{7}{16} |\nabla(U_1 - U_0)(x^\star)|^2 \int_{\mathbb{R}^d} \varphi_\varepsilon(x - y) \varphi\left(\frac{y - x^\star}{\lambda_1}\right) \, \mathrm{d}y > 0.$$

In summary, $\delta \boldsymbol{b}_2$ constructed above satisfies all requirements.

**Step (II):** We prove that $\boldsymbol{b}$ is not a local minimum by showing that $\mathcal{M}(\boldsymbol{b} + \epsilon \delta \boldsymbol{b}) < \mathcal{M}(\boldsymbol{b})$ for any $\epsilon \in (0, \epsilon_0)$, where $\delta \boldsymbol{b}$ satisfies all conditions in Step (I).

By the construction of $\delta \boldsymbol{b}$, we know $\boldsymbol{b}^\epsilon := \boldsymbol{b} + \epsilon \delta \boldsymbol{b}$ does not change the velocity field at $\partial D$, and therefore, $\boldsymbol{X}_t^\epsilon(x) \notin K^\circ$ for any $x \in D$ still holds for the dynamics $\boldsymbol{b}^\epsilon$ i.e., trajectories from $D$ do not enter $K^\circ$. As an immediately consequence, trajectories $t \mapsto \boldsymbol{X}_t^\epsilon(x)$ with $x \in K^\circ$ will not enter $D$ (as ODE trajectories are reversible). The slightly technical part is to consider trajectories $t \mapsto \boldsymbol{X}_t^\epsilon(x)$ with $x \in \partial D \equiv \partial K$, where the trajectory $t \mapsto \boldsymbol{X}_t^\epsilon(x)$ evolves under the dynamics $\boldsymbol{b}^\epsilon$. Let us consider two disjoint sets:

$$\widetilde{D} := \left\{ x \in D \cup \partial D \equiv \Omega \backslash K^\circ \mid \boldsymbol{X}_t^\epsilon(x) \notin K^\circ, \ \forall t \in \mathbb{R} \right\} \supset D, \qquad \widetilde{K} := \Omega \backslash \widetilde{D} \subset K.$$

By such a construction, we can observe that for any trajectory $t \mapsto \boldsymbol{X}_t^\epsilon(x)$ with $x \in \widetilde{K}$ (or $x \in \widetilde{D}$), it must remain inside $\widetilde{K}$ (or $\widetilde{D}$). In other words, the flows within $\widetilde{D}$ and $\widetilde{K}$ are completely separated from each other. Moreover, because $\delta \boldsymbol{b}$ is only supported on $E \subset D$ which is completely inside $D$ by (76c), we know the above definitions of these two sets $\widetilde{D}$ and $\widetilde{K}$ are independent of $\epsilon \in [0, \epsilon_0)$.

Let us use $\mathcal{A}$ to denote the function defined in (24) corresponding to the dynamics $\boldsymbol{b}$ and use $\mathcal{A}_\epsilon$ to denote the one corresponding to the perturbed dynamics $\boldsymbol{b}^\epsilon$. Recall the assumption (i) that $\langle \boldsymbol{b}(x), \nabla(U_1 - U_0)(x) \rangle = 0$ for all $x \in D$. By the fact that $\partial D = \partial K$, and by the continuity of $\boldsymbol{b}$, $\nabla U_0$, and $\nabla U_1$, we know

$$\langle \boldsymbol{b}(x), \nabla(U_1 - U_0)(x) \rangle = 0, \qquad \forall x \in \partial D \cup D \equiv \Omega \backslash K^\circ.$$

For any trajectory $t \mapsto \boldsymbol{X}_t(x)$ with $x \in \widetilde{D}$, we can easily show that $e^{-(U_1 - U_0)\left( \boldsymbol{X}_t(x) \right)} = e^{-(U_1 - U_0)(x)}$ for all $t \in \mathbb{R}$, and thus we have $\mathcal{A}(x) = e^{-(U_1 - U_0)(x)}$ for any $x \in \widetilde{D}$ (the same calculation, in fact, has been shown in the proof of Proposition H.3). Hence,

$$\mathcal{M}(\boldsymbol{b}) = \mathbb{E}_{\rho_0} \left[ \chi_{\widetilde{K}}(\cdot) \big( \mathcal{A}(\cdot) \big)^2 \right] + \mathbb{E}_{\rho_0} \left[ \chi_{\widetilde{D}}(\cdot) \big( \mathcal{A}(\cdot) \big)^2 \right]$$

$$= \mathbb{E}_{\rho_0} \left[ \chi_{\widetilde{K}}(\cdot) \big( \mathcal{A}(\cdot) \big)^2 \right] + \mathbb{E}_{\rho_0} \left[ \chi_{\widetilde{D}}(\cdot) e^{-2(U_1 - U_0)(\cdot)} \right],$$

where $\chi_A(\cdot)$ is an indicator function for a set $A$.

Next we consider the trajectory $t \mapsto \boldsymbol{X}_t^\epsilon(x)$ with $x \in \widetilde{K}$. Since such a trajectory never enters $\widetilde{D} \supset D$ and $\boldsymbol{b}^\epsilon = \boldsymbol{b}$ on $\Omega \backslash D$, we know $\boldsymbol{X}_t^\epsilon(x) = \boldsymbol{X}_t(x)$ for any $t \in \mathbb{R}$ and $x \in \widetilde{K}$, and thus $\mathcal{A} = \mathcal{A}_\epsilon$ on $\widetilde{K}$ for any $\epsilon \in [0, \epsilon_0)$. Hence, $\mathbb{E}_{\rho_0} \left[ \chi_{\widetilde{K}}(\cdot) \big( \mathcal{A}(\cdot) \big)^2 \right] = \mathbb{E}_{\rho_0} \left[ \chi_{\widetilde{K}}(\cdot) \big( \mathcal{A}_\epsilon(\cdot) \big)^2 \right]$. By the same argument as in Proposition H.3 (by treating $\widetilde{D}$ as the domain),

$$\mathcal{M}(\boldsymbol{b}^\epsilon) - \mathcal{M}(\boldsymbol{b}) = \mathbb{E}_{\rho_0} \left[ \chi_{\widetilde{D}}(\cdot) \big( \mathcal{A}_\epsilon(\cdot) \big)^2 \right] - \mathbb{E}_{\rho_0} \left[ \chi_{\widetilde{D}}(\cdot) e^{-2(U_1 - U_0)(\cdot)} \right] \leq 0,$$

where the equality is achieved only if $\langle \boldsymbol{b}^\epsilon, \nabla(U_1 - U_0) \rangle = 0$ on $\widetilde{D}$. Note that on $\widetilde{D}$,

$$\langle \boldsymbol{b}^\epsilon, \nabla(U_1 - U_0) \rangle = \langle \boldsymbol{b} + \epsilon \delta \boldsymbol{b}, \nabla(U_1 - U_0) \rangle = \epsilon \langle \delta \boldsymbol{b}, \nabla(U_1 - U_0) \rangle.$$

However, due to the fact that $\langle \delta \boldsymbol{b}, \nabla(U_1 - U_0) \rangle \neq 0$ for some open subset of $D$ with strictly positive $\rho_0$-measure (as constructed in Step (I)), the equality $\mathcal{M}(\boldsymbol{b}^\epsilon) = \mathcal{M}(\boldsymbol{b})$ cannot be achieved and thus

$$\mathcal{M}(\boldsymbol{b}^\epsilon) < \mathcal{M}(\boldsymbol{b}).$$

Since we can find a local perturbation $\delta \boldsymbol{b}$ such that $\mathcal{M}(\boldsymbol{b}^\epsilon) < \mathcal{M}(\boldsymbol{b})$ for any $\epsilon \in (0, \epsilon_0)$, then $\boldsymbol{b}$ must not be a local minimum of $\mathcal{M}$. $\qquad \square$

## H.4  Proof of Proposition H.1

By Proposition D.1, we know either

$$\nabla \mathcal{A}(x) = \boldsymbol{0}_d, \ \text{or} \ \int_0^\infty \mathcal{F}_t^{(0)}(x) \, \mathrm{d}t \int_{-\infty}^0 \mathcal{F}_t^{(1)}(x) \, \mathrm{d}t - \int_{-\infty}^0 \mathcal{F}_t^{(0)}(x) \, \mathrm{d}t \int_0^\infty \mathcal{F}_t^{(1)}(x) \, \mathrm{d}t = 0.$$

Define

$$K := \left\{ x \in \Omega : \ \nabla \mathcal{A}(x) = \boldsymbol{0}_d \right\},$$

which is a closed subset of $\Omega$ by the continuity assumption on $\nabla \mathcal{A}$.

Hence, $D := \Omega \backslash K$ is open and by Lemma H.5 (i), we know $\langle \boldsymbol{b}, \nabla(U_1 - U_0) \rangle = 0$ on $D$. Here are a few cases to discuss.

*Case (I)*: $K^\circ = \emptyset$.

If $K^\circ = \emptyset$, then we claim that $\boldsymbol{b}$ must be a global maximum and this contradicts with the assumption that $\text{Var}(\boldsymbol{b}) < \text{Var}^{(\text{max})}$. It is not hard to observe that $K^\circ = \emptyset$ implies that $\overline{D} = \Omega$. By continuity, we know $\langle \boldsymbol{b}, \nabla(U_1 - U_0) \rangle = 0$ on $\overline{D} = \Omega$. Then by Lemma H.5 part (ii), $\boldsymbol{b}$ must be a global maximum.

*Case (II)*: $D = \emptyset$.

If $D = \emptyset$ (i.e., $K = \Omega$), then it is clear that $\mathcal{A}$ is a constant function, and thus the variance $\text{Var}(\boldsymbol{b}) = 0$. This means $\boldsymbol{b}$ is a zero-variance dynamics.

*Case (III)*: $K^\circ \neq \emptyset$ and $D \neq \emptyset$

In order to use Proposition H.6, we need to deal with the case that some trajectories $t \mapsto \boldsymbol{X}_t(x)$ for $x \in D$ might enter $K^\circ$. Let us introduce

$$ S = \left\{ x \in D : \ \{\boldsymbol{X}_t(x)\}_{t \in \mathbb{R}} \cap K^\circ \neq \emptyset \right\}, \tag{77} $$

which essentially contains all points in $D$ whose trajectories enter $K^\circ$ at some time. We can easily show that $S$ is open: because $\boldsymbol{b}$ is assumed to be smooth, the trajectories are continuous under a small perturbation for initial states. Then the new disjoint sub-regions to consider are $\widetilde{K} := \overline{K^\circ \cup S}$ and $\widetilde{D} := \Omega \backslash \widetilde{K}$. We collect some facts for clarity:

- $\widetilde{K}^\circ = K^\circ \cup S \neq \emptyset$;

- $\widetilde{D} \subset D$, which immediately implies that $\langle \boldsymbol{b}, \nabla(U_1 - U_0) \rangle = 0$ on $\widetilde{D}$.

- If $x \in \widetilde{D}$, then $\boldsymbol{X}_t(x)$ must not enter $\widetilde{K}^\circ$. Indeed, if not, then we have either $t \mapsto \boldsymbol{X}_t(x)$ entering $K^\circ$ (which contradicts with $x \notin S$), or $\boldsymbol{X}_t(x)$ entering $S$ (which still means $\boldsymbol{X}_t(x)$ will enter $K^\circ$ due to the reversibility of deterministic trajectories).

We need to discuss two cases:

(a) Firstly, let us consider $\widetilde{D} = \emptyset$, i.e., $\widetilde{K} = \Omega$ and thus $\widetilde{K}^\circ = \Omega$ by the assumption that $\Omega$ is open in the topology of the space $\mathbb{R}^d$. The connectivity granted by the definition of $S$ in (77) implies that we can divide $\widetilde{K}^\circ$ into a countable number of sub-regions on which the function $\mathcal{A}$ is a constant. More specifically, for $x \in \Omega$, define

$$ R_x := \left\{ y \in \Omega : \ \exists \gamma. \in C([0,1], \mathbb{R}^d), \gamma_0 = x, \gamma_1 = y, \ \mathcal{A}(\gamma_t) = \mathcal{A}(x), \ \forall t \in [0,1] \right\}. $$

Obviously, $x \in R_x$. By the invariance of $\mathcal{A}$ under the dynamical flow (see (26)) and the definition of $S$ (77), we have $\cup_{x \in K^\circ} R_x \supset \widetilde{K}^\circ = \Omega$, which implies that $\cup_{x \in K^\circ} R_x = \Omega$.

- Suppose $x, y \in K^\circ$ and $R_x \cap R_y \neq \emptyset$, then there exists a $z \in R_x \cap R_y$ such that $z$ connects to both $x$ and $y$ via a continuous path and thus $\mathcal{A}(x) = \mathcal{A}(z) = \mathcal{A}(y)$. It is then clear that $R_x = R_y$ via treating $z$ as a bridge. Therefore, $\cup_{x \in K^\circ} R_x$ can be simplified as $\cup_{x \in E} R_x$ where $\{R_x\}_{x \in E}$ are disjoint and $E \subset K^\circ$.
- Next, we can show that $E$ is countable. Since $K^\circ$ is open and $\nabla \mathcal{A}(x) = \boldsymbol{0}_d$ on $K$, we can easily verify that for each $x \in K^\circ$, there exists a local neighbor $B_\delta(x) \subset R_x$ and due to the fact that $\mathbb{Q}^d$ is dense and countable, $E$ is at most countable.

To summarize, we have

$$ \bigcup_{x \in E} R_x = \Omega, $$

where $\{R_x\}_{x \in E}$ are disjoint and $E$ is countable.

As $\mathcal{A}$ is a constant function on $R_x$ by the definition of $R_{(\cdot)}$, $\mathcal{A}$ is a step function on $\Omega$. By the continuity of $\mathcal{A}$ from the assumption and $\Omega$ is a connected open domain, we readily know $\mathcal{A}$ must be a constant function on $\Omega$ instead, and such a $\boldsymbol{b}$ must be a zero-variance dynamics.

(b) Next, let us consider the case $\widetilde{D} \neq \emptyset$. By the assumption that $\nabla(U_1 - U_0) = \mathbf{0}_d$ only on a set with Lebesgue measure zero, we know there must exist $y \in \widetilde{D}$ such that $\nabla(U_1 - U_0)(y) \neq \mathbf{0}_d$. By the assumption that $\mathbf{b}$-unstable points has Lebesgue measure zero, there must exist a $\mathbf{b}$-stable point $x^\star$ (around $y$) such that $\nabla(U_1 - U_1)(x^\star) \neq \mathbf{0}_d$. Then Proposition H.6 tells us that $\mathbf{b}$ must not be a local minimum, which contradicts with the assumption.

# I  Supplementary material for numerical experiments

## I.1  Details about AIS.

For AIS method, we use the equally-spaced temperature distribution, i.e., $\pi_k \propto \rho_0^{1-\beta_k} \rho_1^{\beta_k}$, where $\beta_k = k/K$ for $0 \leq k \leq K$; for each transition step, we use Metropolis-adjusted Langevin algorithm with time step $0.1$ to generate the chain.

More specifically, suppose $M_j(\cdot, \cdot)$ is a transition kernel which leaves $\pi_j$ invariant, then

$$\mathcal{Z}_1 = \mathbb{E}\left[ e^{-\sum_{j=1}^{K}(\beta_j - \beta_{j-1})\left(U_1(x_{j-1}) - U_0(x_{j-1})\right)} \right], \tag{78}$$

where $x_j \sim M_j(x_{j-1}, \cdot)$ and $x_0 \sim \rho_0$ [26, 2]. To implement such a transition kernel, we use Metropolis-adjusted Langevin algorithm: suppose $\tau$ (which is chosen as $0.1$ as a prescribed parameter) is the time step, then let $\widetilde{x}_j := x_{j-1} + \tau \nabla \log \pi_j(x_{j-1}) + \sqrt{2\tau}\xi_j$, where $\xi_j$ are *i.i.d.* $d$-dimensional standard normal random variables; the state $\widetilde{x}_j$ is accepted with a rate

$$\min\left\{1, \frac{\pi_j(\widetilde{x}_j)q(x_{j-1} \mid \widetilde{x}_j)}{\pi_j(x_{j-1})q(\widetilde{x}_j \mid x_{j-1})}\right\},$$

where $q(x' \mid x) = \exp\left(-\frac{1}{4\tau}|x' - x - \tau \nabla \log \pi_j(x)|^2\right)$ coming from the transition probability for the Langevin step.

For each $j$, inside the Metropolis-Hasting correction term, we need 2 queries to $\nabla U_1$ (i.e., $\nabla U_1(x_{j-1})$ and $\nabla U_1(\widetilde{x}_j)$ hidden inside the computation of $\nabla \log \pi_j$ for the acceptance rate), and we need two queries to $U_1$ when computing $U_1(x_{j-1})$ (inside (78) and $\pi_j(x_{j-1})$) and $U_1(\widetilde{x}_j)$ (inside $\pi_j(\widetilde{x}_j)$). However, since the queries to $\nabla U_1(x_{j-1})$ and $U_1(x_{j-1})$ can be borrowed from the step $j-1$ (if one saves these information), the total number of queries for each AIS trajectory is, in fact, $K+1$ for both $\nabla U_1$ and $U_1$.

## I.2  More implementation details

**Neural network architecture.**  We use the following $\ell$-layer neural network [10, 31] to parameterize the dynamics $\mathbf{b} : \Omega \to \mathbb{R}^d$ during training:

$$x \mapsto W_\ell\left(f_{\ell-1} \circ \cdots f_2 \circ f_1(x)\right) + b_\ell,$$

where $\ell$ is the layer depth (one output layer and $\ell - 1$ hidden layers), $f_j(\cdot) = \eta(W_j(\cdot) + b_j)$ for $j = 1, 2, \cdots, \ell - 1$, $\eta$ is the activation function, $W_j \in \mathbb{R}^{n_j} \times \mathbb{R}^{n_{j+1}}$, $b_j \in \mathbb{R}^{n_{j+1}}$ for $j = 1, 2, \cdots, \ell$. When we choose $n_j = m$ for all $2 \leq j \leq \ell$, we refer such a neural network as $(\ell, m)$-architecture which is mentioned in Section 5.

More specifically, let us take $\ell = 2$ as an example: an $(\ell, m) = (2, m)$ architecture for a generic ansatz for $\mathbf{b} : \Omega \to \mathbb{R}^d$ refers to the following parameterization

$$\mathbf{b}(x) = W_2 \eta(W_1 x + b_1) + b_2, \qquad \text{(generic ansatz)},$$

where weights $W_1 \in \mathbb{R}^d \times \mathbb{R}^m$, $W_2 \in \mathbb{R}^m \times \mathbb{R}^d$ and bias vectors $b_1 \in \mathbb{R}^m$, $b_2 \in \mathbb{R}^d$.

For the gradient-form ansatz, as we essentially need to parameterize a potential $V : \Omega \to \mathbb{R}$, the $(\ell, m)$-architecture for $V$ refers to the following choice when $\ell = 2$,

$$V(x) = W_2 \eta(W_1 x + b_1)$$

where $W_2 \in \mathbb{R}^m \times \mathbb{R}$, $W_1 \in \mathbb{R}^d \times \mathbb{R}^m$, $b_1 \in \mathbb{R}^m$. The bias vector in the output layer is chosen as zero (i.e., $b_2 = 0$) because $b_2$ is a redundant parameter after taking the gradient. The dynamics $\mathbf{b}$ in the gradient form refers to $\mathbf{b} = \nabla V$.

**Initialization and trial repetition.** When we parameterize the flow via neural networks, weights and biases in the neural network are randomly generated. Therefore, we consider two (or three) independent trials (associated with different random initializations of $\boldsymbol{b}$) for the same neural network architecture characterized by a pair $(\ell, m)$.

**Optimization algorithm.** We use the SGD algorithm to optimize parameters for $\boldsymbol{b}$. The Armijo line search algorithm (see e.g., [3, 44]) can used to find the learning rate; in practice, we notice that solving $\dot{\vartheta} = -\frac{\nabla_\vartheta \mathcal{M}_{t_-,t_+}(\boldsymbol{b}_\vartheta)}{|\nabla_\vartheta \mathcal{M}_{t_-,t_+}(\boldsymbol{b}_\vartheta)|}$ with a relatively large learning rate also works well for numerical examples considered in Section 6, and it is used for training in Section 6; $\vartheta$ are trainable parameters and $\boldsymbol{b}_\vartheta$ is the flow parameterized by $\vartheta$. The way to approximate the loss function $\mathcal{M}_{t_-,t_+}(\boldsymbol{b}) \equiv \mathbb{E}_0\big[|\mathcal{A}_{t_-,t_+}|^2\big]$ (see Section 4) and in particular its gradient with respect to parameters in $\boldsymbol{b}$ will be explained in Appendix I.3 and I.4.

### I.3 An integration-based forward propagation method to compute the estimator and its gradient

To estimate $\mathcal{A}_{t_-,t_+}(x)$ in (8) with $t_+ = t_- + 1$ and $t_- \in [-1, 0]$, the most straightforward approach is to compute $\mathcal{F}_t^{(k)}(x)$ at time grid points and then employ some integration scheme like the trapezoidal quadrature method.

More specifically, we first discretize the time interval $[-1, 1]$ by $2N_t + 1$ equally-spaced points $t_m := \frac{1}{N_t}m$ where $-N_t \le m \le N_t$. Then we use classical ODE integration schemes (we use RK4) to propagate the dynamics $\boldsymbol{X}_t(x)$ both forward and backward in time to estimate the following quantities:

$$\hat{U}_{0,m} = U_0\big(\boldsymbol{X}_{t_m}(x)\big), \quad \hat{U}_{1,m} = U_1\big(\boldsymbol{X}_{t_m}(x)\big), \quad \hat{D}_m = \boldsymbol{\nabla} \cdot \boldsymbol{b}\big(\boldsymbol{X}_{t_m}(x)\big).$$

Then

$$\mathcal{F}_{t_m}^{(k)}(x) \approx e^{-\hat{U}_{k,m} + \frac{1}{N_t}\text{sign}(m)\mathcal{Q}_{\text{Tr}}(\hat{D}_0, \hat{D}_1, \cdots, \hat{D}_m)},$$

where

$$\mathcal{Q}_{\text{Tr}}(\hat{D}_0, \hat{D}_1, \cdots, \hat{D}_m) = \sum_{j=0}^{m} \hat{D}_j - \frac{1}{2}\big(\hat{D}_0 + \hat{D}_m\big)$$

is the trapezoidal quadrature scheme. Then similarly, we can use the trapezoidal quadrature scheme to estimate $\int_{t_j - t_+}^{t_j - t_-} \mathcal{F}_s^{(0)}(x)\,\mathrm{d}s$ given the values $\mathcal{F}_{t_m}^{(0)}(x)$ for $-N_t \le m \le N_t$ and the time $t_j$ with

$t_j \in [t_-, t_+]$. After we have approximated values for $\dfrac{\mathcal{F}_{t_j}^{(1)}(x)}{\int_{t_j - t_+}^{t_j - t_-} \mathcal{F}_s^{(0)}(x)\,\mathrm{d}s}$, the trapezoidal quadrature

scheme is utilized again to approximate $\mathcal{A}_{t_-,t_+}(x)$. The computational cost is mostly dominated by the ODE integration, e.g., propagating $\boldsymbol{X}_t(x)$ or evaluating $\boldsymbol{\nabla} \cdot \boldsymbol{b}(\boldsymbol{X}_s(x))$ in general. Therefore, this straightforward integration-based method has linear computational cost with respect to $N_t$ and is thus expected to be optimal. Using the same principle, we can estimate the gradient of the second moment with respect to parameters in the dynamics (see (34)) during the training.

### I.4 An ODE-based forward propagation method to compute the estimator and its gradient for $t_- = 0, t_+ = 1$

To simplify notations, let us introduce

$$\mathscr{B}_t(x) := \int_{t-t_+}^{t-t_-} \mathcal{F}_s^{(0)}(x)\,\mathrm{d}s, \quad \text{and thus,} \quad \mathcal{A}_{t_-,t_+}(x) \overset{(8)}{=} \int_{t_-}^{t_+} \frac{\mathcal{F}_t^{(1)}(x)}{\mathscr{B}_t(x)}\,\mathrm{d}t, \tag{79}$$

which implicitly depends on $\boldsymbol{b}$. Let us denote $\alpha_t(x) := \int_{t_-}^{t} \frac{\mathcal{F}_s^{(1)}(x)}{\mathscr{B}_s(x)}\,\mathrm{d}s$.

### I.4.1 ODE dynamics to compute the estimator

Then the estimator $\mathcal{A}_{0,1}$ is simply $\alpha_1(x)$. To compute $\mathcal{A}_{0,1}$, we simply need to run the following ODE:

$$
\begin{cases}
\dfrac{\mathrm{d}}{\mathrm{d}t}\alpha_t(x) = e^{-U_1\left(\boldsymbol{X}_t(x)\right)}\mathcal{J}_t(x)/\mathscr{B}_t(x), & \alpha_0(x) = 0, \\[2mm]
\dfrac{\mathrm{d}}{\mathrm{d}t}\mathscr{B}_t(x) = e^{-U_0\left(\boldsymbol{X}_t(x)\right)}\mathcal{J}_t(x) - e^{-U_0\left(\boldsymbol{X}_t^{\mathrm{lag}}(x)\right)}\mathcal{J}_t^{\mathrm{lag}}(x), & \mathscr{B}_0(x) = \mathscr{B}_1^R(x), \\[2mm]
\dfrac{\mathrm{d}}{\mathrm{d}t}\boldsymbol{X}_t(x) = \boldsymbol{b}\left(\boldsymbol{X}_t(x)\right), & \boldsymbol{X}_0(x) = x, \\[2mm]
\dfrac{\mathrm{d}}{\mathrm{d}t}\boldsymbol{X}_t^{\mathrm{lag}}(x) = \boldsymbol{b}\left(\boldsymbol{X}_t^{\mathrm{lag}}(x)\right), & \boldsymbol{X}_0^{\mathrm{lag}}(x) = \boldsymbol{X}_1^R(x), \\[2mm]
\dfrac{\mathrm{d}}{\mathrm{d}t}\mathcal{J}_t(x) = \boldsymbol{\nabla}\cdot\boldsymbol{b}\left(\boldsymbol{X}_t(x)\right)\mathcal{J}_t(x), & \mathcal{J}_0(x) = 1, \\[2mm]
\dfrac{\mathrm{d}}{\mathrm{d}t}\mathcal{J}_t^{\mathrm{lag}}(x) = \boldsymbol{\nabla}\cdot\boldsymbol{b}\left(\boldsymbol{X}_t^{\mathrm{lag}}(x)\right)\mathcal{J}_t^{\mathrm{lag}}(x), & \mathcal{J}_0^{\mathrm{lag}}(x) = \mathcal{J}_1^R(x),
\end{cases}
\tag{80}
$$

where $\boldsymbol{X}_t^{\mathrm{lag}}(x) := \boldsymbol{X}_{t-1}(x)$ and $\mathcal{J}_t^{\mathrm{lag}}(x) := \mathcal{J}_{t-1}(x)$. Moreover, in order to obtain the initial condition, we shall run the dynamics backward in time, i.e., simulating the following ODE on $[0, 1]$,

$$
\begin{cases}
\dfrac{\mathrm{d}}{\mathrm{d}t}\boldsymbol{X}_t^R(x) = -\boldsymbol{b}\left(\boldsymbol{X}_t^R(x)\right), & \boldsymbol{X}_0^R(x) = x, \\[2mm]
\dfrac{\mathrm{d}}{\mathrm{d}t}\mathcal{J}_t^R(x) = -\boldsymbol{\nabla}\cdot\boldsymbol{b}\left(\boldsymbol{X}_t^R(x)\right)\mathcal{J}_t^R(x), & \mathcal{J}_0^R(x) = 1, \\[2mm]
\dfrac{\mathrm{d}}{\mathrm{d}t}\mathscr{B}_t^R(x) = e^{-U_0\left(\boldsymbol{X}_t^R(x)\right)}\mathcal{J}_t^R(x), & \mathscr{B}_0^R(x) = 0,
\end{cases}
\tag{81}
$$

where the superscript $R$ means the reversed process. As a remark, the auxiliary backward ODE (81) has dimension $d + 2$ and the ODE (80) has dimension $2d + 4$.

### I.4.2 ODE dynamics to compute the gradient

Denote $\vartheta$ as a vector containing parameters in $\boldsymbol{b}$ and let $N_p$ be the number of parameters. In what follows, we also vectorize all quantities involving $\vartheta$, e.g., for each parameter $\vartheta_j$, there is a corresponding $\boldsymbol{Y}_t^{(j)}(x)$ in (36) and we simply use the notation $\boldsymbol{Y}_t(x)$ to be a matrix whose $j^{\mathrm{th}}$ column is $\boldsymbol{Y}_t^j(x)$ to save notations; the same convention applies to other quantities.

Next, we shall similarly re-write the expression $\nabla_\vartheta\mathcal{A}_{0,1}$ (implicitly given in $\nabla_\vartheta\mathcal{M}_{0,1}(\boldsymbol{b})$ (34)) in terms of outputs from an ODE. Using the same idea, let us introduce

$$
\begin{cases}
\boldsymbol{D}_t(x) := \displaystyle\int_0^t \dfrac{\mathcal{G}_r^{(1)}(x)\int_{r-1}^r \mathcal{F}_s^{(0)}(x)\,\mathrm{d}s - \mathcal{F}_r^{(1)}(x)\int_{r-1}^r \mathcal{G}_s^{(0)}(x)\,\mathrm{d}s}{\left(\int_{r-1}^r \mathcal{F}_s^{(0)}(x)\,\mathrm{d}s\right)^2}\,\mathrm{d}r, \\[4mm]
\boldsymbol{g}_t(x) := \displaystyle\int_{t-1}^t \mathcal{G}_s^{(0)}(x)\,\mathrm{d}s, \\[4mm]
\boldsymbol{L}_t(x) := \displaystyle\int_0^t \nabla_\vartheta(\nabla_x\cdot\boldsymbol{b})\left(\boldsymbol{X}_s(x)\right)\,\mathrm{d}s, \\[4mm]
\boldsymbol{H}_t(x) := \displaystyle\int_0^t \left(\nabla(\boldsymbol{\nabla}\cdot\boldsymbol{b})(\boldsymbol{X}_s(x))\right)^T\boldsymbol{Y}_s(x)\,\mathrm{d}s,
\end{cases}
$$

and then we can rewrite quantities involved inside $\nabla_\vartheta\mathcal{M}_{0,1}(\boldsymbol{b})$ as follows:

the evolution of $\boldsymbol{X}_t(x),\,\boldsymbol{X}_t^{\mathrm{lag}}(x),\,\mathcal{J}_t(x),\,\mathcal{J}_t^{\mathrm{lag}}(x),\,\mathscr{B}_t(x),\,\alpha_t(x)$ in (80),

$$
\frac{\mathrm{d}}{\mathrm{d}t}\boldsymbol{D}_t(x) = \left(\begin{array}{c}
\dfrac{e^{-U_1\left(\boldsymbol{X}_t(x)\right)}\mathcal{J}_t(x)\left(-\nabla U_1\left(\boldsymbol{X}_t(x)\right)^T\boldsymbol{Y}_t(x) + \boldsymbol{H}_t(x) + \boldsymbol{L}_t(x)\right)}{\mathscr{B}_t(x)} \\[5mm]
-\dfrac{e^{-U_1\left(\boldsymbol{X}_t(x)\right)}\mathcal{J}_t(x)\boldsymbol{g}_t(x)}{\left(\mathscr{B}_t(x)\right)^2}
\end{array}\right), \quad \boldsymbol{D}_0(x) = \boldsymbol{0}_{N_p},
$$

$$\frac{\mathrm{d}}{\mathrm{d}t}\boldsymbol{g}_t(x) = \begin{pmatrix} e^{-U_0(\boldsymbol{X}_t(x))}\mathcal{J}_t(x)\big(-\nabla U_0(\boldsymbol{X}_t(x))^T\boldsymbol{Y}_t(x) \\ \qquad + \boldsymbol{H}_t(x) + \boldsymbol{L}_t(x)\big) \\ -e^{-U_0(\boldsymbol{X}_t^{\mathrm{lag}}(x))}\mathcal{J}_t^{\mathrm{lag}}(x)\big(-\nabla U_0(\boldsymbol{X}_t^{\mathrm{lag}}(x))^T\boldsymbol{Y}_t^{\mathrm{lag}}(x) \\ \qquad + \boldsymbol{H}_t^{\mathrm{lag}}(x) + \boldsymbol{L}_t^{\mathrm{lag}}(x)\big) \end{pmatrix}, \qquad \boldsymbol{g}_0(x) = \boldsymbol{g}_1^R(x),$$

$$\frac{\mathrm{d}}{\mathrm{d}t}\boldsymbol{H}_t(x) = \big(\nabla(\boldsymbol{\nabla}\cdot\boldsymbol{b})(\boldsymbol{X}_t(x))\big)^T\boldsymbol{Y}_t(x), \qquad \boldsymbol{H}_0(x) = \boldsymbol{0}_{N_p},$$

$$\frac{\mathrm{d}}{\mathrm{d}t}\boldsymbol{H}_t^{\mathrm{lag}}(x) = \big(\nabla(\boldsymbol{\nabla}\cdot\boldsymbol{b})(\boldsymbol{X}_t^{\mathrm{lag}}(x))\big)^T\boldsymbol{Y}_t^{\mathrm{lag}}(x), \qquad \boldsymbol{H}_0^{\mathrm{lag}}(x) = \boldsymbol{H}_1^R(x),$$

$$\frac{\mathrm{d}}{\mathrm{d}t}\boldsymbol{L}_t(x) = \nabla_\vartheta(\nabla_x\cdot\boldsymbol{b})(\boldsymbol{X}_t(x)), \qquad \boldsymbol{L}_0(x) = \boldsymbol{0}_{N_p},$$

$$\frac{\mathrm{d}}{\mathrm{d}t}\boldsymbol{L}_t^{\mathrm{lag}}(x) = \nabla_\vartheta(\nabla_x\cdot\boldsymbol{b})(\boldsymbol{X}_t^{\mathrm{lag}}(x)), \qquad \boldsymbol{L}_0^{\mathrm{lag}}(x) = \boldsymbol{L}_1^R(x),$$

$$\frac{\mathrm{d}}{\mathrm{d}t}\boldsymbol{Y}_t(x) = \nabla\boldsymbol{b}(\boldsymbol{X}_t(x))\boldsymbol{Y}_t(x) + \nabla_\vartheta\boldsymbol{b}(\boldsymbol{X}_t(x)), \qquad \boldsymbol{Y}_0(x) = \boldsymbol{0}_{d\times N_p},$$

$$\frac{\mathrm{d}}{\mathrm{d}t}\boldsymbol{Y}_t^{\mathrm{lag}}(x) = \nabla\boldsymbol{b}(\boldsymbol{X}_t^{\mathrm{lag}}(x))\boldsymbol{Y}_t^{\mathrm{lag}}(x) + \nabla_\vartheta\boldsymbol{b}(\boldsymbol{X}_t^{\mathrm{lag}}(x)), \qquad \boldsymbol{Y}_0^{\mathrm{lag}}(x) = \boldsymbol{Y}_1^R(x),$$

where the initial conditions come from solving an auxiliary backward equation on $[0,1]$, given as follows:

the evolution of $\boldsymbol{X}_t^R(x), \mathcal{J}_t^R(x), \mathscr{B}_t^R(x)$ in (81),

$$\frac{\mathrm{d}}{\mathrm{d}t}\boldsymbol{g}_t^R(x) = e^{-U_0(\boldsymbol{X}_t^R(x))}\mathcal{J}_t^R(x)\begin{pmatrix}-\nabla U_0(\boldsymbol{X}_t^R(x))^T\boldsymbol{Y}_t^R(x) \\ +\boldsymbol{H}_t^R(x) + \boldsymbol{L}_t^R(x)\end{pmatrix}, \qquad \boldsymbol{g}_0^R(x) = \boldsymbol{0}_{N_p},$$

$$\frac{\mathrm{d}}{\mathrm{d}t}\boldsymbol{H}_t^R(x) = -\big(\nabla(\boldsymbol{\nabla}\cdot\boldsymbol{b})(\boldsymbol{X}_t^R(x))\big)^T\boldsymbol{Y}_t^R(x), \qquad \boldsymbol{H}_0^R(x) = \boldsymbol{0}_{N_p},$$

$$\frac{\mathrm{d}}{\mathrm{d}t}\boldsymbol{L}_t^R(x) = -\nabla_\vartheta(\nabla_x\cdot\boldsymbol{b})(\boldsymbol{X}_t^R(x)), \qquad \boldsymbol{L}_0^R(x) = \boldsymbol{0}_{N_p},$$

$$\frac{\mathrm{d}}{\mathrm{d}t}\boldsymbol{Y}_t^R(x) = -\nabla\boldsymbol{b}(\boldsymbol{X}_t^R(x))\boldsymbol{Y}_t^R(x) - \nabla_\vartheta\boldsymbol{b}(\boldsymbol{X}_t^R(x)), \qquad \boldsymbol{Y}_0^R(x) = \boldsymbol{0}_{d\times N_p}.$$

As a remark, we combine the auxiliary states for all parameters together inside $\boldsymbol{Y}_t(x)$, so that all three terms $\boldsymbol{Y}_t(x), \boldsymbol{Y}_t^{\mathrm{lag}}(x), \boldsymbol{Y}_t^R(x)$ have dimension $d \times N_p$. Below is a table that summarizes the dimension of all components.

Table 2: This table summarizes all variables and their corresponding dimensions.

| notations | dimensions |
|---|---|
| $\mathcal{J}_t(x), \mathcal{J}_t^{\mathrm{lag}}(x), \mathscr{B}_t(x), \alpha_t(x), \mathcal{J}_t^R(x), \mathscr{B}_t^R(x)$ | 1 |
| $\boldsymbol{X}_t(x), \boldsymbol{X}_t^{\mathrm{lag}}(x), \boldsymbol{X}_t^R(x)$ | $d$ |
| $\boldsymbol{D}_t(x), \boldsymbol{g}_t(x), \boldsymbol{H}_t(x), \boldsymbol{H}_t^{\mathrm{lag}}(x), \boldsymbol{L}_t(x), \boldsymbol{L}_t^{\mathrm{lag}}(x), \boldsymbol{g}_t^R(x), \boldsymbol{L}_t^R(x), \boldsymbol{H}_t^R(x)$ | $N_p$ |
| $\boldsymbol{Y}_t(x), \boldsymbol{Y}_t^{\mathrm{lag}}(x), \boldsymbol{Y}_t^R(x)$ | $d \times N_p$ |

In the end, the outputs we need are

$$\mathcal{A}_{0,1}(x) = \alpha_1(x), \quad \nabla_\vartheta\mathcal{A}_{0,1}(x) = \boldsymbol{D}_1(x), \quad \text{so that } \nabla_\vartheta\mathcal{M}_{0,1}(\boldsymbol{b}) = 2\mathbb{E}_{\rho_0}\big[\alpha_1(x)\boldsymbol{D}_1(x)\big].$$

### I.5 A discussion on the backward propagation for differentiation

In the formulas (34) and (35), a crucial step is to evaluate

$$\Phi(t,x) := \big\langle\nabla\varphi(\boldsymbol{X}_t(x)), \boldsymbol{Y}_t(x)\big\rangle, \quad \varphi = U_k, \text{ or } \boldsymbol{\nabla}\cdot\boldsymbol{b}$$

for time $t \in [-T, T]$ where $T = t_+ - t_-$. For any fixed $t$, this value can be efficiently measured by the adjoint equation with backward propagation proposed in [9]. For instance, let us consider $t > 0$

and

$$\Phi(t,x) \stackrel{(39)}{=} \left\langle \nabla\varphi\big(\boldsymbol{X}_t(x)\big), \int_0^t \boldsymbol{C}_{t,s}(x)\,\delta\boldsymbol{b}\big(\boldsymbol{X}_s(x)\big)\,\mathrm{d}s \right\rangle$$

$$= \int_0^t \Big\langle \underbrace{\boldsymbol{C}_{t,s}(x)^T \nabla\varphi(\boldsymbol{X}_t(x))}_{=:\mathscr{A}(s,x)}, \delta\boldsymbol{b}\big(\boldsymbol{X}_s(x)\big) \Big\rangle\,\mathrm{d}s.$$

Moreover, it can be easily verified by the definition (40) that for $s \in [0,t]$,

$$\frac{\mathrm{d}}{\mathrm{d}s}\mathscr{A}(s,x) = -\nabla\boldsymbol{b}^T\big(\boldsymbol{X}_s(x)\big)\mathscr{A}(s,x), \quad \mathscr{A}(t,x) = \nabla\varphi\big(\boldsymbol{X}_t(x)\big).$$

The above two equations are Eqs. (4) & (5) in [9]. This can eliminate the need to simulate $\boldsymbol{Y}_t(x)$, whose memory cost scales like $\mathcal{O}(dN_p)$. However, in the finite-time NEIS scheme, we need to estimate $\Phi(t,x)$ not only for a fixed $t$ but also for $t \in [-1,1]$. Therefore, the computational cost scales like $\mathcal{O}(N_t^2)$ where $N_t$ is the number of time-discretization (or say the depth of the flow map in the normalizing flow context). As a comparison, the forward propagation method uses more memory but could be computationally cheaper as it only needs to visit the whole trajectory for $\mathcal{O}(1)$ times (see the table below).

Table 3: A comparison between the forward and backward propagation in computing the derivative of $\mathcal{M}_{0,1}(\boldsymbol{b})$ with respect to parameters; see (34) and (35). The notation $N_p$ is the number of parameters for the training and $N_t$ is the number of grid points in time-discretization.

| Method | Key difference | Memory | Computational cost |
|---|---|---|---|
| Backward propagation | simulate $\mathscr{A}(s,x)$ instead of $\boldsymbol{Y}_s(x)$ | $\mathcal{O}(N_p)$ | $\mathcal{O}(N_t^2)$ |
| Forward propagation | simulate $\boldsymbol{Y}_s(x)$ | $\mathcal{O}(dN_p)$ | $\mathcal{O}(N_t)$ |

## I.6 A discussion on query complexity

The query complexities to estimate $\mathcal{Z}_1$ by both AIS and NEIS are summarized in the Table 4 below:

Table 4: A summary of query complexities to $U_1$ when estimating $\mathcal{Z}_1$ for various methods. $K$ is the transition step in AIS; $N$ is the number of time steps ($\Delta t = 1/N$) for either integration-based or ODE-based discretization; $s$ is the order of ODE integration schemes. The integration-based method does not depend on $s$ as the query to $U_1$ is achieved during trapezoidal integral (see Appendix I.3). See the AIS-$K$ algorithm and relevant analysis in Appendix I.1

| | AIS-$K$ | ODE-based discretization (80) | Integration-based discretization (see Appendix I.3) |
|---|---|---|---|
| $U_1$ | $K+1$ | $sN$ | $2(N+1)$ |
| $\nabla U_1$ | $K+1$ | $0$ | $0$ |

## J  More training and comparisons results

Table 5: Comparison of NEIS with AIS: we include training and estimation query costs for AIS and NEIS, as well as statistics for 10 independent estimates of $\mathcal{Z}_1$; for each method, we first set the query cost to obtain one approximated value of $\mathcal{Z}_1$ and then accordingly choose the sample size; we repeatedly estimate $\mathcal{Z}_1$ 10 times and report the mean and std of these 10 estimates (in the form of mean $\pm$ std). For NEIS, we consider multiple random initializations for training and therefore, there are multiple estimates about $\mathcal{Z}_1$ in the last row in each panel (either 2 or 3 estimates). Estimates in AIS, however, simply refer to estimating results from independent experiments. The exact value $\mathcal{Z}_1 = 1$ and 1 MB $= 10^6$; the best result using AIS is colored in blue and the best result using NEIS is colored in green. The layer number $\ell = 2$ for both generic and gradient ansatz.

*Asymmetric 2-mode Gaussian mixture (2D):*

| | AIS-10 | AIS-100 | NEIS (Generic, $m = 20$) | NEIS (Generic, $m = 30$) | NEIS (Grad, $m = 20$) | NEIS (Grad, $m = 30$) |
|---|---|---|---|---|---|---|
| training cost | N/A | N/A | $U_1$: 2 MB $\nabla U_1$: 2.1 MB | | | |
| cost per estimate | $U_1$: $6.1 \sim 6.2$ MB $\nabla U_1$: $6.1 \sim 6.2$ MB | | $U_1$: 8.2 MB $\nabla U_1$: 0 | | | |
| 10 estimates | $1.106 \pm 0.409$ | $1.031 \pm 0.039$ | $1.004 \pm 0.008$ | $0.999 \pm 0.009$ | $0.998 \pm 0.006$ | $0.998 \pm 0.009$ |
| | $1.048 \pm 0.245$ | $1.016 \pm 0.097$ | $0.998 \pm 0.006$ | $0.999 \pm 0.007$ | $1.004 \pm 0.006$ | $1.001 \pm 0.007$ |

*Symmetric 4-mode Gaussian mixture (10D):*

| | AIS-10 | AIS-100 | NEIS (Grad, $m = 30$) | NEIS (Grad, $m = 40$) |
|---|---|---|---|---|
| training cost | N/A | N/A | $U_1$: 11.5 MB $\nabla U_1$: 12.8 MB | |
| cost per estimate | $U_1$: 48.6 MB $\nabla U_1$: 48.6 MB | | $U_1$: $72.9 \sim 73$ MB $\nabla U_1$: 0 | |
| 10 estimates | $0.991 \pm 0.074$ | $1.006 \pm 0.017$ | $0.998 \pm 0.004$ | $0.999 \pm 0.011$ |
| | $0.994 \pm 0.058$ | $1.000 \pm 0.014$ | $1.001 \pm 0.005$ | $0.997 \pm 0.008$ |

*Funnel distribution (10D):*

| | AIS-10 | AIS-100 | NEIS (linear (22a)) | NEIS (two-parametric (22b)) |
|---|---|---|---|---|
| training cost | N/A | N/A | $U_1$: 40.4 MB $\nabla U_1$: 48.8 MB | $U_1$: 20.2 MB $\nabla U_1$: 20.2 MB |
| cost per estimate | $U_1$: 178.3 MB $\nabla U_1$: 178.3 MB | | $U_1$: $267.5 \sim 267.7$ MB $\nabla U_1$: 0 | $U_1$: 121.2 MB $\nabla U_1$: 0 |
| 10 estimates | $0.682 \pm 0.024$ | $0.766 \pm 0.033$ | $0.820 \pm 0.098$ $0.853 \pm 0.057$ $0.894 \pm 0.187$ | $0.984 \pm 0.036$ |

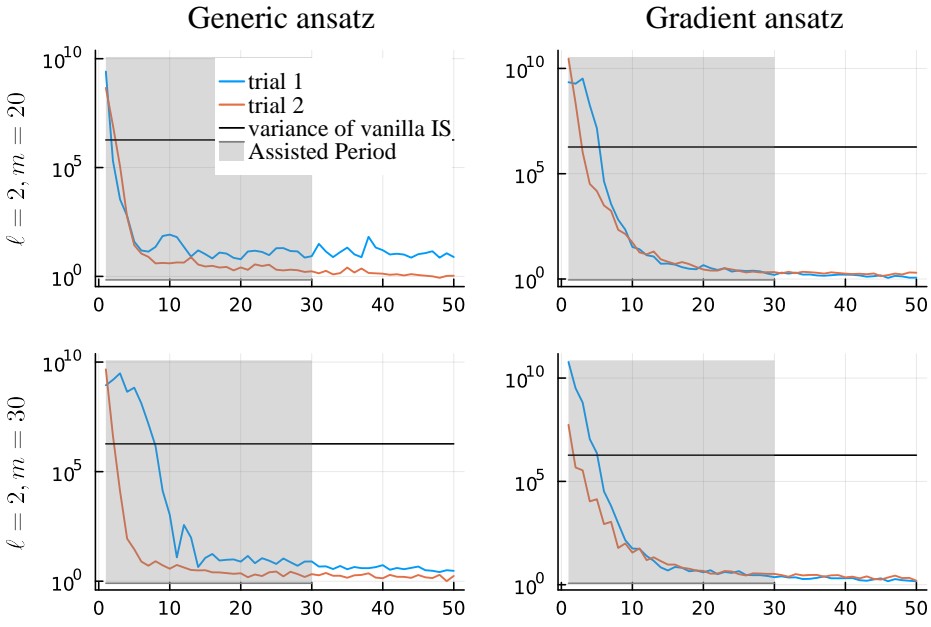

Figure 7: Asymmetric 2-mode Gaussian mixture in 2D: variance against SGD steps for the asymmetric 2-mode Gaussian mixture (20) in two trial runs. (8) was discretized with time step $\frac{1}{50}$. A mini-batch of sample size 200 was used throughout the training. The biasing parameters used in (19) were $\upsilon = 0.6$, $c = 0.1$ and $\varsigma = 1$ (see (19) and the follow-up paragraph).

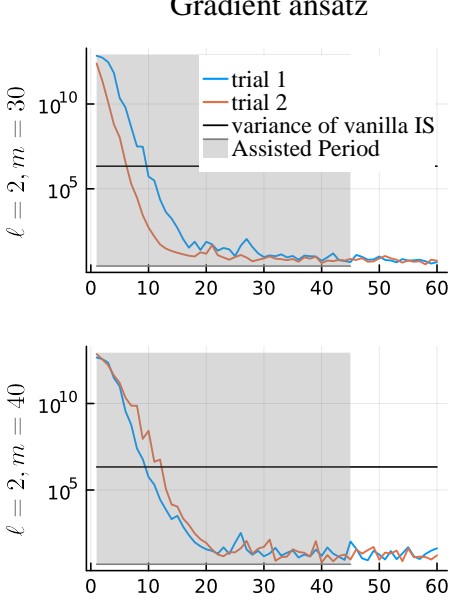

Figure 8: Symmetric 4-mode Gaussian mixture in 10D: variance as a function of SGD training step in 2 trial runs, with gradient form ansatz for $b$. (8) was discretized with time step $\frac{1}{60}$. The sample size of the mini-batch was 800 during the training. The biasing parameters were $\upsilon = 0.75$, $c = 0.3$ and $\varsigma = 1$ (see (19) and the follow-up paragraph).

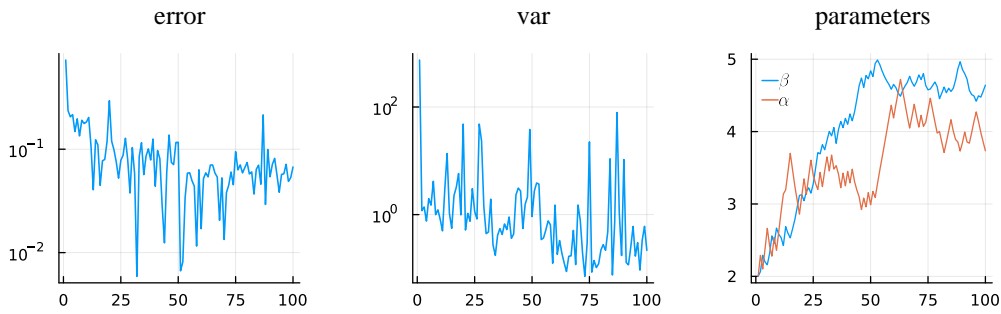

Figure 9: Funnel distribution in 10D: variance as a function of SGD training step in 3 trial runs. We consider the finite-time NEIS scheme with $t_- = -\frac{1}{2}$ and use the linear ansatz (22a). During the training, the integral inside $\mathcal{A}_{-\frac{1}{2},\frac{1}{2}}$ was discretized with time step $\frac{1}{100}$, and the mini-batch sample size was $10^3$. The biasing parameters were $\upsilon = 0.7$, $c = 0.3$ and $\varsigma = 1$ (see (19) and the follow-up paragraph).

Figure 10: Funnel distribution in 10D: we consider the finite-time NEIS scheme with $t_- = -\frac{1}{2}$ and use the two-parametric ansatz (22b). During the training, the integral inside $\mathcal{A}_{-\frac{1}{2},\frac{1}{2}}$ was discretized with time step $\frac{1}{100}$, and the mini-batch sample size was $10^3$. The direct training method was used.

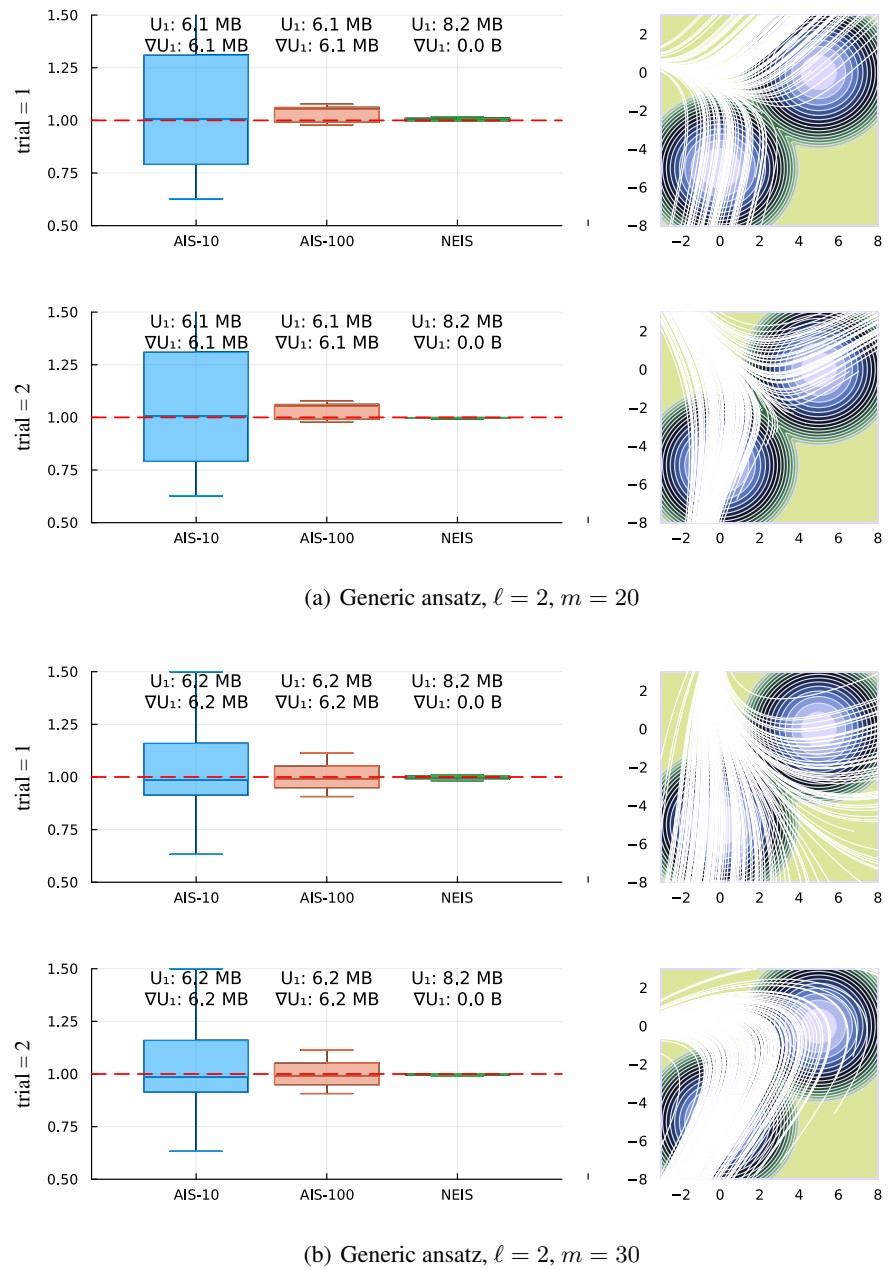

(a) Generic ansatz, $\ell = 2$, $m = 20$

(b) Generic ansatz, $\ell = 2$, $m = 30$

Figure 11: Asymmetric 2-mode Gaussian mixture in 2D: we consider the **generic ansatz** with $\ell = 2$ as the architecture for training. In the left part, we show a comparison of NEIS using optimized flow with AIS, under fixed query budget: we estimate $\mathcal{Z}_1$ using the above mentioned query budget for each method and then repeat the experiment 10 times; we show a boxplot of these 10 independent estimates for each method. In the right part, we plot a contour of $U_1$ together with streamlines of optimized flows. For each architecture, we use two random initializations for training, which refer to two trials above; the estimates using AIS are reused within each panel.

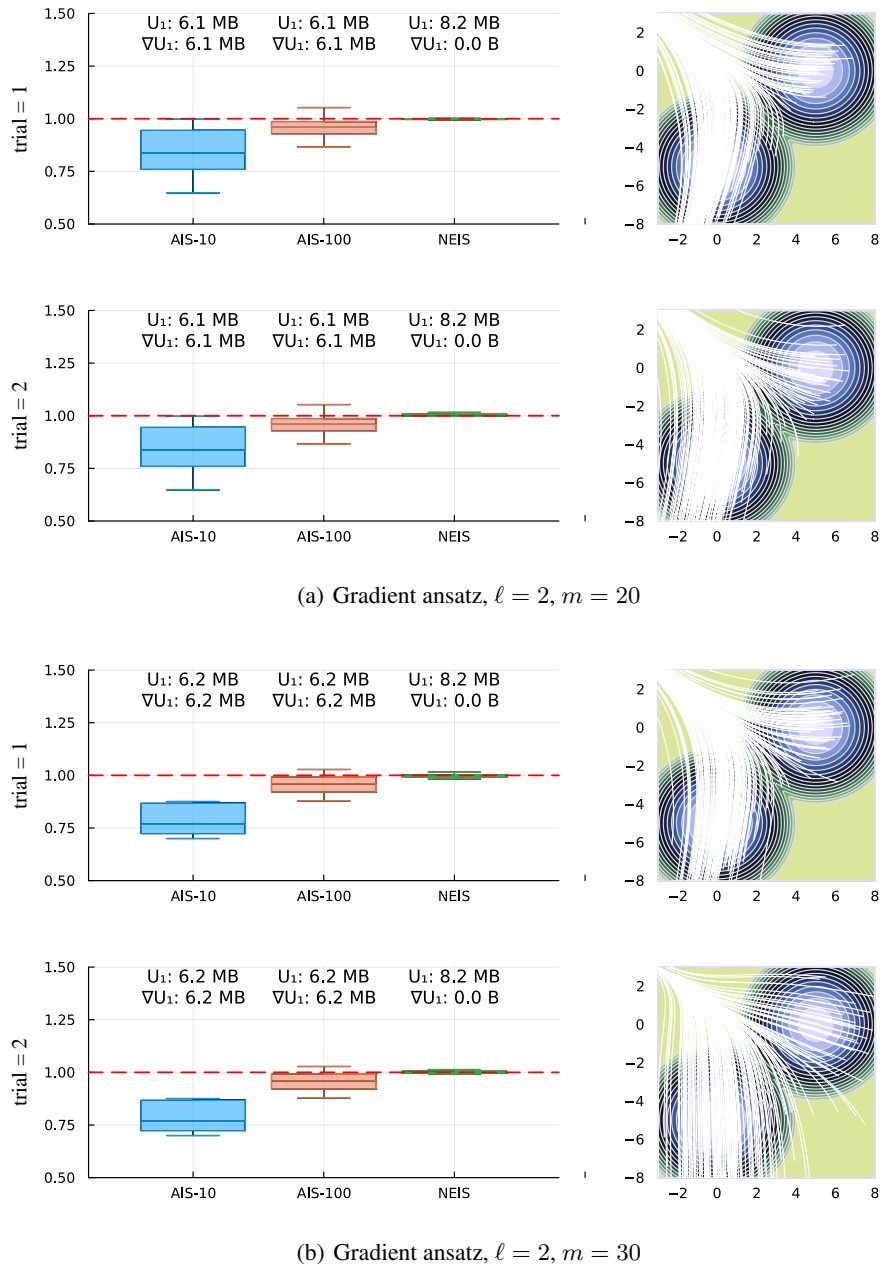

(a) Gradient ansatz, $\ell = 2$, $m = 20$

(b) Gradient ansatz, $\ell = 2$, $m = 30$

Figure 12: Asymmetric 2-mode Gaussian mixture in 2D: we consider the **gradient ansatz** with $\ell = 2$ as the architecture for training. In the left part, we show a comparison of NEIS using optimized flow with AIS, under fixed query budget: we estimate $\mathcal{Z}_1$ using the above mentioned query budget for each method and then repeat the experiment 10 times; we show a boxplot of these 10 independent estimates for each method. In the right part, we plot a contour of $U_1$ together with streamlines of optimized flows. For each architecture, we use two random initializations for training, which refer to two trials above; the estimates using AIS are reused within each panel.

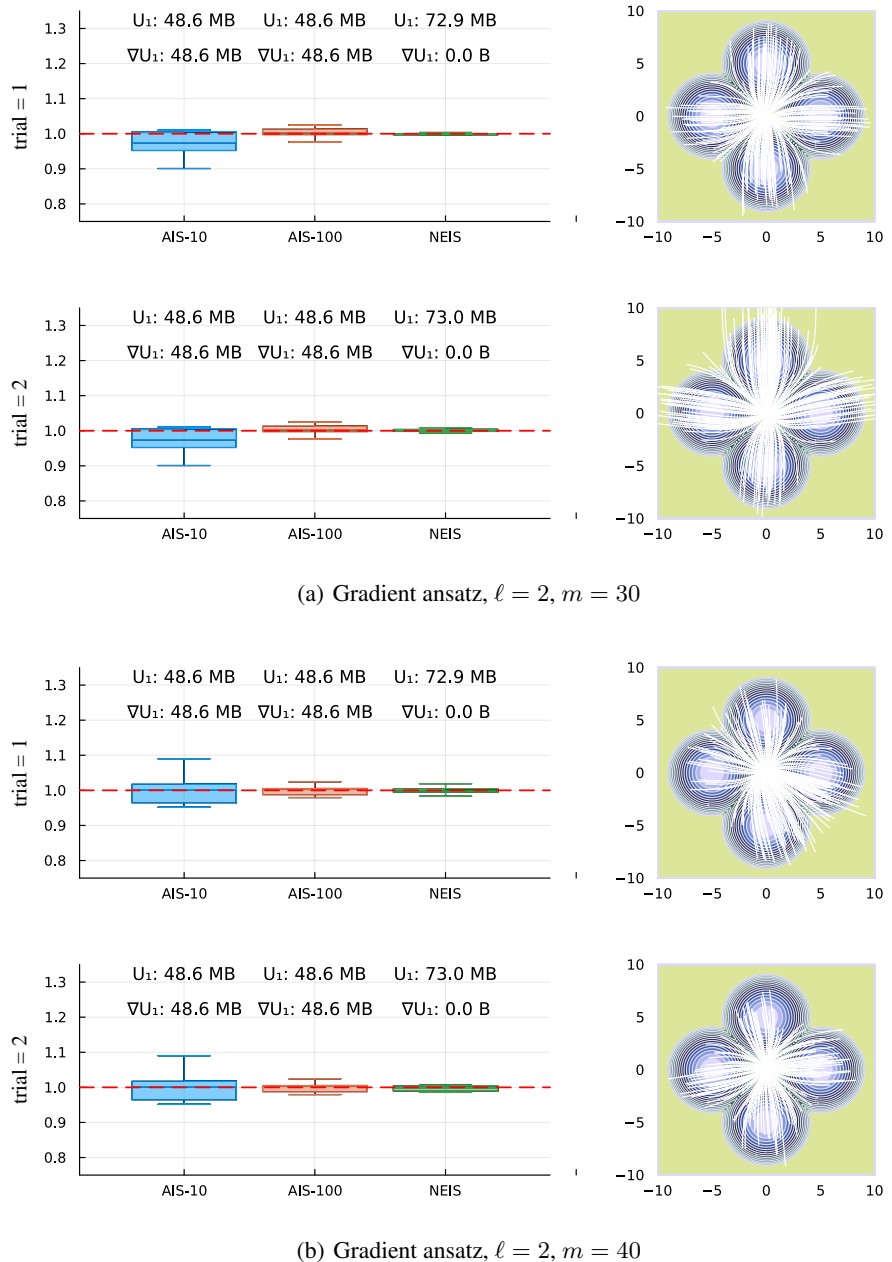

(a) Gradient ansatz, $\ell = 2$, $m = 30$

(b) Gradient ansatz, $\ell = 2$, $m = 40$

Figure 13: Symmetric 4-mode Gaussian mixture in 10D: we consider the **gradient ansatz** with $\ell = 2$ as the architecture for training. In the left part, we show a comparison of NEIS using optimized flow with AIS, under fixed query budget: we estimate $\mathcal{Z}_1$ using the above mentioned query budget for each method and then repeat the experiment 10 times; we show a boxplot of these 10 independent estimates for each method. In the right part, we plot a contour of projected $U_1$ (more specifically, the function $(x_1, x_2) \mapsto U_1\left(\begin{bmatrix} x_1 & x_2 & 0 \cdots \end{bmatrix}\right)$) together with streamlines of optimized flows projected to the $x_1$-$x_2$ plane. For each architecture, we use two random initializations for training, which refer to two trials above; the estimates using AIS are reused within each panel.

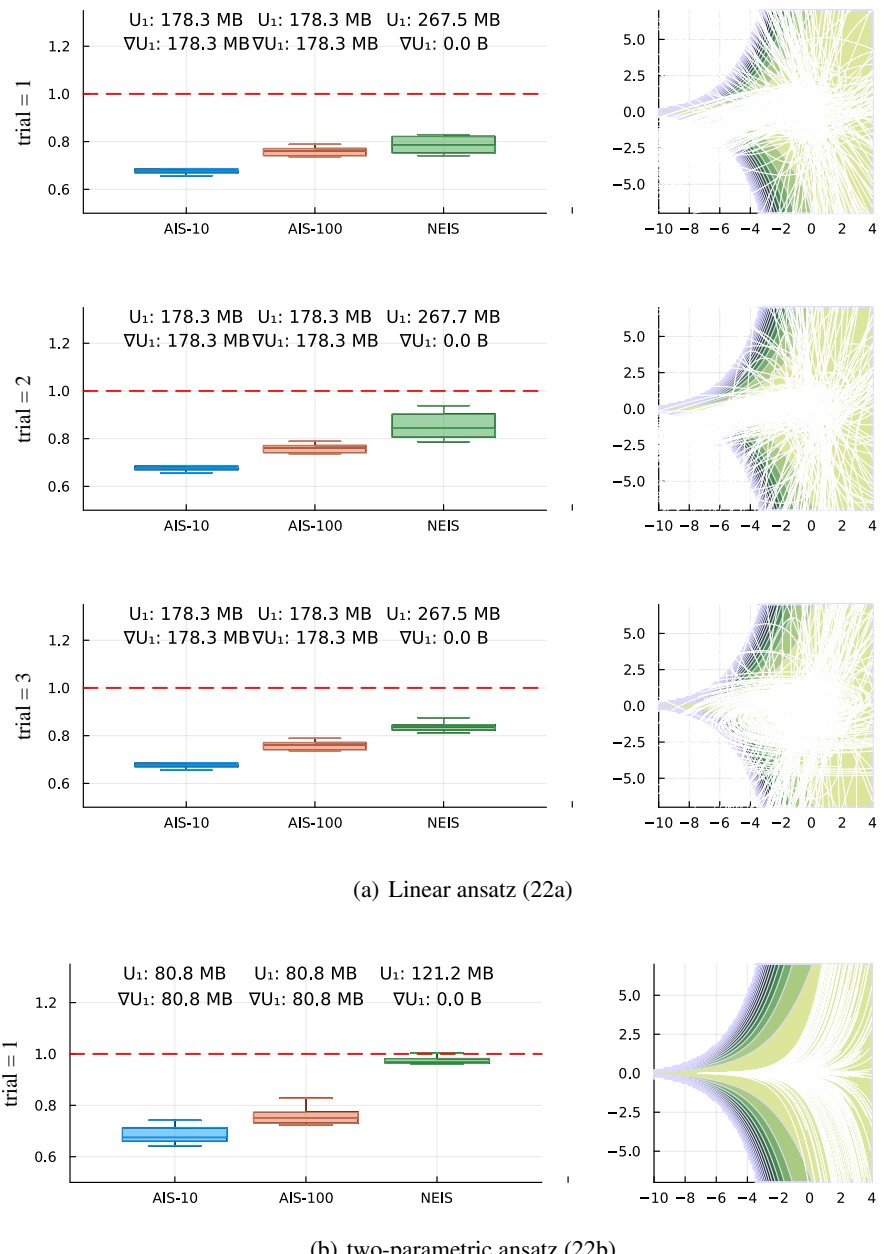

(a) Linear ansatz (22a)

(b) two-parametric ansatz (22b)

Figure 14: Funnel distribution in 10D: we consider the **generic linear ansatz** and a **two-parametric ansatz** as the architecture for training. In the left part, we show a comparison of NEIS using optimized flow with AIS, under fixed query budget: we estimate $\mathcal{Z}_1$ using the above mentioned query budget for each method and then repeat the experiment 10 times; we show a boxplot of these 10 independent estimates for each method. In the right part, we plot the contour of projected $\rho_1$ in log10 scale (more specifically, we plot the function $(x_1, x_2) \mapsto \log_{10}\left(\rho_1\left(\begin{bmatrix} x_1 & x_2 & 0 \cdots \end{bmatrix}\right)\right)$ together with streamlines of optimized flows projected to the $x_1$-$x_2$ plane. For the linear ansatz, we use three random initializations for training, which refer to three trials above; for the two-parametric ansatz, we only use a particular initialization (see the paragraph below (22b)), which refers to the only trial in part (b); the estimates using AIS are reused within each panel.