# OpenReview forum: "Learning Optimal Flows for Non-Equilibrium Importance Sampling"
_NeurIPS.cc/2022/Conference — NeurIPS 2022 Accept_

### Official Review · Reviewer_QJgR · 2022-06-29

**Rating:** 5
**Confidence:** 2
**Soundness:** 2 fair
**Presentation:** 3 good
**Contribution:** 3 good

**Summary:**

The paper has two main contributions. First, a theoretical analysis of Non-Equilibrium Importance Sampling (NEIS), a technique used to get unbiased estimates (in theory) of expectations and normalizing constants under intractable distributions, based on performing averages averages along the flow generated by the evolution of a dynamical system (given as a first order ODE). The main theoretical results proved in the paper involved showing the existence (under certain assumptions) of an optimal velocity field that yields an estimator with zero variance, and a connection with optimal transport maps. And second, the paper explains how NEIS with a learnable velocity field parameterized by a NN can be used in practice by training the method's parameters to minimize the estimator's variance, and shows an empirical evaluation on three low-dimensional models.

**Questions:**

- Have you tried using smaller batch-sizes?

- If I understand correctly, theoretical results (prop 3.1, prop 4.1) involve the formulation for equation 10, which integrates time from $-\infty$ to $\infty$. Do the results hold for the case where finite integration limits are used (i.e. eq. 8), which is the one used in practice?

**Limitations:**

See above regarding limitations.

**Strengths And Weaknesses:**

Strengths: The paper is written in a clear way, and presents original theoretical findings. The theoretical analysis answers relevant questions about NEIS, including conditions for the existence of the optimal dynamics that lead to an estimator with zero variance. All statements are clearly stated including all assumptions, and most assumptions are adequately discussed in the main text.

Weaknesses: In my opinion, the bigger weaknesses involve the empirical evaluation, the claims made around it, and the method's practicality. For instance:

- The conclusion claims that the empirical evaluation shows the method's effectiveness in high dimensions, when the evaluation is done on three models of dimensionalities 2, 10 and 10.

- The method appears to be extremely slow in practice, even for the simple 2d model, but this is not mentioned in the main text. It takes 1.5 hours to train for 200 SGD steps on the two dimensional model (mixture of two Gaussians) and 5 hours on one of the ten dimensional model (mixture of 4 Gaussians). I understand that the method uses neural networks, and simulations were done using CPUs, but all NNs are fairly small, of around 2 layers with 10 neurons each, so I don't think that should be a big bottleneck even when using CPUs. (Are training times on the funnel model, for which NNs are not used, reported?).

- The training uses minibatches of thousands of samples per training step, which may indicate that there are some issues with the variance of the objective and its gradients. This affects the cost of the method. Is it necessary to use such large minibatches?

- The vanilla estimator is taken to be importance sampling with a standard Gaussian as proposal, which is expected to work very poorly, specially since it is an extremely bad approximation of the targets used. I think that replacing this vanilla baseline with a diagonal Gaussian trained with VI, for instance, would provide a more interesting comparison.

Overall, I feel the empirical evaluation does not really show the practicality of the method, somewhat contradicting the claims made in the paper (if any of the statements above is wrong, please let me know). I think that more training details should be included in the main text explicitly (training times, sizes of the networks, steps used to numerically evaluate integrals and simulate dynamics, etc), as they are extremely relevant and currently only present in the Appendix. Also, I think that a more explicit discussion regarding the method's cost should be included, specially since the objective is somewhat complex, involving two nested integrals whose integrands depend on the values taken by a dynamical system at different times.

---

> ### Author Response · Authors · 2022-08-02
> **Author Response**
>
> We thank the referee for comments and review.
>
> - **(High dimension of the example)** We have changed the wording about high-dimension. We note however that $d=10$ already makes for challenging examples, as show by the results from the AIS estimator.
> - **(Comparing the costs of the methods)** This is quite tricky since the method tested are very different in nature and they involve computational steps whose cost is problem specific. AIS does not require any training, which makes it cheaper to use *a priori*, but its variance can be quite high  --  this why in our examples we see an apparent bias in the AIS estimator, because their asymptotic value depend on unlikely configurations. That is, running AIS for 1h helps, but not that much. In contrast, training the velocity field in NEIS is costly, but it may pay off ultimately. We believe that, the more challenging the example (e.g. the higher dimension), the higher the pay-off.
> - We have performed more comparison between NEIS and AIS under the same cost budget and NEIS still outperforms AIS, in particular, in funnel distribution. These results are shown in Fig. 9 in Appendix I and commented in the tables next to this figure. We will keep working on  comparisons at equal computational cost for the camera-ready version of our paper.
> - **(Training time for Neal's funnel distribution)** It is much shorter than in the other two cases, due to the simplicity of the class of velocity field chosen in the optimization. We have included the time in the revised version. In  the camera-ready version of our paper we  will also include results using a neural network to represent the velocity field in Neal's example.
> - **(About sample size of mini-batches)**.
>   - We deliberately choose $\rho_0$ in a way that $\rho_0$ has little overlap with $\rho_1$ and the variance for vanilla IS has the order like $10^6$ for the examples about Gaussian mixtures and it is even infinity for the Neal's funnel distribution. Therefore, we believe that a few thousand samples during the training is a reasonable size, considering the quite large variance.
>   - If the sample size is too small, the training scheme might not converge to the optima in a short time. We believe that the same issue will occur in e.g., neural importance sampling (incorporating normalizing flows into importance sampling).
> - **(About $\rho_0$)**. Thanks for suggesting replacing $\rho_0$ by diagonal Gaussian trained with VI. This is interesting to consider in experiments. We completely agree that a better $\rho_0$ can improve the sampling performance. We choose a $\rho_0$ not only for simplicity but also to test NEIS when $\rho_0$ and $\rho_1$ have very little overlap, which is a typical scenario that a vanilla IS fails.
> - **(Using finite vs infinite $t_\pm$)** Prop. 3.1 \& Prop 4.1 hold for unbounded time intervals, $t_\pm = \pm \infty$, and only show there exists a dynamics $\boldsymbol{b}^{\star}$ with zero-variance in the infinite-time NEIS, $\mathcal{A}^{\boldsymbol{b}^{\star}}(x) = Z_1$ for all $x\in \Omega$. However, if we choose $\boldsymbol{b} = \alpha \boldsymbol{b}^{\star}$ with $\alpha\to\infty$ into the finite-time scheme (with $-1<t_{-}<0< t_{+}<1$), then the estimator
> \begin{align*}
> 	\lim_{\alpha\to\infty} \mathcal{A}^{\boldsymbol{b}}_{t_{-}, t_{+}}(x) = \mathcal{A}^{\boldsymbol{b}^{\star}}(x) \equiv Z_1, \qquad \forall x\in\Omega.
> \end{align*}
> Therefore, no matter how we choose the values $t_{-}<0$ and $t_+>0$, we can find an almost zero-variance dynamics in the limit of $\alpha\to\infty$ for finite-time NEIS. We will clarify this point in the camera-ready version of our paper.

---

> > ### Comment · Reviewer_QJgR · 2022-08-04
> > **Thanks for the reply**
> >
> > Thanks for you reply. After reading the other reviews and responses, and checking the updated manuscript, my opinion is not changed much. I'm inclined to keeping my score. While I find the theoretical results interesting, I still have serious concerns about the method's practical applicability and about the presentation of the empirical evaluation (further details in the review).

---

> > > ### Author Response · Authors · 2022-08-07
> > > **Thanks and some clarifications**
> > >
> > > Clearly, many simpler MCMC algorithms exist that have wide (but not full) applicability in statistics and other fields. However, for us, one important motivation to the development of adaptive NEIS comes from its appealing and robust theoretical properties, a rare feat amongst importance sampling strategies that are often ad hoc. We believe that ultimately the very possibility that NEIS can be trained towards a zero variance estimator yields benefits that outweigh the cost of this training step, and will render possible calculations that would not be doable otherwise. Our aim was to demonstrate these features on nontrivial examples, without tailoring the procedure to them nor optimizing the training step (which obviously could be done in practical applications). We are confident that the training can be made more efficient and hope that our results will spur activities in that direction: we have noticed that simply by optimizing the implementation of gradient function, the training computational cost has already reduced to 15% of the original one (even without GPU acceleration).
> > >
> > > We also stress that it is easy to design examples in which AIS fails and NEIS works well. For instance, the estimator of AIS has very high variance in situations where, for some value of $\beta <1$, most of the mass of the density $\rho_0^{1-\beta} \rho_1^\beta$ is concentrated on a mode that ends up becoming insignificant on $\rho_1$ (i.e. at $\beta =1$), and another mode not seen at intermediate value of $\beta$ is the dominant one on $\rho_1$. In this case, it is hard  with AIS to get samples in the right mode (leading to apparent bias), and if we get any, they acquire very high weights (leading to high variance of the AIS estimator).  In contrast, NEIS will work in situations of this type as long as the velocity field pushes the mass from $\rho_0$ through this dominant mode of $\rho_1$. We did not include examples of this type that are purposely engineered to be unfavorable to AIS because our aim is not to criticize this method (whose shortcomings are known) but rather introduce an alternative.

---

### Official Review · Reviewer_vi5E · 2022-07-07

**Rating:** 7
**Confidence:** 2
**Soundness:** 3 good
**Presentation:** 3 good
**Contribution:** 3 good

**Summary:**

This paper considers non-equilibrium importance sampling (NEIS), wherein samples from a base distribution are evolved along flow lines and a particular quantity is averaged along these flow lines.  This results in a theoretically unbiased estimate of a normalizing constant (like in standard importance sampling), and since standard importance sampling can be derived as a special case of NEIS, this approach should allow one to construct lower variance estimators.  The authors derive a PDE which is satisfied by vector fields that produce optimal (zero variance) importance schemes, but then use a neural ODE approach to minimize the empirical variances of samples.  Finally, a few problems are considered where the NEIS method greatly reduces the variance of normalizing constant estimators compared to adaptive importance sampling.

**Questions:**

* How were the flow fields chosen for the direct GD method for Figures 2, 3 and 4?  in Figure 2 it is $-\nabla U_1$, in Figure 3 it is $-5\nabla U_1$, in Figure 4 it is $-2\nabla U_1$.
* Why was a neural ODE not used in the funnel toy model?  In general one does not know a priori what parameterization of the vector field will be good, and so using the neural ODE on the funnel model would be a good test-case for the behavior of the neural ODE + NEIS approach to pathological distributions that may arise in practice.
* In figures 2, 3, and 4 the NEIS-based estimators (especially Direct GD in figures 2 and 3, and the Linear (before model  and even Linear (after) model to some extent in Figure 4) show substantial bias.  This is at odds with the theoretical result that regardless of the vector field NEIS should be unbiased.  Is this due to the discretization of the ODE?  Or are there just rare events where a very large normalizing constant is estimated and the runs presented here got "unlucky" in having exclusively underestimated normalizing constants?  In any case, this disconnect between the theoretical result that all of these estimates should be unbiased and the observed substantial bias should be discussed / addressed.

**Limitations:**

In terms of limitations, please see the discussion above in "Strengths and Weaknesses" about issues surrounding discretization.  I agree with the authors that there are no obvious negative societal impacts of the work.

**Strengths And Weaknesses:**

Strengths:

* Overall I really enjoyed the paper.  The main results were technical, but clearly presented, and the approach seems to be promising.

* The theoretical results in the appendix were very thorough and I appreciated the examples of explicitly solvable dynamics and other concrete examples.

Weaknesses:

* A major weakness of the present paper, in my opinion, is the general lack of discussion of the impact of discretizing the flow along the vector field.  All of the theoretical results are about the idealized flow, but in practice the ODEs have to be solved numerically.  Presumably this introduces complicated biases and if the vector field is learned using a neural net, there is no obvious reason that the learned ODE needs to be amenable to easy and accurate numerical integration.  More discussion of and investigation of the impact of discretization would strengthen the paper, in my opinion.

* A minor point -- 3 different methods of using the neural ODE framing (directly estimating an arbitrary vector field, estimating a scalar potential and then taking gradients, or the divergence-free form) but then results were only presented from one of these for each toy problem (other than training curves in Figure 9).

* The training of the neural ODE seems difficult -- the two trials seem to rarely converge to similar optima in Figure 9, and the runtime of the training seems quite long -- in the hours that it takes to train the neural ODE to construct the vector field, many more samples could be drawn from the proposal distribution and used in a less fancy importance sampling scheme.  It would be good to consider comparisons of the various estimators in terms of accuracy per fixed amount of wall time (e.g., how does AIS or vanilla IS do if it just samples continuously for the amount of time it takes to train the NEIS vector field?).

* One other aspect that I felt was neglected was the importance of the proposal distribution, $\rho_0$.  In standard importance sampling, how to choose $\rho_0$ (from some family) is well-understood -- we want it to be as close to $\rho_1$ as possible, where closeness is measured by the chi divergence.  In the NEIS case, the optimal $\rho_0$ is less obvious -- presumably choosing it close to $\rho_1$ is useful, but one also wants the vector field to be "nice" and easy to learn.

Minor comments and Typos:

* Line 16: "order" --> "orders"
* Line 45: "based on Jarzynski" --> "based on the Jarzynski"
* Line 75: "order" --> "orders
* Line 77: "lead into" --> "lead to"
* Line 83: "one use in" --> "one used in"
* Line 168: "sampling procedure" --> "a sampling procedure"
* Line 188: "will necessarily produces" --> "will necessarily produce"
* Line 211: I believe the plural of "anstaz" is "anstazes"
* Line 227: "minimize" --> "minimizing"
* Figure 2b caption: "descend" --> "descent"
* Line 289: "descend" --> "descent"
* The notation $|\cdot|$ is used throughout.  I believe that this is the $\ell_2$ norm, but that should be made explicit.
* It should be made explicit that the minimization of equation (15) is over $V$

---

> ### Author Response · Authors · 2022-08-02
> **Author Response**
>
> We thank the referee for comments and review.
>
> - **(Lack of discussion of time discretization)** The discretization error coming from the numerical discretization of the ODE in NEIS may indeed introduce a bias: However this error can  be systematically controlled using more sophisticated integrator for the ODE. In the examples,  we used a high-order scheme (RK4) with a relatively small time step to ensure relatively accurate time-integration in ODEs and checked that discretization error introduce negligible/no bias.
> How to select the numerical scheme or tune time-step as a hyper-parameter in a systemic way are questions we leave for future investigation.
>
> - **(Choice of class for the velocity field)** Not all ansatzes are suitable universally and we only mention the one with more promising training outcomes. E.g., it is natural to expect that the divergence-free vector field (i.e., in-compressible flow) cannot reduce too much variance for extremely localized $\rho_1$.
>
> - **(Comparing the costs of the methods)** This is quite tricky since the method tested are very different in nature and they involve computational steps whose cost is problem specific. AIS does not require any training, which makes it cheaper to use *a priori*, but its variance can be quite high  --  this why in our examples we see an apparent bias in the AIS estimator, because their asymptotic value depend on unlikely configurations. That is, running AIS for 1h helps, but not that much. In contrast, training the velocity field in NEIS is costly, but it may pay off ultimately. We believe that, the more challenging the example (e.g. the higher dimension), the higher the pay-off.
>   - We have performed more comparison between NEIS and AIS under the same cost budget and NEIS still outperforms AIS, in particular, in funnel distribution. These results are shown in Fig. 9 in Appendix I and commented in the tables next to this figure. We will keep working on  comparisons at equal computational cost for the camera-ready version of our paper.
>
> - (Choice of $\rho_o$) We completely agree that a nicely chosen $\rho_0$ can substantially improve the sampling performance -- for example we could take the NEIS velocity field to be identically zero if $\rho_0=\rho_1$, and the closer $\rho_0$ from $\rho_1$, the simpler this velocity field should be.  We deliberately chose $\rho_0$ to be a standard Gaussian in the experiment to examine how NEIS works in an undesirable setup (e.g. one where the vanilla estimator for $\mathcal{Z}_1/\mathcal{Z}_0$ has extremely large or infinite variance).
>
>   - Still, we agree that the question of how to find an optimal $\rho_0$ to overall improve the training performance is interesting and we thank the referee for suggesting it; for example we could pick $\rho_0$ in some parametric class (e.g. Gaussian with adjustable mean and covariance) and include the tuning of these parameters in the optimization. For the examples treated in the paper, this generalization would clearly help.
>
> - (Choice of scaling of the gradient) The choice of the constant $\alpha\in\mathbb{R}$ for $\boldsymbol{b} = -\alpha \nabla U_1$ only matters  if $t_\pm $ are fixed.  We simply choose a reasonable one given the amplitude of $\nabla U_1$ so that the NEIS trajectories  travel far enough within and outside the core of the target $\rho_1$.
>
>   - We also stress that, when comparing NEIS (with training) to a direct gradient dynamics $\boldsymbol{b} = -\nabla U_1$,  our main goal is simply to suggest that the training scheme (with certain cost) can achieve much better variance reduction compared to a naive prescribed dynamics like a gradient flow of $U_1$.   We will clarify this point in the camera-ready version of our paper.
>
> - **(Choice of velocity class in Neal's funnel)** We used a linear parametrization of the velocity field simply to show that it is not always necessary to use neural networks if we have a reasonable guess of how the optimal velocity would look like -- indeed, the simpler the velocity, the easier the training (but potentially the worse the outcome if the class is too restrictive). That being said, we also trained with success a neural network to represent the velocity field in Neal's example.  We will add these results  in the camera-ready version of our paper.
>
> - **(Apparent bias of the estimators)** This apparent bias in the AIS estimator arises because its variance is very large and its asymptotic value depends on unlikely configurations.
> In particular, in Neal's funnel distribution, we believe the challenge is to effectively consider contributions from a local peak at the bottom of the funnel-shape (i.e., the extremely narrow region near $(x_1, 0, 0, \cdots, 0)$ where $x_1 \ll 0$). Therefore,  all methods in the experiment are more likely to provide an underestimates of the true value (though NEIS after training clearly perform better than AIS here).  We will clarify this point in the camera-ready version of our paper.

---

> > ### Comment · Reviewer_vi5E · 2022-08-03
> > **Acknowledgement of Response**
> >
> > Thank you for the thorough response.  I've updated my score for the paper to 7 based on the response and the understanding that the authors will add a more thorough comparison of efficiency per unit of wall time to a final version of the paper.

---

### Official Review · Reviewer_LyJz · 2022-07-11

**Rating:** 7
**Confidence:** 4
**Soundness:** 3 good
**Presentation:** 3 good
**Contribution:** 3 good

**Summary:**

The paper approaches the problem of normalization constant estimation for the case when the density of the target distribution is given as an unnormalized function. To be precise, the paper builds upon the Non-Equilibrium Importance Sampling (NEIS) [1]. Unlike the conventional Importance Sampling, NEIS takes the samples from the reference density (known normalized density) and propagates them along some vector field, which we are free to choose. One can obtain the resulting estimator of the normalization constant as the properly weighted average of the target density (unnormalized) along the sampled trajectories.

The main contribution of this paper is the following. The authors use the fact that NEIS yields an unbiased estimate for any choice of the vector field. Using this degree of freedom, they learn a parametric model of the vector field that minimizes the variance of the resulting estimator.

The main theoretical result of the paper is the achievability of the zero-variance estimate of the normalization constant. Namely, when we take the infinite time for the trajectory integration, there exists a vector field that yields a zero-variance NEIS estimator. Furthermore, the authors demonstrate that every local minimum for this problem is actually global. In practice, the authors consider only finite-time estimators, though.

For the empirical evaluation, the authors consider 2d, 10d mixtures of Gaussians, and 10d Funnel distribution. They demonstrate that the proposed method outperforms AIS and the original NEIS in a reasonable setting. Moreover, for the Funnel distribution, the authors parameterize the vector field as a linear model, significantly reducing the computational efforts.

1. Grant M. Rotskoff and Eric Vanden-Eijnden. Dynamical computation of the density of states and Bayes factors using nonequilibrium importance sampling. Phys. Rev. Lett., 122(15):150602, 2019. doi: 10.1103/PhysRevLett.122.150602.


**Questions:**

In the section "Strengths and Weaknesses," I gave my suggestion for the improvements of the experimental section.

Regarding the approach, I have the following question for the authors. Assuming the network can generate arbitrary vector fields, does it make sense to change the time interval $[t_-, t_+]$?

**Limitations:**

As I've mentioned in "Strengths and Weaknesses,"  the computational side of the proposed method is not fully addressed, and I would expect a comparison of the methods with the fixed computational budget.

**Strengths And Weaknesses:**

Overall, the paper does a decent job on every single aspect of a NeurIPS publication. It's clearly presented. It focuses on the intuition of the method in the main body and exhaustively describes all the details in the appendix. The theory is novel and immediately results in a motivated application. One might say that the empirical evaluation is not extensive. However, it's neat (properly compared with relevant algorithms) and demonstrates the clear superiority of the approach.

I feel that the weakest point of the method is its computational burden. Namely, backpropagation through the trajectories of ODEs is computationally expensive. Also, the unbiased estimate is formulated as a continuous integral, which, in practice, has to be discretized. While the latter seems to be not an issue since the integral is 1d, I could imagine that the computational efforts spent on learning might overweight the performance gains during the evaluation. I think that a comparison of the methods with the fixed computational budget is essential here (I haven't found the corresponding numbers in Appendix I).

---

> ### Author Response · Authors · 2022-08-02
> **Author Response**
>
> We thank the referee for comments and review.
>
> - **(Impact of disctretization, origin of the bias)** The discretization error coming from the numerical discretization of the ODE may introduce a bias: However this error can be systematically controlled using more sophisticated integrator for the ODE (a subject area that is very well mapped out in numerical analysis), especially since the velocity field in NEIS does not depend on time. In addition, the code profile suggests that most computational cost comes from computing high-order derivatives like  $\partial_{\theta}\nabla\cdot\boldsymbol{b}$ and $\partial_{\theta} \boldsymbol{b}$.
> Right now the code relies on automatic differentiation; a more explicit implementation should be able to provide a significant speed-up, which we will improve soon.
>
> - **(Comparing the costs of the methods)** This is quite tricky since the methods tested are very different in nature and they involve computational steps whose cost is problem specific. AIS does not require any training, which makes it cheaper to use *a priori*, but its variance can be quite high  --  this why in our examples we see an apparent bias in the AIS estimator, because their asymptotic value depend on unlikely configurations. That is, running AIS for 1h helps, but not that much (e.g., in funnel distribution). In contrast, training the velocity field in NEIS is costly, but it may pay off ultimately.
>
>   - We have performed more comparison between NEIS and AIS under the same cost budget and NEIS still outperforms AIS, in particular, in funnel distribution. These results are shown in Fig. 9 in Appendix I and commented in the tables next to this figure. We will keep working on  comparisons at equal computational cost for the camera-ready version of our paper.
>
> - **(Choice of $t_{\pm}$)** If we can generate arbitrary vector fields, then we can fix the  $t_-<0$ and $t_+>0$ as allowing them to vary only brings redundant parameterizations. Indeed using a dynamics $\alpha \boldsymbol{b}$ in an interval $[0,1]$ ($\alpha > 0$ is a constant) is the same as using a dynamics $\boldsymbol{b}$ in an interval $[0,\alpha]$.
> Therefore, we can without loss of generality assume $t_{\pm} = \pm 1/2$. In particular,
> from the existence result, we know there exists a dynamics $\boldsymbol{b}^{\star}$ with zero-variance in the infinite-time NEIS, $\mathcal{A}^{\boldsymbol{b}^{\star}}(x) = Z_1$ for all $x\in \Omega$. Then if we choose $\boldsymbol{b} = \alpha \boldsymbol{b}^{\star}$ with $\alpha\to\infty$ into the finite-time scheme (with $-1<t_{-}<0< t_{+}<1$), the estimator
> \begin{align*}
> 	\lim_{\alpha\to\infty} \mathcal{A}^{\boldsymbol{b}}_{t_{-}, t_{+}}(x) = \mathcal{A}^{\boldsymbol{b}^{\star}}(x) \equiv Z_1, \qquad \forall x\in\Omega.
> \end{align*}
> Therefore, no matter how we choose the values $t_{-}<0$ and $t_+>0$, we can find an almost zero-variance dynamics in the limit of $\alpha\to\infty$ for finite-time NEIS.  We will clarify this point in the camera-ready version of our paper.

---

> > ### Comment · Reviewer_LyJz · 2022-08-03
> > **I'm keeping my score**
> >
> > Thank you for the response. I went through other reviews and responses, and I would like to keep my score.
> >
> > I wouldn't agree that the comparison with AIS is tricky since you formally should just include the training time to the sampling generation time. Then you can check if AIS can get the same performance just by using this time for the sampling. However, I find the paper good enough even without this proper comparison.
> >
> > I agree with the other two points.

---

### Official Review · Reviewer_Hz5v · 2022-07-15

**Rating:** 6
**Confidence:** 3
**Soundness:** 3 good
**Presentation:** 3 good
**Contribution:** 4 excellent

**Summary:**

The paper analyzes a recently proposed non-equilibrium importance sampling approach to integrating / estimating the partition function of intractable densities. This approach provides an unbiased estimator with bounded variances typically much better than the variances provided by other estimators (which are in some cases infinite). Some experiments on basic multivariate densities including asymmetric and symmetric MoG, and Neal's funnel in 10d, provide empirical support.

**Questions:**

Why were these particular densities chosen for numerical examples?

If we ran AIS for 1h, could we get much more exhaustive exploration and tighter variance?

How does the training approach compare/contrast with diffusion models, where we are training a map that pushes a perturbed sample back toward the original sample?

Is the bias on Neal's funnel attributable to the variational approximation, or some other approximation in the procedure?

**Limitations:**

The academic nature of the examples is noted. Perhaps the expense of training ought to be more explicitly called out.

**Strengths And Weaknesses:**

Originality
The NEIS method seems to be new to the ML/stats community. The predecessor work in physics is only a couple years old. Connecting this work to Neural ODEs and neutral transport maps is a novel contribution, as are the variational approximations enabling these connections.

Quality
The math is developed reasonably well and connections to existing work are shown (might also think about connections to diffusion models, neural transport HMC, neural optimal transport). A possible complaint might be that the math is developed fairly quickly/compactly in the body of the paper, with >30pg of detail reserved to supplementaries (which I've only browsed).
A weaker portion of the paper, in my view, is the experiments. On the positive side, the experiments demonstrate fairly well the unbiased and low variance nature of this estimation approach. However, I would like to see some richer densities than [hierarchical/mixture] gaussians. Multiple libraries exist that could provide other interesting densities (inference gym, ppl bench, posterior db). One reason the authors might have chosen these particular densities is for analytically computable Z, which enables scaling all plots to a 0-1 size. But I do see a log-Gaussian cox process in the anon github, which suggests other densities might have been considered? The supplementary indicates that the method takes 1-2h to fit/optimize on these densities, which calls into question the fairness of comparison with 100 transitions of AIS (which should take only seconds unless I've misunderstood something). If we ran AIS for 1h, could we get much more exhaustive exploration and tighter variance? I might like to see plots of Z bias against number of density evaluations for each of the methods considered.

Clarity
I would have been helped by adding some prose about usage of $\nabla$ to the notation section. Presumably it is used throughout as an operator, but some notations like $\nabla \cdot$ (e.g. eq 5) are unfamiliar.

Significance
Because the approach seems to be fairly expensive and is only compared to much less expensive AIS, I am somewhat unclear about its practical significance. I do think the approach seems academically interesting to the NeurIPS audience and the numerical experiments indicate that, at least in cases where we are willing to pay compute, this provides an interesting new point on the frontier of approximate inference fidelity vs cost. I wonder whether comparisons with a trainable-RealNVP-reparameterized AIS/SMC might give a more fair comparison in terms of cost.

---

> ### Author Response · Authors · 2022-08-02
> **Author Response**
>
> We thank the referee for comments and review.
>
> - **(Choice of examples)** We have indeed tested the log-Gaussian cox process, but since the potential $U_1$ is convex in this example, traditional methods like AIS already works quite nicely on this example.
>   - The examples we choose are more challenging problems in which $\rho_0$ is intentionally not well-adapted to $\rho_1$ so that the variance of the naive estimator is very large or even infinite. In these examples NEIS after training can provide a significant variance reduction (like in the case of Neal's funnel).
>   - As for the Gaussian mixture densities, we choose them because we know the ground-truth (exact value of $Z$) so that we can better estimate the performence of NEIS (as well as to validate that the numerical experiments are correctly conducted).
>
> - **(Notations)** $\nabla$ is the gradient operator and $\nabla\cdot$ means the divergence for a vector field.
>
> - **(Running AIS longer.)**  The longer we run the simulations, the more accurate the results (with a decay in $O(1/\sqrt{n})$ if $n$ is the number of samples drawn from $\rho_0$). However the quality of the estimators are intrinsically limited by their variance, which is quite high for AIS in our examples -- this why, for instance, we see an apparent bias in the AIS estimator, because their asymptotic value depend on unlikely configurations that were not drawn in our experiments.
> As a result, running AIS for 1h, say, helps, but not that much (e.g., in funnel distribution). In contrast, training the velocity field in NEIS is costly, but it  pays off ultimately.  In Appendix I of the revised version, we have included new results making this point more clearly, and we will keep working on  comparisons at equal computational cost for the camera-ready version of our paper.
>
> - **(Comparison with diffusion models)** Diffusion models typically requires samples for $\rho_1$  to be trained -- this is the case for example in the context of score-based diffusion models that have proven very efficient for image generation.  However, the task herein is different as we assume that no or only few samples are available from $\rho_1$. In this case training diffusion models becomes harder.
>   - In addition, we believe that NEIS has an advantage over diffusion models (or the related transport models using e.g. normalizing flows): as explained in text. In particular,  variance of the NEIS estimator will be small  if samples likely on $\rho_0$ become likely on $\rho_1$ *sometime* along the NEIS trajectories,  rather than at the same fixed time for all the samples as in diffusion or transport models. The former requirement seems easier to fulfill than the latter.  We will clarify this point in the camera-ready version of our paper.
>
> - **(Bias in Neal's example)** This apparent bias in the AIS estimator arises because its variance is very large and its asymptotic value depend on unlikely configurations.  In particular, in Neal's funnel distribution, we believe the challenge is to effectively consider contributions from a local peak at the bottom of the funnel-shape (i.e., the extremely narrow region near $(x_1, 0, 0, \cdots, 0)$ where $x_1 \ll 0$). Therefore,  all methods in the experiment are more likely to provide an underestimates of the true value (though NEIS after training clearly perform better than AIS here).
>
> - **(Comparisons with a trainable-RealNVP-reparameterized AIS/SMC )** We thank the reviewer for this suggestion. We are working on a comparison between NEIS and other Real NVP parameterized AIS, e.g., annealed flow transport, and will try to include these results in the camera-ready version of our paper.

---

### Author Response · Authors · 2022-08-04
**General comment from authors**

We would like to begin by thanking the reviewers for their detailed reading and constructive criticism of our work. A consistent theme in the reviewer comments was that  our proposed method involves an expensive training step, and a better comparison of its cost compared to methods like AIS should be provided. In the revised version, we have added results in this direction. We stress however that a systematic cost comparison is tricky because  the methods tested are very different in nature and they involve computational steps whose cost is problem-specific. For example, AIS does not require any training, which makes it cheaper to use a priori, but its variance can be quite high  --  this why in our examples we see an apparent bias in the AIS estimator:  their asymptotic value depend on unlikely configurations that were not drawn in our experiments. In particular, running AIS for longer helps, but not that much, as shown in the revised  version. In contrast, training the velocity field in NEIS is costly, but it may pay off ultimately.  This is also better illustrated in the revised version, and we believe that, the more complicated the situation, the better  the trade-off in favor of NEIS will be. We also believe that the strong theoretical foundations that we provide for NEIS show the promises of the method and will motivate further studies to make it more efficient computationally.

---

### Meta-Review · Area_Chair_SDFt · 2022-08-25

**Recommendation:** Accept
**Confidence:** Certain

**Metareview:**

This paper studies the properties of a non-equilibium importance sampling method for unnormalized densities. The reviewers unanimously agreed the paper is well-written and the contribution is novel. Several reviewers expressed concerns on the practicality of the algorithm and that the empirical results do not adequately show the benefits of the method. Overall, the reviews felt that the rebuttal did not adequately address their concerns.

After discussion, the reviewers agreed that the paper is a worthwhile contribution and unanimously recommended acceptance. The final revision of the paper should address the limitations of the method more explicitly and tone down the claims about the empirical contribution and the method’s practicality. Another reviewer expressed that the paper needs to address the issue that in practice, numerical integration is required to solve ODEs and can lead to longer running times. Please revise the paper carefully based on the reviewers' feedback.


**Award:**

No

---

### Decision · Program_Chairs · 2022-09-14

Accept